# Denoising Diffusion Bridge Models

**Linqi Zhou**    **Aaron Lou**    **Samar Khanna**    **Stefano Ermon**
Department of Computer Science, Stanford University
`{linqizhou, aaronlou, samar.khanna, ermon}@stanford.edu`

## Abstract

Diffusion models are powerful generative models that map noise to data using stochastic processes. However, for many applications such as image editing, the model input comes from a distribution that is not random noise. As such, diffusion models must rely on cumbersome methods like guidance or projected sampling to incorporate this information in the generative process. In our work, we propose Denoising Diffusion Bridge Models (DDBMs), a natural alternative to this paradigm based on *diffusion bridges*, a family of processes that interpolate between two paired distributions given as endpoints. Our method learns the score of the diffusion bridge from data and maps from one endpoint distribution to the other by solving a (stochastic) differential equation based on the learned score. Our method naturally unifies several classes of generative models, such as score-based diffusion models and OT-Flow-Matching, allowing us to adapt existing design and architectural choices to our more general problem. Empirically, we apply DDBMs to challenging image datasets in both pixel and latent space. On standard image translation problems, DDBMs achieve significant improvement over baseline methods, and, when we reduce the problem to image generation by setting the source distribution to random noise, DDBMs achieve comparable FID scores to state-of-the-art methods despite being built for a more general task.

## 1 Introduction

Diffusion models are a powerful class of generative models which learn to reverse a diffusion process mapping data to noise (Sohl-Dickstein et al., 2015; Song and Ermon, 2019; Ho et al., 2020; Song et al., 2020b). For image generation tasks, they have surpassed GAN-based methods (Goodfellow et al., 2014) and achieved a new state-of-the-art for perceptual quality (Dhariwal and Nichol, 2021). Furthermore, these capabilities have spurred the development of modern text-to-image generative AI systems(Ramesh et al., 2022).

Despite these impressive results, standard diffusion models are ill-suited for other tasks. In particular, the diffusion framework assumes that the prior distribution is random noise, which makes it difficult to adapt to tasks such as image translation, where the goal is to map between pairs of images. As such, one resorts to cumbersome techniques, such as conditioning the model (Ho and Salimans, 2022; Saharia et al., 2021) or manually altering the sampling procedure (Meng et al., 2022; Song et al., 2020b). These methods are not theoretically principled and map in one direction (typically from corrupted to clean images), losing the cycle consistency condition (Zhu et al., 2017).

Instead, we consider methods which directly model a transport between two arbitrary probability distributions. This framework naturally captures the desiderata of image translation, but existing methods fall short empirically. For instance, ODE based flow-matching methods (Lipman et al., 2023; Albergo and Vanden-Eijnden, 2023; Liu et al., 2022a), which learn a deterministic path between two arbitrary probability distributions, have mainly been applied to image generation problems and have not been investigated for image translation. Furthermore, on image generation, ODE methods have not achieved the same empirical success as diffusion models. Schrödinger Bridge and models (De Bortoli et al., 2021) are another type of model which instead learn an entropic optimal transport between two probability distributions. However, these rely on expensive iterative approximation methods and have also found limited empirical use. More recent extensions including Diffusion Bridge Matching (Shi et al., 2023; Peluchetti, 2023) similarly require expensive iterative calculations.

In our work, we seek a scalable alternative that unifies diffusion-based unconditional generation methods and transport-based distribution translation methods, and we name our general framework Denoising Diffusion Bridge Models (DDBMs). We consider a reverse-time perspective of *diffusion bridges*, a diffusion process conditioned on given endpoints, and use this perspective to establish a general framework for distribution translation. We then note that this framework subsumes existing generative modeling paradigms such as score matching diffusion models (Song et al., 2020b) and flow matching optimal transport paths (Albergo and Vanden-Eijnden, 2023; Lipman et al., 2023; Liu et al., 2022a). This allows us to reapply many design choices to our more general task. In particular, we use this to generalize and improve the architecture pre-conditioning, noise schedule, and model sampler, minimizing input sensitivity and stabilizing performance. We then apply DDBMs to high-dimensional images using both pixel and latent space based models. For standard image translation tasks, we achieve better image quality (as measured by FID (Heusel et al., 2017)) and significantly better translation faithfulness (as measured by LPIPS (Zhang et al., 2018) and MSE). Furthermore, when we reduce our problem to image generation, we match standard diffusion model performance.

## 2 PRELIMINARIES

Recent advances in generative models have relied on the classical notion of transporting a data distribution $q_{\text{data}}(\mathbf{x})$ gradually to a prior distribution $p_{\text{prior}}(\mathbf{x})$ (Villani, 2008). By learning to reverse this process, one can sample from the prior and generate realistic samples.

### 2.1 GENERATIVE MODELING WITH DIFFUSION MODELS

**Diffusion process.** We are interested in modeling the distribution $q_{\text{data}}(\mathbf{x})$, for $\mathbf{x} \in \mathbb{R}^d$. We do this by constructing a diffusion process, which is represented by a set of time-indexed variables $\{\mathbf{x}_t\}_{t=0}^T$ such that $\mathbf{x}_0 \sim p_0(\mathbf{x}) := q_{\text{data}}(\mathbf{x})$ and $\mathbf{x}_T \sim p_T(\mathbf{x}) := p_{\text{prior}}(\mathbf{x})$. Here $q_{\text{data}}(\mathbf{x})$ is the initial "data" distribution and $p_{\text{prior}}(\mathbf{x})$ is the final "prior" distribution. The process can be modeled as the solution to the following SDE

$$d\mathbf{x}_t = \mathbf{f}(\mathbf{x}_t, t)dt + g(t)d\mathbf{w}_t \tag{1}$$

where $\mathbf{f} : \mathbb{R}^d \times [0, T] \to \mathbb{R}^d$ is vector-valued *drift* function, $g : [0, T] \to \mathbb{R}$ is a scalar-valued *diffusion* coefficient, and $\mathbf{w}_t$ is a Wiener process. Following this diffusion process forward in time constrains the final variable $\mathbf{x}_T$ to follow distribution $p_{\text{prior}}(\mathbf{x})$. The reverse of this process is given by

$$d\mathbf{x}_t = \mathbf{f}(\mathbf{x}_t, t) - g(t)^2 \nabla_{\mathbf{x}_t} \log p(\mathbf{x}_t))dt + g(t)d\mathbf{w}_t \tag{2}$$

where $p(\mathbf{x}_t) := p(\mathbf{x}_t, t)$ is the marginal distribution of $\mathbf{x}_t$ at time $t$. Furthermore, one can derive an equivalent deterministic process called the probability flow ODE (Song et al., 2020b), which has the same marginal distributions:

$$d\mathbf{x}_t = \left[ \mathbf{f}(\mathbf{x}_t, t) - \frac{1}{2}g(t)^2 \nabla_{\mathbf{x}_t} \log p(\mathbf{x}_t) \right] dt \tag{3}$$

In particular, one can draw $\mathbf{x}_T \sim q_{\text{data}}(\mathbf{y})$ and sample $q_{\text{data}}$ by solving either the above reverse SDE or ODE backward in time.

**Denoising score-matching.** The score, $\nabla_{\mathbf{x}_t} \log p(\mathbf{x}_t)$, can be learned by the score-matching loss

$$\mathcal{L}(\theta) = \mathbb{E}_{\mathbf{x}_t \sim p(\mathbf{x}_t|\mathbf{x}_0), \mathbf{x}_0 \sim q_{\text{data}}(\mathbf{x}), t \sim \mathcal{U}(0,T)} \left[ \|\mathbf{s}_\theta(\mathbf{x}_t, t) - \nabla_{\mathbf{x}_t} \log p(\mathbf{x}_t \mid \mathbf{x}_0)\|^2 \right] \tag{4}$$

such that the minimizer $\mathbf{s}_\theta^*(\mathbf{x}_t, t)$ of the above loss approximates the true score. Crucially, the above loss is tractable because the transition kernel $p(\mathbf{x}_t \mid \mathbf{x}_0)$, which depends on specific choices of drift and diffusion functions, is designed to be Gaussian $\mathbf{x}_t = \alpha_t \mathbf{x}_0 + \sigma_t \boldsymbol{\epsilon}$, where $\alpha_t$ and $\sigma_t$ are functions of time and $\boldsymbol{\epsilon} \sim \mathcal{N}(\mathbf{0}, \mathbf{I})$. It is also common to view the diffusion process in terms of the $\mathbf{x}_t$'s signal-to-noise ratio (SNR), defined as $\alpha_t^2/\sigma_t^2$.

### 2.2 DIFFUSION PROCESS WITH FIXED ENDPOINTS

Diffusion models are limited because they can only transport complex data distributions to a standard Gaussian distribution and cannot be naturally adapted to translating between two arbitrary distributions, *e.g.* in the case of image-to-image translation. Luckily, classical results have shown that one can condition a diffusion process on a fixed known endpoint via the famous Doob's $h$-transform:

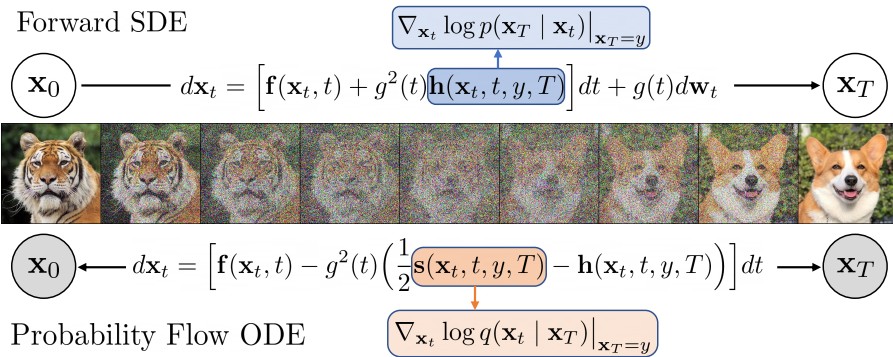

Figure 1: **A schematic for Denoising Diffusion Bridge Models.** DDBM uses a diffusion process guided by a drift adjustment (in blue) towards an endpoint $\mathbf{x}_T = y$. They lears to reverse such a bridge process by matching the denoising bridge score (in orange), which allows one to reverse from $\mathbf{x}_T$ to $\mathbf{x}_0$ for any $\mathbf{x}_T = \mathbf{y} \sim q_{\text{data}}(\mathbf{y})$. The forward SDE process shown on the top is unidirectional while the probability flow ODE shown at the bottom is deterministic and bidirectional. White nodes are stochastic while grey nodes are deterministic.

**Stochastic bridges via $h$-transform.** Specifically, a diffusion process defined in Eq. (1) can be driven to arrive at a particular point of interest $y \in \mathbb{R}^d$ almost surely via Doob's $h$-transform (Doob and Doob, 1984; Rogers and Williams, 2000),

$$d\mathbf{x}_t = \mathbf{f}(\mathbf{x}_t, t)dt + g(t)^2 \mathbf{h}(\mathbf{x}_t, t, y, T) + g(t)d\mathbf{w}_t, \quad \mathbf{x}_0 \sim q_{\text{data}}(\mathbf{x}), \quad \mathbf{x}_T = y \tag{5}$$

where $\mathbf{h}(x, t, y, T) = \nabla_{\mathbf{x}_t} \log p(\mathbf{x}_T \mid \mathbf{x}_t)\big|_{\mathbf{x}_t = x, \mathbf{x}_T = y}$ is the gradient of the log transition kernel of from $t$ to $T$ generated by the original SDE, evaluated at points $\mathbf{x}_t = x$ and $\mathbf{x}_T = y$, and each $\mathbf{x}_t$ now explicitly depends on $y$ at time $T$. Furthermore, $p(\mathbf{x}_T = y \mid \mathbf{x}_t)$ satisfies the Kolmogorov backward equation (specified in Appendix A). With specific drift and diffusion choices, *e.g.* $\mathbf{f}(\mathbf{x}_t, t) = \mathbf{0}$, $\mathbf{h}$ is tractable due to the tractable (Gaussian) transition kernel of the underlying diffusion process.

When the initial point $\mathbf{x}_0$ is fixed, the process is often called a *diffusion bridge* (Särkkä and Solin, 2019; Heng et al., 2021; Delyon and Hu, 2006; Schauer et al., 2017; Peluchetti; Liu et al., 2022b), and its ability to connect any given $\mathbf{x}_0$ to a given value of $\mathbf{x}_T$ is promising for image-to-image translation. Furthermore, the transition kernel may be tractable, which serves as further motivation.

## 3 DENOISING DIFFUSION BRIDGE MODELS

Assuming that the endpoints of a diffusion bridge both exist in $\mathbb{R}^d$ and come from an arbitrary and unknown joint distribution, *i.e.* $(\mathbf{x}_0, \mathbf{x}_T) = (\mathbf{x}, \mathbf{y}) \sim q_{\text{data}}(\mathbf{x}, \mathbf{y})$, we wish to devise a process that learns to approximately sample from $q_{\text{data}}(\mathbf{x} \mid \mathbf{y})$ by reversing the diffusion bridge with boundary distribution $q_{\text{data}}(\mathbf{x}, \mathbf{y})$, given a training set of *paired* samples drawn from $q_{\text{data}}(\mathbf{x}, \mathbf{y})$.

### 3.1 TIME-REVERSED SDE AND PROBABILITY FLOW ODE

Inspired by diffusion bridges, we construct the stochastic process $\{\mathbf{x}_t\}_{t=0}^T$ with marginal distribution $q(\mathbf{x}_t)$ such that $q(\mathbf{x}_0, \mathbf{x}_T)$ approximates $q_{\text{data}}(\mathbf{x}_0, \mathbf{x}_T)$. Reversing the process amounts to sampling from $q(\mathbf{x}_t \mid \mathbf{x}_T)$. Note that distribution $q(\cdot)$ is different from $p(\cdot)$, *i.e.* the diffusion marginal distribution, in that the endpoint distributions are now $q_{\text{data}}(\mathbf{x}_0, \mathbf{x}_T) = q_{\text{data}}(\mathbf{x}, \mathbf{y})$ instead of the distribution of a diffusion $p(\mathbf{x}_0, \mathbf{x}_T) = p(\mathbf{x}_T \mid \mathbf{x}_0)q_{\text{data}}(\mathbf{x}_0)$, which defines a Gaussian $\mathbf{x}_T$ given $\mathbf{x}_0$. We can construct the time-reversed SDE/probability flow ODE of $q(\mathbf{x}_t \mid \mathbf{x}_T)$ via the following theorem.

**Theorem 1.** *The evolution of conditional probability $q(\mathbf{x}_t \mid \mathbf{x}_T)$ has a time-reversed SDE of the form*

$$d\mathbf{x}_t = \left[\mathbf{f}(\mathbf{x}_t, t) - g^2(t)\Big(\mathbf{s}(\mathbf{x}_t, t, y, T) - \mathbf{h}(\mathbf{x}_t, t, y, T)\Big)\right]dt + g(t)d\hat{\mathbf{w}}_t, \quad \mathbf{x}_T = y \tag{6}$$

*with an associated probability flow ODE*

$$d\mathbf{x}_t = \left[\mathbf{f}(\mathbf{x}_t, t) - g^2(t)\Big(\frac{1}{2}\mathbf{s}(\mathbf{x}_t, t, y, T) - \mathbf{h}(\mathbf{x}_t, t, y, T)\Big)\right]dt, \quad \mathbf{x}_T = y \tag{7}$$

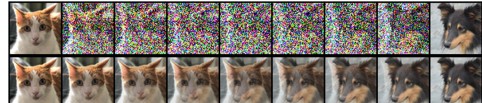 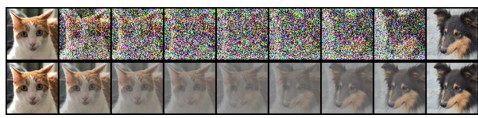

Figure 2: VE bridge (left) and VP bridge (right) with their SDE (top) and ODE (bottom) visualization.

| | $\mathbf{f}(\mathbf{x}_t, t)$ | $g^2(t)$ | $p(\mathbf{x}_t \mid \mathbf{x}_0)$ | $\text{SNR}_t$ | $\nabla_{\mathbf{x}_t} \log p(\mathbf{x}_T \mid \mathbf{x}_t)$ |
|---|---|---|---|---|---|
| VP | $\frac{d \log \alpha_t}{dt} \mathbf{x}_t$ | $\frac{d}{dt} \sigma_t^2 - 2 \frac{d \log \alpha_t}{dt} \sigma_t^2$ | $\mathcal{N}(\alpha_t \mathbf{x}_0, \sigma_t^2 \boldsymbol{I})$ | $\alpha_t^2 / \sigma_t^2$ | $\frac{(\alpha_t/\alpha_T)\mathbf{x}_T - \mathbf{x}_t}{\sigma_t^2 (\text{SNR}_t/\text{SNR}_T - 1)}$ |
| VE | $\mathbf{0}$ | $\frac{d}{dt} \sigma_t^2$ | $\mathcal{N}(\mathbf{x}_0, \sigma_t^2 \boldsymbol{I})$ | $1/\sigma_t^2$ | $\frac{\mathbf{x}_T - \mathbf{x}_t}{\sigma_T^2 - \sigma_t^2}$ |

Table 1: VP and VE instantiations of diffusion bridges.

*on $t \le T - \epsilon$ for any $\epsilon > 0$, where $\hat{\mathbf{w}}_t$ denotes a Wiener process, $\mathbf{s}(x, t, y, T) = \nabla_{\mathbf{x}_t} \log q(\mathbf{x}_t \mid \mathbf{x}_T)\big|_{\mathbf{x}_t = x, \mathbf{x}_T = y}$ and $\mathbf{h}$ is as defined in Eq. (5).*

A schematic of the bridge process is shown in Figure 1. Note that this process is defined up to $T - \epsilon$. To recover the initial distribution in the SDE case, we make an approximation that $\mathbf{x}_{T-\epsilon} \approx y$ for some small $\epsilon$ simulate SDE backward in time. For the ODE case, since we need to sample from $p(\mathbf{x}_{T-\epsilon})$ which cannot be Dirac delta, we cannot approximate $\mathbf{x}_{T-\epsilon}$ with a single $y$. Instead, we can first approximate $\mathbf{x}_{T-\epsilon'} \approx y$ where $\epsilon > \epsilon' > 0$, and then take an Euler-Maruyama step to $\mathbf{x}_{T-\epsilon}$, and Eq. (7) can be used afterward. A toy visualization of VE bridge and VP bridges are shown in Figure 2. The top and bottom shows the respective SDE and ODE paths for VE and VP bridges.

## 3.2 MARGINAL DISTRIBUTIONS AND DENOISING BRIDGE SCORE MATCHING

The sampling process in Theorem 1 requires approximation of the score $\mathbf{s}(x, t, y, T) = \nabla_{\mathbf{x}_t} \log q(\mathbf{x}_t \mid \mathbf{x}_T)\big|_{\mathbf{x}_t = x, \mathbf{x}_T = y}$ where $q(\mathbf{x}_t \mid \mathbf{x}_T) = \int_{\mathbf{x}_0} q(\mathbf{x}_t \mid \mathbf{x}_0, \mathbf{x}_T) q_{\text{data}}(\mathbf{x}_0 \mid \mathbf{x}_T) d\mathbf{x}_0$. However, as the true score is not known in closed-form, we take inspiration from denoising score-matching (Song et al., 2020b) and use a neural network to approximate the true score by matching against a tractable quantity. This usually results in closed-form marginal sampling of $\mathbf{x}_t$ given data (*e.g.* $\mathbf{x}_0$ in the case of diffusion models and $(\mathbf{x}_0, \mathbf{x}_T)$ in our case), and given $\mathbf{x}_t$, the model is trained to match against the closed-form denoising score objective. We are motivated to follow a similar approach because (1) tractable marginal sampling of $\mathbf{x}_t$ and (2) closed-form objectives enable a simple and scalable algorithm. We specify how to design the marginal sampling distribution and the tractable score objective below to approximate the ground-truth conditional score $\nabla_{\mathbf{x}_t} \log q(\mathbf{x}_t \mid \mathbf{x}_T)$.

**Sampling distribution.** Fortunately, for the former condition, we can design our sampling distribution $q(\cdot)$ such that $q(\mathbf{x}_t \mid \mathbf{x}_0, \mathbf{x}_T) := p(\mathbf{x}_t \mid \mathbf{x}_0, \mathbf{x}_T)$, where $p(\cdot)$ is the diffusion distribution pinned at both endpoints as in Eq. (5). For diffusion processes with Gaussian transition kernels, *e.g.* VE, VP (Song et al., 2020b), our sampling distribution is a Gaussian distribution of the form

$$q(\mathbf{x}_t \mid \mathbf{x}_0, \mathbf{x}_T) = \mathcal{N}(\hat{\mu}_t, \hat{\sigma}_t^2 \boldsymbol{I}), \quad \text{where}$$
$$\hat{\mu}_t = \frac{\text{SNR}_T}{\text{SNR}_t} \frac{\alpha_t}{\alpha_T} \mathbf{x}_T + \alpha_t \mathbf{x}_0 (1 - \frac{\text{SNR}_T}{\text{SNR}_t})$$
$$\hat{\sigma}_t^2 = \sigma_t^2 (1 - \frac{\text{SNR}_T}{\text{SNR}_t}) \tag{8}$$

where $\alpha_t$ and $\sigma_t$ are pre-defined signal and noise schedules and $\text{SNR}_t = \alpha_t^2 / \sigma_t^2$ is the signal-to-noise ratio at time $t$. For VE schedule, we assume $\alpha_t = 1$ and derivation details are provided in Appendix A.1. Notably, the mean of this distribution is a linear interpolation between the (scaled) endpoints, and the distribution approaches a Dirac distribution when nearing either end. For concreteness, we present the bridge processes generated by both VP and VE diffusion in Table 1 and recommend choosing $\mathbf{f}$ and $g$ specified therein.

**Training objective.** For the latter condition, diffusion bridges benefit from a similar setup as in diffusion models, since a pre-defined signal/noise schedule gives rise to a closed-form conditional score $\nabla_{\mathbf{x}_t} \log q(\mathbf{x}_t \mid \mathbf{x}_0, \mathbf{x}_T)$. We show in the following theorem that with $\mathbf{x}_t \sim q(\mathbf{x}_t \mid \mathbf{x}_0, \mathbf{x}_T)$, a neural network $\mathbf{s}_\theta(\mathbf{x}_t, \mathbf{x}_T, t)$ that matches against this closed-form score approximates the true score.

**Theorem 2** (Denoising Bridge Score Matching). *Let* $(\mathbf{x}_0, \mathbf{x}_T) \sim q_{\text{data}}(\mathbf{x}, \mathbf{y})$, $\mathbf{x}_t \sim q(\mathbf{x}_t \mid \mathbf{x}_0, \mathbf{x}_T)$, $t \sim p(t)$ *for any non-zero time sampling distribution* $p(t)$ *in* $[0, T]$, *and* $w(t)$ *be a non-zero loss weighting term of any choice. Minimum of the following objective:*

$$\mathcal{L}(\theta) = \mathbb{E}_{\mathbf{x}_t, \mathbf{x}_0, \mathbf{x}_T, t}\Big[w(t)\|\mathbf{s}_\theta(\mathbf{x}_t, \mathbf{x}_T, t) - \nabla_{\mathbf{x}_t} \log q(\mathbf{x}_t \mid \mathbf{x}_0, \mathbf{x}_T)\|^2\Big] \tag{9}$$

*satisfies* $\mathbf{s}_\theta(\mathbf{x}_t, \mathbf{x}_T, t) = \nabla_{\mathbf{x}_t} \log q(\mathbf{x}_t \mid \mathbf{x}_T)$.

In short, we establish a tractable diffusion bridge over two endpoints and, by matching the conditional score of the Gaussian bridge, we can learn the score of the new distribution $q(\mathbf{x}_t \mid \mathbf{x}_T)$ that satisfies the boundary distribution $q_{\text{data}}(\mathbf{x}, \mathbf{y})$.

## 4 GENERALIZED PARAMETERIZATION FOR DISTRIBUTION TRANSLATION

Building the bridge process upon diffusion process allows us to further adapt many recent advancements in the score network parameterization $\mathbf{s}_\theta(\mathbf{x}_t, \mathbf{x}_T, t)$ (Ho et al., 2020; Song et al., 2020b; Salimans and Ho, 2022; Ho et al., 2022; Karras et al., 2022), different noise schedules, and efficient ODE sampling (Song et al., 2020a; Karras et al., 2022; Lu et al., 2022a;b; Zhang and Chen, 2022) to our more general framework. Among these works, EDM (Karras et al., 2022) proposes to parameterize the model output to be $D_\theta(\mathbf{x}_t, t) = c_{\text{skip}}(t)\mathbf{x}_t + c_{\text{out}}(t)F_\theta(c_{\text{in}}(t)\mathbf{x}_t, c_{\text{noise}}(t))$ where $F_\theta$ is a neural network with parameter $\theta$ that predicts $\mathbf{x}_0$. In a similar spirit, we adopt this pred-$\mathbf{x}$ parameterization and additionally derive a set of scaling functions for distribution translation, which we show is a strict superset.

**Score reparameterization.** Following the sampling distribution proposed in (8), a pred-$\mathbf{x}$ model can predict bridge score by

$$\nabla_{\mathbf{x}_t} \log q(\mathbf{x}_t \mid \mathbf{x}_T) \approx -\frac{\mathbf{x}_t - \left(\frac{\text{SNR}_T}{\text{SNR}_t}\frac{\alpha_t}{\alpha_T}\mathbf{x}_T + \alpha_t D_\theta(\mathbf{x}_t, \mathbf{x}_T, t)(1 - \frac{\text{SNR}_T}{\text{SNR}_t})\right)}{\sigma_t^2(1 - \frac{\text{SNR}_T}{\text{SNR}_t})} \tag{10}$$

**Scaling functions and loss weighting.** Following Karras et al. (2022), and let $a_t = \alpha_t/\alpha_T * \text{SNR}_T/\text{SNR}_t$, $b_t = \alpha_t(1 - \text{SNR}_T/\text{SNR}_t)$, $c_t = \sigma_t^2(1 - \text{SNR}_T/\text{SNR}_t)$, the scaling functions and weighting function $w(t)$ can be derived to be

$$c_{\text{in}}(t) = \frac{1}{\sqrt{a_t^2\sigma_T^2 + b_t^2\sigma_0^2 + 2a_tb_t\sigma_{0T} + c_t)}}, \quad c_{\text{out}}(t) = \sqrt{a_t^2(\sigma_T^2\sigma_0^2 - \sigma_{0T}^2) + \sigma_0^2c_t} * c_{\text{in}}(t) \tag{11}$$

$$c_{\text{skip}}(t) = \left(b_t\sigma_0^2 + a_t\sigma_{0T}\right) * c_{\text{in}}^2(t), \quad w(t) = \frac{1}{c_{\text{out}}(t)^2}, \quad c_{\text{noise}}(t) = \frac{1}{4}\log(t) \tag{12}$$

where $\sigma_0^2$, $\sigma_T^2$, and $\sigma_{0T}$ denote the variance of $\mathbf{x}_0$, variance of $\mathbf{x}_T$, and the covariance of the two, respectively. The only additional hyperparameters compared to EDM are $\sigma_T$ and $\sigma_{0T}$, which characterize the distribution of $\mathbf{x}_T$ and its correlation with $\mathbf{x}_0$. One can notice that in the case of EDM, $\sigma_t = t$, $\sigma_T^2 = \sigma_0^2 + T^2$ because $\mathbf{x}_T = \mathbf{x}_0 + T\boldsymbol{\epsilon}$ for some Gaussian noise $\boldsymbol{\epsilon}$, $\sigma_{0T} = \sigma_0^2$, and $\text{SNR}_T/\text{SNR}_t = t^2/T^2$. One can show that the scaling functions then reduce to those in EDM. We leave details in Appendix A.5.

**Generalized time-reversal.** Due to the probability flow ODE's resemblance with classifier-guidance (Dhariwal and Nichol, 2021; Ho and Salimans, 2022), we can introduce an additional parameter $w$ to set the "strength" of drift adjustment as below.

$$d\mathbf{x}_t = \Big[\mathbf{f}(\mathbf{x}_t, t) - g^2(t)\Big(\frac{1}{2}\mathbf{s}(\mathbf{x}_t, t, y, T) - w\mathbf{h}(\mathbf{x}_t, t, y, T)\Big)\Big]dt, \quad \mathbf{x}_T = y \tag{13}$$

which allows for a strictly wider class of marginal density of $\mathbf{x}_t$ generated by the resulting probability flow ODE. We examine the effect of this parameter in our ablation studies.

## 5 STOCHASTIC SAMPLING FOR DENOISING DIFFUSION BRIDGES

Although the probability flow ODE allows for one to use fast integration techniques to accelerate the sampling process (Zhang and Chen, 2022; Song et al., 2020a; Karras et al., 2022), purely

following an ODE path is problematic because diffusion bridges have fixed starting points given as data $\mathbf{x}_T = \mathbf{y} \sim q_{\text{data}}(\mathbf{y})$, and following the probability flow ODE backward in time generates a deterministic "expected" path. This can result in "averaged" or blurry outputs given initial conditions. Thus, we are motivated to introduce noise into our sampling process to improve the sampling quality and diversity.

**Higher-order hybrid sampler.** Our sampler is built upon prior higher-order ODE sampler in (Karras et al., 2022), which discretizes the sampling steps into $t_N > t_{N-1} > \cdots > t_0$ with decreasing intervals (see Appendix A.6 for details). Inspired by the predictor-corrector sampler introduced by Song et al. (2020b), we additionally introduce a scheduled Euler-Maruyama step which follows the backward SDE in between higher-order ODE steps. This ensures that the marginal distribution at each step approximately stays the same. We introduce additional scaling hyperparameter $s$, which define a step ratio in between $t_{i-1}$ and $t_i$ such that the interval $[t_i - s(t_i - t_{i-1}), t_i]$ is used for Euler-Maruyama steps and $[t_{i-1}, t_i - s(t_i - t_{i-1})]$ is used for Heun steps, as described in Algorithm 1.

# 6 RELATED WORKS AND SPECIAL CASES

**Diffusion models.** Diffusion models' advancements have boosted image generation, focusing on network design, noise schedules, samplers, and guidance methods. The advancements in diffusion models (Sohl-Dickstein et al., 2015; Ho et al., 2020; Song et al., 2020b) have improved state-of-the-art in image generation with improved network design (Song et al., 2020b; Karras et al., 2022; Nichol and Dhariwal, 2021; Hoogeboom et al., 2023; Peebles and Xie, 2023), noise-schedules (Nichol and Dhariwal, 2021; Karras et al., 2022; Peebles and Xie, 2023), better samplers (Song et al., 2020a; Lu et al., 2022a;b; Zhang and Chen, 2022), and guidance methods (Dhariwal and Nichol, 2021; Ho and Salimans, 2022). Following these successful design choices, we seek to construct our bridge formulation to allow for seamless integration with this literature. As such, we adopt a time-reversal perspective to directly extend these methods.

**Diffusion bridges, Schödinger bridges, and Doob's h-transform.** Diffusion bridges (Särkkä and Solin, 2019) have been actively studied in recent years in the context of generative modeling (Liu et al., 2022b; Somnath et al., 2023; De Bortoli et al., 2021; Peluchetti; 2023).Heng et al. (2021); Liu et al. (2022b) use Doob's h-transform to bridge between two points/distributions by simulating bridge paths. Other works (Somnath et al., 2023; Peluchetti; Delbracio and Milanfar, 2023) propose simulation-free alternatives for forward-time generation. Another approach De Bortoli et al. (2021) proposes Iterative Proportional Fitting (IPF) to tractably solve Schödinger Bridge (SB) problems in translating between different distributions. Liu et al. (2023) is built on a tractable class of SB with a simulation-free algorithm and demonstrates strong performance in image translation tasks. Extending SB with IPF, Bridge-Matching (Shi et al., 2023) proposes to use Iterative Markovian Fitting to solve the SB problem. A similar algorithm is also developed by Peluchetti (2023) for distribution translation.

**Flow and Optimal Transport** Works based on Flow-Matching (Lipman et al., 2023; Tong et al., 2023b; Pooladian et al., 2023; Tong et al., 2023a; Liu et al., 2022a) learn an ODE-based transport map to bridge two distributions. Lipman et al. (2023); Liu et al. (2022a) has demonstrated that by matching the velocity field of predefined transport maps, one can create powerful generative models competitive with the diffusion counterparts. Improving this approach, Tong et al. (2023b); Pooladian et al. (2023) exploit potential couplings between distributions using minibatch simulation-free OT. Stochastic interpolants (Albergo and Vanden-Eijnden, 2023; Albergo et al., 2023) build flow models and directly avoid the use of Doob's h-functions and provide an easy way to construct interpolation maps between distributions. Separate from these methods, our model uses a different denoising bridge score-matching loss and the construction allows integration with existing designs of diffusion models to push state-of-the-art further for image translation while retaining strong performance for unconditional generation.

## 6.1 SPECIAL CASES OF DENOISING DIFFUSION BRIDGE MODELS

**Case 1: Unconditional diffusion process (Song et al., 2020b).** We can show that the marginal $p(\mathbf{x}_t)$ when $p(\mathbf{x}_0) = q_{\text{data}}(\mathbf{x})$ exactly matches that of a regular diffusion process when $\mathbf{x}_T \sim q_{\text{data}}(\mathbf{y} \mid$

$\mathbf{x}) = \mathcal{N}(\alpha_T \mathbf{x}, \sigma_T^2 \boldsymbol{I})$. By taking expectation over $\mathbf{x}_T$ in Eq. (8), we have

$$p(\mathbf{x}_t \mid \mathbf{x}_0) = \mathcal{N}(\alpha_t \mathbf{x}_0, \sigma_t \boldsymbol{I}) \tag{14}$$

One can further show that during sampling, Eq. (6) and (7) reduce to those of a diffusion process when $\mathbf{x}_T$ is sampled from a Gaussian (see Appendix A.4).

**Case 2: OT-Flow-Matching (Lipman et al., 2023; Tong et al., 2023b) and Rectified Flow (Liu et al., 2022a).** These works learn to match deterministic dynamics defined through ODEs instead of SDEs. In this particular case, they work with "straight line" paths defined by $\mathbf{x}_T - \mathbf{x}_0$.

To see that our framework generalizes this, first let us define a family of diffusion bridges with variance scaled by $c \in (0, 1)$ such that $p(\mathbf{x}_t \mid \mathbf{x}_0, \mathbf{x}_T) = \mathcal{N}(\hat{\mu}_t, c^2 \hat{\sigma}_t^2 \boldsymbol{I})$ where $\hat{\mu}_t$ and $\hat{\sigma}_t$ are as defined in Eq. (8). One can therefore show that with a VE diffusion where $\sigma_t^2 = c^2 t$, given some fixed $\mathbf{x}_0$ and $\mathbf{x}_t$, *i.e.* $T = 1$, and $\mathbf{x}_t$ sampled from Eq. (8),

$$\lim_{c \to 0} \left[ \mathbf{f}(\mathbf{x}_t, t) - c^2 g^2(t) \left( \frac{1}{2} \nabla_{\mathbf{x}_t} \log p(\mathbf{x}_t \mid \mathbf{x}_0, \mathbf{x}_1) - \nabla_{\mathbf{x}_t} \log p(\mathbf{x}_1 \mid \mathbf{x}_t)) \right) \right] = \mathbf{x}_1 - \mathbf{x}_0 \tag{15}$$

where inside the bracket is the drift of probability flow ODE in Eq. (7) given $\mathbf{x}_0$ and $\mathbf{x}_1$, and the right hand side is exactly the straight line path term. In other words, these methods learn to match the drift in the bridge probability flow ODE (with a specific VE schedule) in the noiseless limit. The score model can then be matched against $\mathbf{x}_T - \mathbf{x}_0$, with some additional caveat to handle additional input $\mathbf{x}_T$, our framework exactly reduces to that of OT-Flow-Matching and Rectified Flow (details in Appendix A.4).

## 7 EXPERIMENTS

In this section we verify the generative capability of DDBM , and we want to answer the following questions: (1) How well does DDBM perform in image-to-image translation in pixel space? (2) Can DDBM perform well in unconditional generation when one side of the bridge reduces to Gaussian distribution? (3) How does the additional design choices introduced affect the final performance? Unless noted otherwise, we use the same VE diffusion schedule as in EDM for our bridge model by default. We leave further experiment details to Appendix B.

### 7.1 IMAGE-TO-IMAGE TRANSLATION

that DDBM can deliver competitive results in general image-to-image translation tasks. We evaluate on datasets with different image resolutions to demonstrate its applicability on a variety of scales. We choose Edges→Handbags (Isola et al., 2017) scaled to $64 \times 64$ pixels, which contains image pairs for translating from edge maps to colored handbags, and DIODE-Outdoor (Vasiljevic et al., 2019) scaled to $256 \times 256$, which contains normal maps and RGB images of real-world outdoor scenes. For evaluation metrics, we use Fréchet Inception Distance (FID) (Heusel et al., 2017) and Inception Scores (IS) (Barratt and Sharma, 2018) evaluated on all training samples translation quality, and we use LPIPS (Zhang et al., 2018) and MSE (in $[-1, 1]$ scale) to measure perceptual similarity and translation faithfulness.

We compare with Pix2Pix (Isola et al., 2017), SDEdit (Meng et al., 2022), DDIB (Su et al., 2022), Rectified Flow (Liu et al., 2022a), and I²SB (Liu et al., 2023) as they are built for image-to-image translation. For SDEdit we train unconditional EDM on the target domain, *e.g.* colored images, and initialize the translation by noising source image, *e.g.* sketches, and generate by EDM sampler given the noisy image. The other baseline methods are run with their respective repo while using the same network architecture as ours. Diffusion and transport-based methods are evaluated with the same number of function evaluations ($N = 40$, which is the default for EDM sampler for $64 \times 64$ images) to demonstrate our sampler's effectiveness in the regime when the number of sampling steps are low. Results are shown in Table 2 and additional settings are specified in Appendix B.

We observe that our model can perform translation with both high generation quality and faithfulness, and we find that VP bridges outperform VE bridges in some cases. In contrast, Rectified-Flow as an OT-based method struggles to perform well when the two domains share little low-level similarities (*e.g.* color, hue). DDIB also fails to produce coherent translation due to the wide differences in pixel-space distribution between the paired data. I²SB comes closest to our method, but falls short when

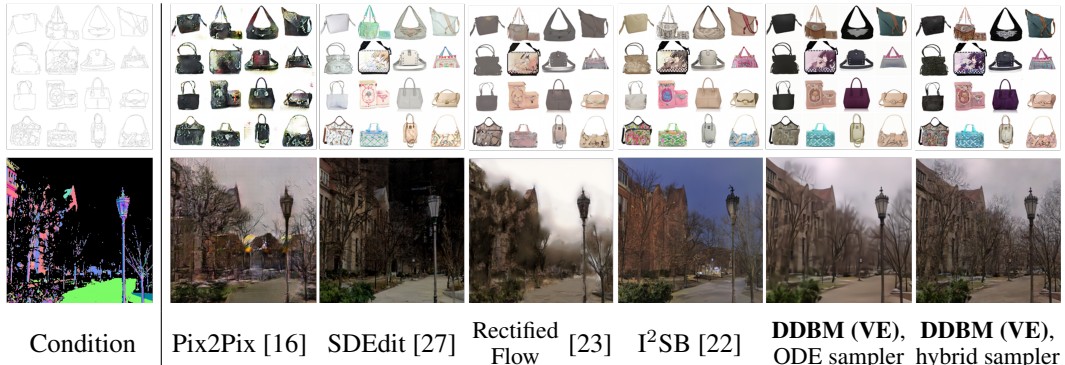

| | Condition | Pix2Pix [16] | SDEdit [27] | Rectified Flow [23] | I²SB [22] | **DDBM (VE),** ODE sampler | **DDBM (VE),** hybrid sampler |

Figure 3: Qualitative comparison with the most relevant baselines.

| | Edges→Handbags-64×64 | | | | DIODE-256×256 | | | |
|---|---|---|---|---|---|---|---|---|
| | FID ↓ | IS ↑ | LPIPS ↓ | MSE ↓ | FID ↓ | IS ↑ | LPIPS ↓ | MSE ↓ |
| Pix2Pix (Isola et al., 2017) | 74.8 | 4.24 | 0.356 | 0.209 | 82.4 | 4.22 | 0.556 | 0.133 |
| DDIB (Su et al., 2022) | 186.84 | 2.04 | 0.869 | 1.05 | 242.3 | 4.22 | 0.798 | 0.794 |
| SDEdit (Meng et al., 2022) | 26.5 | **3.58** | 0.271 | 0.510 | 31.14 | 5.70 | 0.714 | 0.534 |
| Rectified Flow (Liu et al., 2022a) | 25.3 | 2.80 | 0.241 | 0.088 | 77.18 | 5.87 | 0.534 | 0.157 |
| I²SB (Liu et al., 2023) | 7.43 | 3.40 | 0.244 | 0.191 | 9.34 | 5.77 | 0.373 | 0.145 |
| **DDBM (VE)** | 2.93 | 3.58 | **0.131** | **0.013** | 8.51 | 6.03 | **0.226** | **0.0107** |
| **DDBM (VP)** | **1.83** | **3.73** | 0.142 | 0.0402 | **4.43** | **6.21** | 0.244 | 0.0839 |

Table 2: Quantitative evaluation of pixel-space image-to-image translation.

limited by computational constraints, *i.e.* NFE is low. We additionally show qualitative comparison with the most performant baselines in Figure 3. More visual results can be found in Appendix **??**.

We demonstrate

## 7.2 ABLATION STUDIES

We now study the effect of our preconditioning and hybrid samplers on generation quality in the context of both VE and VP bridge (see Appendix B for VP bridge parameterization). In the left column of Figure 4, we fix the guidance scale $w$ at 1 and vary the Euler step size $s$ from 0 to 0.9 to introduce stochasticity. We see a significant decrease in FID score as we increase $s$ which produces the best performance at some value between 0 and 1 (*e.g.* $s = 0.3$ for Edges→Handbags). Figure 3 also shows that the ODE sampler (*i.e.* $s = 0$) produces blurry images while our hybrid sampler produces considerably sharper results. On the right column, we study the effect of $w$ (from 0 to 1) with fixed $s$. We observe that VE bridges are not affected by the change in $w$ whereas VP bridges heavily rely on setting $w = 1$. We hypothesize that this is due to the fact that VP bridges follow "curved paths" and destroy signals in between, so it is reliant on Doob's $h$-function for further guidance towards correct probability distribution.

We also study the effect of our preconditioning in Table 3. Our baseline without our preconditioning and our sampler is a simple model that directly matches output of the neural network to the training target and generates using EDM (Karras et al., 2022) sampler. We see that each introduced component further boosts the generation performance. Therefore, we can conclude that the introduced practical components are essential for the success of our DDBM .

## 7.3 UNCONDITIONAL GENERATION

When one side of the distribution becomes Gaussian distribution, our framework exactly reduces to that of diffusion models. Specifically, during training when the end point $\mathbf{x}_T \sim \mathcal{N}(\alpha_T \mathbf{x}_0, \sigma_T^2 \boldsymbol{I})$, our intermediate bridge samples $\mathbf{x}_t$ follows the distribution $\mathbf{x}_t \sim \mathcal{N}(\alpha_t \mathbf{x}_0, \sigma_t^2 \boldsymbol{I})$. We empirically verify that using our bridge sampling and the pred-$\mathbf{x}$ objective inspired by EDM, we can recover its performance by using our more generalized parameterization.

(a) Edges→Handbags

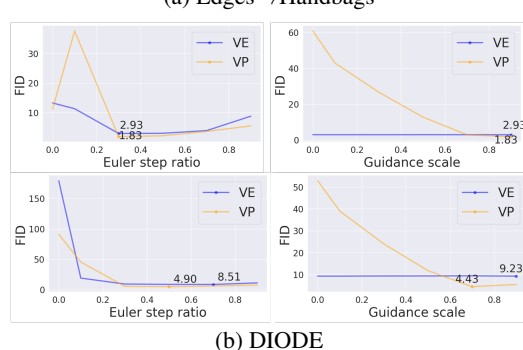

(b) DIODE

Figure 4: Ablation studies on Euler step ratio $s$ and guidance scale $w$: $w = 1$ for all ablation on $s$ and $s$ is set to the best-performing value for each dataset for ablation on $w$.

| Our precond. | Our sampler | E→H-64×64 | | DIODE-256×256 | |
|---|---|---|---|---|---|
| | | VE | VP | VE | VP |
| ✗ | ✗ | 14.02 | 11.76 | 126.3 | 96.93 |
| ✓ | ✗ | 13.26 | 11.19 | 79.25 | 91.07 |
| ✗ | ✓ | 13.11 | 29.91 | 91.31 | 21.92 |
| ✓ | ✓ | **2.93** | **1.83** | **8.51** | **4.43** |

Table 3: Ablation study on the effect of sampler and preconditioning on FID. Cross mark on our preconditioning means no output reparameterization and directly use network output to match training target. Cross mark on our sampler means we reuse the ODE sampler from EDM with the same setting. E→H is a short-hand for Edges→Handbags.

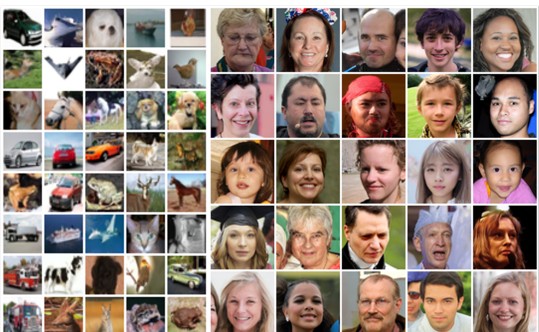

Figure 5: Generation on CIFAR-10 and FFHQ-64× 64.

| | CIFAR-10 | | FFHQ-64 × 64 | |
|---|---|---|---|---|
| | NFE ↓ | FID ↓ | NFE ↓ | FID ↓ |
| DDPM [13] | 1000 | 3.17 | 1000 | 3.52 |
| DDIM [42] | 50 | 4.67 | 50 | 5.18 |
| DDPM++ [44] | 1000 | 3.01 | 1000 | 3.39 |
| NCSN++ [44] | 1000 | 3.77 | 1000 | 25.95 |
| Rectified Flow [23] | 127 | 2.58 | 152 | 4.45 |
| EDM [18] | 35 | **2.04** | 79 | 2.53 |
| **DDBM** | 35 | 2.06 | 79 | **2.44** |

Table 4: Evaluation of unconditional generation.

We evaluate our method on CIFAR-10 (Krizhevsky et al., 2009) and FFHQ-$64 \times 64$ (Karras et al., 2019) which are processed according to Karras et al. (2022). We use FID score for quantitative evaluation using 50K generated images and use number of function evaluations (NFE) for generation efficiency. We compare our generation results against diffusion-based and optimal transport-based models including DDPM (Ho et al., 2020), DDIM (Song et al., 2020a), DDPM++ (Song et al., 2020b), NCSN++ (Song et al., 2020b), Rectified Flow (Liu et al., 2022a), EDM (Karras et al., 2022). Quantitative results are presented in Table 4 and generated samples are shown in Figure 5.

We observe that our model is able to match EDM performance with negligible degradation in FID scores for CIFAR-10 and marginal improvement for FFHQ-$64 \times 64$. This corroborates our claim that our method can benefit from advances in diffusion models and generalize many of the advanced parameterization techniques such as those introduced in EDM.

## 8 CONCLUSION

In this work, we introduce Denoising Diffusion Bridge Models, a novel class of models that builds a stochastic bridge between paired samples with tractable marginal distributions in between. The model is learned by matching the conditional score of a tractable bridge distribution, which allows one to transport from one distribution to another via a new reverse SDE or probability flow ODE. Additionally, this generalized framework shares many similarities with diffusion models, thus allowing us to reuse and generalize many designs of diffusion models. We believe that DDBM is a significant contribution towards a general framework for distribution translation. In the era of generative AI, DDBM has a further role to play.

## 9 ACKNOWLEDGEMENT

This research is supported by HAI, NSF(#1651565), ARO (W911NF-21-1-0125), ONR (N00014-23-1-2159), CZ Biohub. We additionally thank support from and helpful discussions with friends and colleagues at Stanford University.

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
