# Appendix: Denoising Diffusion Bridge Models

## A PROOFS

### A.1 MARGINAL DISTRIBUTION

We note that for tractable transition kernels specified in Table 1, we can derive the marginal distribution of $\mathbf{x}_t$ using Bayes' rule

$$p(\mathbf{x}_t \mid \mathbf{x}_0, \mathbf{x}_T) = \frac{p(\mathbf{x}_T \mid \mathbf{x}_t)p(\mathbf{x}_t \mid \mathbf{x}_0)}{p(\mathbf{x}_T \mid \mathbf{x}_0)}$$

We can directly derive this by looking at the resulting density function. First,

$$p(\mathbf{x}_t \mid \mathbf{x}_0) = \frac{1}{\sqrt{2\pi}\sigma_t} \exp\left(-\frac{(\mathbf{x}_t - \alpha_t \mathbf{x}_0)^2}{2\sigma_t^2}\right) \tag{16}$$

$$p(\mathbf{x}_T \mid \mathbf{x}_t) = \frac{1}{\sqrt{2\pi}\sqrt{\sigma_T^2 - \frac{\alpha_T^2}{\alpha_t^2}\sigma_t^2}} \exp\left(-\frac{(\frac{\alpha_T}{\alpha_t}\mathbf{x}_t - \mathbf{x}_T)^2}{2(\sigma_T^2 - \frac{\alpha_T^2}{\alpha_t^2}\sigma_t^2)}\right) \tag{17}$$

$$= \frac{1}{\sqrt{2\pi}\sqrt{\sigma_T^2 - \frac{\alpha_T^2}{\alpha_t^2}\sigma_t^2}} \exp\left(-\frac{(\mathbf{x}_t - \frac{\alpha_t}{\alpha_T}\mathbf{x}_T)^2}{2\sigma_t^2(\frac{\text{SNR}_t}{\text{SNR}_T} - 1)}\right) \tag{18}$$

$$p(\mathbf{x}_T \mid \mathbf{x}_0) = \frac{1}{\sqrt{2\pi}\sigma_T} \exp\left(-\frac{(\mathbf{x}_T - \alpha_T \mathbf{x}_0)^2}{2\sigma_T^2}\right) \tag{19}$$

and we refer readers to Kingma et al. (2021) for details on $p(\mathbf{x}_s \mid \mathbf{x}_t)$ for any $s > t$. Then we know

$$p(\mathbf{x}_t \mid \mathbf{x}_0, \mathbf{x}_T) \tag{20}$$

$$= \frac{1}{\sqrt{2\pi} \underbrace{\frac{\sigma_t}{\sigma_T}\sqrt{\sigma_T^2 - \frac{\alpha_T^2}{\alpha_t^2}\sigma_t^2}}_{\hat{\sigma}_t}} \exp\left(-\frac{1}{2}\underbrace{\left[\frac{(\mathbf{x}_t - \alpha_t\mathbf{x}_0)^2}{\sigma_t^2} + \frac{(\mathbf{x}_t - \frac{\alpha_t}{\alpha_T}\mathbf{x}_T)^2}{\sigma_t^2(\frac{\text{SNR}_t}{\text{SNR}_T} - 1)} - \frac{(\mathbf{x}_T - \alpha_T\mathbf{x}_0)^2}{\sigma_T^2}\right]}_{-\frac{(\mathbf{x}_t - \hat{\mu}_t)^2}{2\hat{\sigma}_t^2}}\right) \tag{21}$$

where

$$\hat{\sigma}_t^2 = \sigma_t^2\left(1 - \frac{\text{SNR}_T}{\text{SNR}_t}\right) \tag{22}$$

$$\hat{\mu}_t = \frac{\text{SNR}_T}{\text{SNR}_t}\frac{\alpha_t}{\alpha_T}\mathbf{x}_T + \alpha_t\mathbf{x}_0\left(1 - \frac{\text{SNR}_T}{\text{SNR}_t}\right) \tag{23}$$

### A.2 DENOISING BRIDGE SCORE MATCHING

**Theorem 2** (Denoising Bridge Score Matching). *Let $(\mathbf{x}_0, \mathbf{x}_T) \sim q_{\text{data}}(\mathbf{x}, \mathbf{y})$, $\mathbf{x}_t \sim q(\mathbf{x}_t \mid \mathbf{x}_0, \mathbf{x}_T)$, $t \sim p(t)$ for any non-zero time sampling distribution $p(t)$ in $[0, T]$, and $w(t)$ be a non-zero loss weighting term of any choice. Minimum of the following objective:*

$$\mathcal{L}(\theta) = \mathbb{E}_{\mathbf{x}_t, \mathbf{x}_0, \mathbf{x}_T, t}\left[w(t)\|\mathbf{s}_\theta(\mathbf{x}_t, \mathbf{x}_T, t) - \nabla_{\mathbf{x}_t}\log q(\mathbf{x}_t \mid \mathbf{x}_0, \mathbf{x}_T)\|^2\right] \tag{9}$$

*satisfies $\mathbf{s}_\theta(\mathbf{x}_t, \mathbf{x}_T, t) = \nabla_{\mathbf{x}_t}\log q(\mathbf{x}_t \mid \mathbf{x}_T)$.*

*Proof.* We can explicitly write the objective as

$$\int_{\mathbf{x}_t, \mathbf{x}_0, \mathbf{x}_T, t} q(\mathbf{x}_t \mid \mathbf{x}_0, \mathbf{x}_T)q_{\text{data}}(\mathbf{x}_0, \mathbf{x}_T)w(t)p(t)\left[\|\mathbf{s}_\theta(\mathbf{x}_t, \mathbf{x}_T, t) - \nabla_{\mathbf{x}_t}\log q(\mathbf{x}_t \mid \mathbf{x}_0, \mathbf{x}_T)\|^2\right]d\mathbf{x}_t d\mathbf{x}_0 d\mathbf{x}_T dt \tag{24}$$

Since the objective is an $\mathcal{L}_2$ loss and $p(t), w(t)$ are non-zero, its minimum can be derived as

$$\mathbf{s}^*(\mathbf{x}_t, \mathbf{x}_T, t)$$

$$= \int_{\mathbf{x}_0, t} \frac{q(\mathbf{x}_t \mid \mathbf{x}_0, \mathbf{x}_T) q_{\text{data}}(\mathbf{x}_0, \mathbf{x}_T) \cancel{w(t)p(t)}}{\int_{\mathbf{x}_0} p(\mathbf{x}_t \mid \mathbf{x}_0, \mathbf{x}_T) q_{\text{data}}(\mathbf{x}_0, \mathbf{x}_T) \cancel{w(t)p(t)} d\mathbf{x}_0} \nabla_d \log q(\mathbf{x}_t \mid \mathbf{x}_0, \mathbf{x}_T) \mathbf{x}_0 dt \tag{25}$$

$$= \int_{\mathbf{x}_0} \frac{q(\mathbf{x}_t \mid \mathbf{x}_0, \mathbf{x}_T) q_{\text{data}}(\mathbf{x}_0, \mathbf{x}_T)}{q(\mathbf{x}_t, \mathbf{x}_T)} \nabla_{\mathbf{x}_t} \log q(\mathbf{x}_t \mid \mathbf{x}_0, \mathbf{x}_T) d\mathbf{x}_0 \tag{26}$$

$$= \int_{\mathbf{x}_0} \frac{\cancel{q(\mathbf{x}_t \mid \mathbf{x}_0, \mathbf{x}_T)} q_{\text{data}}(\mathbf{x}_0, \mathbf{x}_T)}{q(\mathbf{x}_t, \mathbf{x}_T)} \frac{\nabla_{\mathbf{x}_t} q(\mathbf{x}_t \mid \mathbf{x}_0, \mathbf{x}_T)}{\cancel{q(\mathbf{x}_t \mid \mathbf{x}_0, \mathbf{x}_T)}} d\mathbf{x}_0 \tag{27}$$

$$= \frac{\nabla_{\mathbf{x}_t} \int_{\mathbf{x}_0} q_{\text{data}}(\mathbf{x}_0, \mathbf{x}_T) q(\mathbf{x}_t \mid \mathbf{x}_0, \mathbf{x}_T) d\mathbf{x}_0}{q(\mathbf{x}_t, \mathbf{x}_T)} \tag{28}$$

$$= \frac{\nabla_{\mathbf{x}_t} q(\mathbf{x}_t, \mathbf{x}_T)}{q(\mathbf{x}_t, \mathbf{x}_T)} \tag{29}$$

$$= \nabla_{\mathbf{x}_t} \log q(\mathbf{x}_t \mid \mathbf{x}_T) \tag{30}$$

Thus, minimizing the objective approximates the conditional score. □

## A.3 PROBABILITY FLOW ODE OF DIFFUSION BRIDGES

**Theorem 1.** *The evolution of conditional probability $q(\mathbf{x}_t \mid \mathbf{x}_T)$ has a time-reversed SDE of the form*

$$d\mathbf{x}_t = \left[ \mathbf{f}(\mathbf{x}_t, t) - g^2(t)\Big(\mathbf{s}(\mathbf{x}_t, t, y, T) - \mathbf{h}(\mathbf{x}_t, t, y, T)\Big) \right] dt + g(t) d\hat{\mathbf{w}}_t, \quad \mathbf{x}_T = y \tag{6}$$

*with an associated probability flow ODE*

$$d\mathbf{x}_t = \left[ \mathbf{f}(\mathbf{x}_t, t) - g^2(t)\Big(\frac{1}{2}\mathbf{s}(\mathbf{x}_t, t, y, T) - \mathbf{h}(\mathbf{x}_t, t, y, T)\Big) \right] dt, \quad \mathbf{x}_T = y \tag{7}$$

*on $t \leq T - \epsilon$ for any $\epsilon > 0$, where $\hat{\mathbf{w}}_t$ denotes a Wiener process, $\mathbf{s}(x, t, y, T) = \nabla_{\mathbf{x}_t} \log q(\mathbf{x}_t \mid \mathbf{x}_T)\big|_{\mathbf{x}_t = x, \mathbf{x}_T = y}$ and $\mathbf{h}$ is as defined in Eq. (5).*

*Proof.* To find the time evolution of $q(\mathbf{x}_t \mid \mathbf{x}_T) = \int_{\mathbf{x}_0} p(\mathbf{x}_t \mid \mathbf{x}_0, \mathbf{x}_T) q_{\text{data}}(\mathbf{x}_0 \mid \mathbf{x}_T)$, we can first find the time evolution of $p(\mathbf{x}_t \mid \mathbf{x}_0 = x_0, \mathbf{x}_T = x_T)$ for fixed endpoints $x_0$ and $x_T$, which by Bayes' rule is

$$p(\mathbf{x}_t \mid \mathbf{x}_T = x_T, \mathbf{x}_0 = x_0) = \frac{p(\mathbf{x}_T = x_T \mid \mathbf{x}_t) p(\mathbf{x}_t \mid \mathbf{x}_0 = x_0)}{p(\mathbf{x}_T = x_T \mid \mathbf{x}_0 = x_0)}$$

where $p(\mathbf{x}_t \mid \mathbf{x}_0)$ follows Kolmogorov forward equation

$$\frac{\partial}{\partial t} p(\mathbf{x}_t \mid \mathbf{x}_0 = x_0) = -\nabla_{\mathbf{x}_t} \cdot \left[ \mathbf{f}(\mathbf{x}_t, t) p(\mathbf{x}_t \mid \mathbf{x}_0 = x_0) \right] + \frac{1}{2} g^2(t) \nabla_{\mathbf{x}_t} \cdot \nabla_{\mathbf{x}_t} p(\mathbf{x}_t \mid \mathbf{x}_0 = x_0) \tag{31}$$

and $p(\mathbf{x}_T = x_T \mid \mathbf{x}_t)$ follows Kolmogorov backward equation Szavits-Nossan and Evans (2015) where

$$-\frac{\partial}{\partial t} p(\mathbf{x}_T = x_T \mid \mathbf{x}_t) = \mathbf{f}(\mathbf{x}_t, t) \cdot \nabla_{\mathbf{x}_t} p(\mathbf{x}_T = x_T \mid \mathbf{x}_t) + \frac{1}{2} g^2(t) \nabla_{\mathbf{x}_t} \cdot \nabla_{\mathbf{x}_t} p(\mathbf{x}_T = x_T \mid \mathbf{x}_t) \tag{32}$$

The time derivative of $p(\mathbf{x}_t \mid \mathbf{x}_T = x_T, \mathbf{x}_0 = x_0)$ thus follows

$$\frac{\partial}{\partial t} p(\mathbf{x}_t \mid \mathbf{x}_T = x_T, \mathbf{x}_0 = x_0) \tag{33}$$

$$= \frac{\partial}{\partial t} \frac{p(\mathbf{x}_T = x_T \mid \mathbf{x}_t) p(\mathbf{x}_t \mid \mathbf{x}_0 = x_0)}{p(\mathbf{x}_T = x_T \mid \mathbf{x}_0 = x_0)} \tag{34}$$

$$= \underbrace{\frac{p(\mathbf{x}_t \mid \mathbf{x}_0 = x_0)}{p(\mathbf{x}_T = x_T \mid \mathbf{x}_0 = x_0)} \frac{\partial}{\partial t} p(\mathbf{x}_T = x_T \mid \mathbf{x}_t)}_{①} + \underbrace{\frac{p(\mathbf{x}_T = x_T \mid \mathbf{x}_t)}{p(\mathbf{x}_T = x_T \mid \mathbf{x}_0 = x_0)} \frac{\partial}{\partial t} p(\mathbf{x}_t \mid \mathbf{x}_0 = x_0)}_{②} \tag{35}$$

Further expanding the right-hand-side, we have

$$\textcircled{1} = -\frac{p(\mathbf{x}_t \mid \mathbf{x}_0 = x_0)}{p(\mathbf{x}_T = x_T \mid \mathbf{x}_0 = x_0)} \left( \mathbf{f}(\mathbf{x}_t, t) \cdot \nabla_{\mathbf{x}_t} p(\mathbf{x}_T = x_T \mid \mathbf{x}_t) + \frac{1}{2} g^2(t) \nabla_{\mathbf{x}_t} \cdot \nabla_{\mathbf{x}_t} p(\mathbf{x}_T = x_T \mid \mathbf{x}_t) \right)$$

$$\textcircled{2} = \frac{p(\mathbf{x}_T = x_T \mid \mathbf{x}_t)}{p(\mathbf{x}_T = x_T \mid \mathbf{x}_0 = x_0)} \left( -\nabla_{\mathbf{x}_t} \cdot \left[ \mathbf{f}(\mathbf{x}_t, t) p(\mathbf{x}_t \mid \mathbf{x}_0 = x_0) \right] + \frac{1}{2} g^2(t) \nabla_{\mathbf{x}_t} \cdot \nabla_{\mathbf{x}_t} p(\mathbf{x}_t \mid \mathbf{x}_0 = x_0) \right)$$

We can notice that the sum of the first terms of $\textcircled{1}$ and $\textcircled{2}$ is the result of a product rule, thus

$$\textcircled{1} + \textcircled{2} = -\nabla_{\mathbf{x}_t} \cdot \left[ \mathbf{f}(\mathbf{x}_t, t) p(\mathbf{x}_t \mid \mathbf{x}_T = x_T, \mathbf{x}_0 = x_0) \right]$$

$$+ \frac{1}{2} g^2(t) \left( \frac{p(\mathbf{x}_T = x_T \mid \mathbf{x}_t)}{p(\mathbf{x}_T = x_T \mid \mathbf{x}_0 = x_0)} \nabla_{\mathbf{x}_t} \cdot \nabla_{\mathbf{x}_t} p(\mathbf{x}_t \mid \mathbf{x}_0 = x_0) \right. \tag{36}$$

$$\left. - \frac{p(\mathbf{x}_t \mid \mathbf{x}_0 = x_0)}{p(\mathbf{x}_T = x_T \mid \mathbf{x}_0 = x_0)} \nabla_{\mathbf{x}_t} \cdot \nabla_{\mathbf{x}_t} p(\mathbf{x}_T = x_T \mid \mathbf{x}_t) \right)$$

We now focus on reducing the terms in the last bracket. For clarity, we similarly number the two terms inside the bracket such that

$$\textcircled{3} = \frac{p(\mathbf{x}_T = x_T \mid \mathbf{x}_t)}{p(\mathbf{x}_T = x_T \mid \mathbf{x}_0 = x_0)} \nabla_{\mathbf{x}_t} \cdot \nabla_{\mathbf{x}_t} p(\mathbf{x}_t \mid \mathbf{x}_0 = x_0) \tag{37}$$

$$\textcircled{4} = \frac{p(\mathbf{x}_t \mid \mathbf{x}_0 = x_0)}{p(\mathbf{x}_T = x_T \mid \mathbf{x}_0 = x_0)} \nabla_{\mathbf{x}_t} \cdot \nabla_{\mathbf{x}_t} p(\mathbf{x}_T = x_T \mid \mathbf{x}_t) \tag{38}$$

Now we can complete these terms to be results of product rule by adding and subtracting the following term

$$\frac{\nabla_{\mathbf{x}_t} p(\mathbf{x}_T = x_T \mid \mathbf{x}_t) \cdot \nabla_{\mathbf{x}_t} p(\mathbf{x}_t \mid \mathbf{x}_0 = x_0)}{p(\mathbf{x}_T = x_T \mid \mathbf{x}_0 = x_0)} \tag{39}$$

$$= \underbrace{\frac{\nabla_{\mathbf{x}_t} p(\mathbf{x}_t \mid \mathbf{x}_0 = x_0)}{p(\mathbf{x}_T = x_T \mid \mathbf{x}_0 = x_0)} \cdot \left[ p(\mathbf{x}_T = x_T \mid \mathbf{x}_t) \nabla_{\mathbf{x}_t} \log p(\mathbf{x}_T = x_T \mid \mathbf{x}_t) \right]}_{\textcircled{5}} \tag{40}$$

$$= \underbrace{\frac{\nabla_{\mathbf{x}_t} p(\mathbf{x}_T = x_T \mid \mathbf{x}_t)}{p(\mathbf{x}_T = x_T \mid \mathbf{x}_0 = x_0)} \cdot \left[ p(\mathbf{x}_t \mid \mathbf{x}_0 = x_0) \nabla_{\mathbf{x}_t} \log p(\mathbf{x}_t \mid \mathbf{x}_0 = x_0) \right]}_{\textcircled{6}} \tag{41}$$

which takes 2 equivalent forms $\textcircled{5}$ and $\textcircled{6}$. Now we can write Eq. (36) as

$$\textcircled{1} + \textcircled{2} = -\nabla_{\mathbf{x}_t} \cdot \left[ \mathbf{f}(\mathbf{x}_t, t) p(\mathbf{x}_t \mid \mathbf{x}_T = x_T, \mathbf{x}_0 = x_0) \right] \tag{42}$$

$$+ \frac{1}{2} g^2(t) \left( \textcircled{3} + \textcircled{4} + \textcircled{5} + \textcircled{6} \right) - g^2(t) \left( \textcircled{4} + \textcircled{5} \right) \tag{43}$$

We can notice that

$$\textcircled{3} + \textcircled{6} = \nabla_{\mathbf{x}_t} \cdot \left( p(\mathbf{x}_t \mid \mathbf{x}_T = x_T, \mathbf{x}_0 = x_0) \nabla_{\mathbf{x}_t} \log p(\mathbf{x}_t \mid \mathbf{x}_0 = x_0) \right) \tag{44}$$

$$\textcircled{4} + \textcircled{5} = \nabla_{\mathbf{x}_t} \cdot \left( p(\mathbf{x}_t \mid \mathbf{x}_T = x_T, \mathbf{x}_0 = x_0) \nabla_{\mathbf{x}_t} \log p(\mathbf{x}_T = x_T \mid \mathbf{x}_t) \right) \tag{45}$$

and using Bayes' rule,

$$\nabla_{\mathbf{x}_t} \log p(\mathbf{x}_t \mid \mathbf{x}_T = x_T, \mathbf{x}_0 = x_0) = \nabla_{\mathbf{x}_t} \log p(\mathbf{x}_T = x_T \mid \mathbf{x}_t) + \nabla_{\mathbf{x}_t} \log p(\mathbf{x}_t \mid \mathbf{x}_0 = x_0) \tag{46}$$

we have

$$\text{③} + \text{④} + \text{⑤} + \text{⑥} = \nabla_{\mathbf{x}_t} \cdot \left( p(\mathbf{x}_t \mid \mathbf{x}_T = x_T, \mathbf{x}_0 = x_0) \nabla_{\mathbf{x}_t} \log p(\mathbf{x}_t \mid \mathbf{x}_T = x_T, \mathbf{x}_0 = x_0) \right) \tag{47}$$

$$= \nabla_{\mathbf{x}_t} \cdot \nabla_{\mathbf{x}_t} p(\mathbf{x}_t \mid \mathbf{x}_T = x_T, \mathbf{x}_0 = x_0) \tag{48}$$

Therefore,

$$\frac{\partial}{\partial t} p(\mathbf{x}_t \mid \mathbf{x}_T = x_T, \mathbf{x}_0 = x_0)$$

$$= -\nabla_{\mathbf{x}_t} \cdot \left[ \left( \mathbf{f}(\mathbf{x}_t, t) + g^2(t) \nabla_{\mathbf{x}_t} \log p(\mathbf{x}_T = x_T \mid \mathbf{x}_t) \right) p(\mathbf{x}_t \mid \mathbf{x}_T = x_T, \mathbf{x}_0 = x_0) \right] \tag{49}$$

$$+ \frac{1}{2} g^2(t) \nabla_{\mathbf{x}_t} \cdot \nabla_{\mathbf{x}_t} p(\mathbf{x}_t \mid \mathbf{x}_T = x_T, \mathbf{x}_0 = x_0)$$

which is a Fokker-Planck equation for a (forward) SDE with the modified drift term

$$\mathbf{f}(\mathbf{x}_t, t) + g^2(t) \nabla_{\mathbf{x}_t} \log p(\mathbf{x}_T = x_T \mid \mathbf{x}_t)$$

To find the time derivative for $q(\mathbf{x}_t \mid \mathbf{x}_T) = \int_{\mathbf{x}_0} p(\mathbf{x}_t \mid \mathbf{x}_0, \mathbf{x}_T) q_{\text{data}}(\mathbf{x}_0 \mid \mathbf{x}_T)$, we can simply marginalize out $\mathbf{x}_0$ with distribution $q_{\text{data}}(\mathbf{x}_0 \mid \mathbf{x}_T)$ in the resulting Fokker-Planck, which can be achieved due to linearity of expectation with respect to $\mathbf{x}_0$. That is,

$$\mathbb{E}_{\mathbf{x}_0 \sim q_{\text{data}}(\mathbf{x}_0 \mid \mathbf{x}_T = x_T)} \left[ \frac{\partial}{\partial t} p(\mathbf{x}_t \mid \mathbf{x}_T = x_T, \mathbf{x}_0) \right]$$

$$= -\nabla_{\mathbf{x}_t} \cdot \left[ \left( \mathbf{f}(\mathbf{x}_t, t) + g^2(t) \nabla_{\mathbf{x}_t} \log p(\mathbf{x}_T = x_T \mid \mathbf{x}_t) \right) \mathbb{E}_{\mathbf{x}_0 \sim q_{\text{data}}(\mathbf{x}_0 \mid \mathbf{x}_T = x_T)} \left[ p(\mathbf{x}_t \mid \mathbf{x}_T = x_T, \mathbf{x}_0) \right] \right]$$

$$+ \frac{1}{2} g^2(t) \nabla_{\mathbf{x}_t} \cdot \nabla_{\mathbf{x}_t} \mathbb{E}_{\mathbf{x}_0 \sim q_{\text{data}}(\mathbf{x}_0 \mid \mathbf{x}_T = x_T)} \left[ p(\mathbf{x}_t \mid \mathbf{x}_T = x_T, \mathbf{x}_0) \right] \tag{50}$$

Since for $t \in [0, T - c]$ for some $c > 0$, Doob's h-function is well-defined, and $p(x_t \mid x_0, x_T)$ is smooth, and we can take the expectation inside the equations. Additionally, the drift adjustment $\nabla_{\mathbf{x}_t} \log p(\mathbf{x}_T = x_T \mid \mathbf{x}_t)$ does not depend on $\mathbf{x}_0$ the expectation is simply over $p(\mathbf{x}_t \mid \mathbf{x}_T = x_T, \mathbf{x}_0)$ expectation and by definition LHS is $q(\mathbf{x}_t \mid \mathbf{x}_T = x_T)$,

$$\frac{\partial}{\partial t} q(\mathbf{x}_t \mid \mathbf{x}_T = x_T) = -\nabla_{\mathbf{x}_t} \cdot \left[ \left( \mathbf{f}(\mathbf{x}_t, t) + g^2(t) \nabla_{\mathbf{x}_t} \log p(\mathbf{x}_T = x_T \mid \mathbf{x}_t) \right) q(\mathbf{x}_t \mid \mathbf{x}_T = x_T) \right]$$

$$+ \frac{1}{2} g^2(t) \nabla_{\mathbf{x}_t} \cdot \nabla_{\mathbf{x}_t} q(\mathbf{x}_t \mid \mathbf{x}_T = x_T) \tag{51}$$

This characterizes a reverse SDE specified in Theorem 1.

We can further use conversion trick in Song et al. (2020b) to convert this into a continuity equation without any diffusion term where

$$\frac{\partial}{\partial t} q(\mathbf{x}_t \mid \mathbf{x}_T = x_T) = \nabla_{\mathbf{x}_t} \cdot \left[ \tilde{\mathbf{f}}(\mathbf{x}_t, t) q(\mathbf{x}_t \mid \mathbf{x}_T = x_T) \right] \tag{52}$$

where

$$\tilde{\mathbf{f}}(\mathbf{x}_t, t) = \mathbf{f}(\mathbf{x}_t, t) + g^2(t) \nabla_{\mathbf{x}_t} \log p(\mathbf{x}_T = x_T \mid \mathbf{x}_t) - \frac{1}{2} g^2(t) \nabla_{\mathbf{x}_t} \log q(\mathbf{x}_t \mid \mathbf{x}_T = x_T) \tag{53}$$

$$= \mathbf{f}(\mathbf{x}_t, t) - g^2(t) \left( \frac{1}{2} \nabla_{\mathbf{x}_t} \log q(\mathbf{x}_t \mid \mathbf{x}_T = x_T) - \nabla_{\mathbf{x}_t} \log p(\mathbf{x}_T = x_T \mid \mathbf{x}_t) \right) \tag{54}$$

$$\square$$

## A.4 SPECIAL CASES OF DENOISING DIFFUSION BRIDGES

**Unconditional diffusion models.** We first give a general intuition that the marginal distribution of $\mathbf{x}_t$ sampling from the bridge is the same as sampling marginally from $p(\mathbf{x}_t \mid \mathbf{x}_0)$ for a diffusion transition kernel $p(\cdot)$. We can see this by observing

$$\mathbf{x}_t = \frac{\text{SNR}_T}{\text{SNR}_t} \frac{\alpha_t}{\alpha_T} \mathbf{x}_T + \alpha_t \mathbf{x}_0 (1 - \frac{\text{SNR}_T}{\text{SNR}_t}) + \sigma_t^2 (1 - \frac{\text{SNR}_T}{\text{SNR}_t}) \boldsymbol{\epsilon}_1 \tag{55}$$

where $\boldsymbol{\epsilon}_1 \sim \mathcal{N}(\mathbf{0}, \boldsymbol{I})$. And since we assume $\mathbf{x}_T \sim \mathcal{N}(\alpha_T \mathbf{x}_0, \sigma_T^2 \boldsymbol{I})$, we rewrite the above equation as

$$\mathbf{x}_t = \frac{\text{SNR}_T}{\text{SNR}_t} \frac{\alpha_t}{\alpha_T} (\alpha_T \mathbf{x}_0 + \sigma_T \boldsymbol{\epsilon}_2) + \alpha_t \mathbf{x}_0 (1 - \frac{\text{SNR}_T}{\text{SNR}_t}) + \sigma_t \sqrt{(1 - \frac{\text{SNR}_T}{\text{SNR}_t})} \boldsymbol{\epsilon}_1 \tag{56}$$

$$= \frac{\text{SNR}_T}{\text{SNR}_t} \alpha_t \mathbf{x}_0 + \frac{\text{SNR}_T}{\text{SNR}_t} \frac{\alpha_t}{\alpha_T} \sigma_T \boldsymbol{\epsilon}_2 + \alpha_t \mathbf{x}_0 (1 - \frac{\text{SNR}_T}{\text{SNR}_t}) + \sigma_t \sqrt{(1 - \frac{\text{SNR}_T}{\text{SNR}_t})} \boldsymbol{\epsilon}_1 \tag{57}$$

$$= \alpha_t \mathbf{x}_0 + \sigma_t \boldsymbol{\epsilon} \tag{58}$$

where $\boldsymbol{\epsilon} \sim \mathcal{N}(\mathbf{0}, \boldsymbol{I})$ and the last equality is due to the fact that the addition of two Gaussian with variances $\sigma_1^2, \sigma_2^2$ is another Gaussian with variance $\sigma_1^2 + \sigma_2^2$.

Formally, to show that it is a special case, we first observe that the score matching objective allows our network to approximate $\nabla_{\mathbf{x}_t} \log p(\mathbf{x}_t \mid \mathbf{x}_T)$ which is the conditional score of the diffusion transition kernel. Then we will show that the Fokker-Planck equation reduces to that of a diffusion when marginalizing out dependency on $\mathbf{x}_T$.

From proof of Theorem 1, we know that the Fokker-Planck equation for $p(\mathbf{x}_t \mid \mathbf{x}_T)$ follows

$$\frac{\partial}{\partial t} p(\mathbf{x}_t \mid \mathbf{x}_T = x_T) = - \nabla_{\mathbf{x}_t} \cdot \left[ \left( \mathbf{f}(\mathbf{x}_t, t) + g^2(t) \nabla_{\mathbf{x}_t} \log p(\mathbf{x}_T = x_T \mid \mathbf{x}_t) \right) p(\mathbf{x}_t \mid \mathbf{x}_T = x_T) \right] \tag{59}$$

$$+ \frac{1}{2} g^2(t) \nabla_{\mathbf{x}_t} \cdot \nabla_{\mathbf{x}_t} p(\mathbf{x}_t \mid \mathbf{x}_T = x_T) \tag{60}$$

Here we note that we use $p(\mathbf{x}_t \mid \mathbf{x}_T)$ because we are considering a diffusion process as a special case of a general $q(\mathbf{x}_t \mid \mathbf{x}_T)$ introduced in Theorem 2. We can marginalize out $\mathbf{x}_T$ such that

$$\frac{\partial}{\partial t} \mathbb{E}_{\mathbf{x}_T \sim p(\mathbf{x}_T)} \left[ p(\mathbf{x}_t \mid \mathbf{x}_T) \right]$$

$$= \mathbb{E}_{\mathbf{x}_T \sim p(\mathbf{x}_T)} \left[ - \nabla_{\mathbf{x}_t} \cdot \left[ \left( \mathbf{f}(\mathbf{x}_t, t) + g^2(t) \nabla_{\mathbf{x}_t} \log p(\mathbf{x}_T \mid \mathbf{x}_t) \right) p(\mathbf{x}_t \mid \mathbf{x}_T) \right] \right]$$

$$+ \frac{1}{2} g^2(t) \nabla_{\mathbf{x}_t} \cdot \nabla_{\mathbf{x}_t} \mathbb{E}_{\mathbf{x}_T \sim p(\mathbf{x}_T)} \left[ p(\mathbf{x}_t \mid \mathbf{x}_T) \right] \tag{61}$$

and so

$$\frac{\partial}{\partial t} p(\mathbf{x}_t) = - \nabla_{\mathbf{x}_t} \cdot \left( \mathbf{f}(\mathbf{x}_t, t) p(\mathbf{x}_t) \right) - g^2(t) \nabla_{\mathbf{x}_t} \cdot \mathbb{E}_{\mathbf{x}_T \sim p(\mathbf{x}_T)} \left[ p(\mathbf{x}_t \mid \mathbf{x}_T) \nabla_{\mathbf{x}_t} \log p(\mathbf{x}_T \mid \mathbf{x}_t) \right]$$

$$+ \frac{1}{2} g^2(t) \nabla_{\mathbf{x}_t} \cdot \nabla_{\mathbf{x}_t} p(\mathbf{x}_t) \tag{62}$$

and the second term can be reduced by writing the expectation explicitly as

$$\mathbb{E}_{\mathbf{x}_T \sim p(\mathbf{x}_T)} \left[ p(\mathbf{x}_t \mid \mathbf{x}_T) \nabla_{\mathbf{x}_t} \log p(\mathbf{x}_T \mid \mathbf{x}_t) \right]$$

$$= \int_{\mathbf{x}_T} p(\mathbf{x}_T) p(\mathbf{x}_t \mid \mathbf{x}_T) \nabla_{\mathbf{x}_t} \log p(\mathbf{x}_T \mid \mathbf{x}_t) d\mathbf{x}_T \tag{63}$$

$$= p(\mathbf{x}_t) \int_{\mathbf{x}_T} p(\mathbf{x}_T \mid \mathbf{x}_t) \nabla_{\mathbf{x}_t} \log p(\mathbf{x}_T \mid \mathbf{x}_t) d\mathbf{x}_T \tag{64}$$

$$= p(\mathbf{x}_t) \int_{\mathbf{x}_T} \cancel{p(\mathbf{x}_T \mid \mathbf{x}_t)} \frac{\nabla_{\mathbf{x}_t} p(\mathbf{x}_T \mid \mathbf{x}_t)}{\cancel{p(\mathbf{x}_T \mid \mathbf{x}_t)}} d\mathbf{x}_T \tag{65}$$

$$= p(\mathbf{x}_t) \nabla_{\mathbf{x}_t} \int_{\mathbf{x}_T} p(\mathbf{x}_T \mid \mathbf{x}_t) d\mathbf{x}_T \tag{66}$$

$$= \mathbf{0} \tag{67}$$

Therefore, the resulting probability flow ODE is

$$\frac{\partial}{\partial t} p(\mathbf{x}_t) = -\nabla_{\mathbf{x}_t} \cdot \Big(\mathbf{f}(\mathbf{x}_t, t) p(\mathbf{x}_t)\Big) + \frac{1}{2} g^2(t) \nabla_{\mathbf{x}_t} \cdot \nabla_{\mathbf{x}_t} p(\mathbf{x}_t) \tag{68}$$

which is that of a regular diffusion. Therefore, by setting data distribution $q_{\text{data}}(\mathbf{x}_0, \mathbf{x}_T)$ to be $p(\mathbf{x}_T \mid \mathbf{x}_0) q_{\text{data}}(\mathbf{x}_0)$ we recover unconditional diffusion models.

**OT-Flow Matching and Rectified Flow.** As proposed, we use a VE schedule such that $\mathbf{f}(\mathbf{x}_t, t) = \mathbf{0}$ and $\sigma_t^2 = c^2 t$ for some constant $c \in [0, 1]$. Then the probability flow ODE conditioned on $\mathbf{x}_0, \mathbf{x}_T$ becomes

$$d\mathbf{x}_t = -c^2 \Big[ \frac{1}{2} \nabla_{\mathbf{x}_t} \log q(\mathbf{x}_t \mid \mathbf{x}_0, \mathbf{x}_T) - \log p(\mathbf{x}_T \mid \mathbf{x}_0) \Big] dt \tag{69}$$

Specifically, the drift term $D = -c^2 \Big[ \frac{1}{2} \nabla_{\mathbf{x}_t} \log q(\mathbf{x}_t \mid \mathbf{x}_0, \mathbf{x}_T) - \log p(\mathbf{x}_T \mid \mathbf{x}_0) \Big]$ becomes

$$D = -\frac{1}{2} c^2 \left[ -\frac{\boldsymbol{\epsilon}}{c\sqrt{t(1 - \frac{t}{T})}} + 2 \frac{(\frac{t}{T}\mathbf{x}_T + (1 - \frac{t}{T})\mathbf{x}_0 + c\sqrt{t(1 - \frac{t}{T})}\boldsymbol{\epsilon} - \mathbf{x}_T)}{c^2(T - t)} \right] \tag{70}$$

where $\mathbf{x}_t = \frac{t}{T}\mathbf{x}_T + (1 - \frac{t}{T})\mathbf{x}_0 + c\sqrt{t(1 - \frac{t}{T})}\boldsymbol{\epsilon}$. And we can rearrange the terms to be

$$D = \left[ -\frac{(\frac{t}{T}\mathbf{x}_T + (1 - \frac{t}{T})\mathbf{x}_0 - \mathbf{x}_T)}{(T - t)} \right] + \mathcal{O}(c) \tag{71}$$

$$= \left[ -\frac{((1 - \frac{t}{T})\mathbf{x}_0 - (1 - \frac{t}{T})\mathbf{x}_T)}{(T - t)} \right] + \mathcal{O}(c) \tag{72}$$

$$= \left[ \frac{(\mathbf{x}_T - \mathbf{x}_0)}{T} \right] + \mathcal{O}(c) \tag{73}$$

And by taking $c \to 0$, we have $\lim_{c \to 0} D = \mathbf{x}_1 - \mathbf{x}_0$ for $T = 1$. Therefore the network learns to match this drift term in the noiseless limit of denoising diffusion bridge in OT-Flow Matching and Rectified Flow case.

We next note that the original score-matching loss is no longer valid as bridge noise $\hat{\sigma}_t \to 0$ causes exploding magnitude of bridge score $\nabla_{\mathbf{x}_t} \log q(\mathbf{x}_t | \mathbf{x}_0, \mathbf{x}_T)$. We can then resort to matching against $\lim_{c \to 0} D$ altogether.

One additional caveat is that our framework as presented needs to take in $\mathbf{x}_T$ as an additional condition. To handle this, we note the generalized parameterization can be used to define $s_\theta(\mathbf{x}_t, \mathbf{x}_T, t) = c_{\text{skip1}}(t)\mathbf{x}_t + c_{\text{skip2}}(t)\mathbf{x}_T + c_{\text{out}}(t)V_\theta(\mathbf{x}_t, t)$ where $V_\theta(\mathbf{x}_t, t)$ is our actual network. We then set $c_{\text{skip1}}(t) = c_{\text{skip2}}(t) = 0$ and $c_{\text{out}}(t) = 1$ and uses loss $\mathbb{E}_{\mathbf{x}_t, t}\left[ \|s_\theta(\mathbf{x}_t, \mathbf{x}_T, t) - (\mathbf{x}_T - \mathbf{x}_0)\|^2 \right] = \mathbb{E}_{\mathbf{x}_t, t}\left[ \|V_\theta(\mathbf{x}_t, \mathbf{x}_T, t) - (\mathbf{x}_T - \mathbf{x}_0)\|^2 \right]$, which is the case of OT-Flow-Matching and Rectified Flow.

## A.5  Generalized Parameterization

We now derive the EDM scaling functions from first principle, as suggested by (Karras et al., 2022).

Let $a_t = \alpha_t/\alpha_T * \text{SNR}_T/\text{SNR}_t$, $b_t = \alpha_t(1 - \text{SNR}_T/\text{SNR}_t)$, $c_t = \sigma_t^2(1 - \text{SNR}_T/\text{SNR}_t)$. First, we expand the pred-$\mathbf{x}$ objective as

$$\mathbb{E}_{\mathbf{x}_t, \mathbf{x}_0, \mathbf{x}_T, t}\left[ \tilde{w}(t) \|c_{\text{skip}}(t)\mathbf{x}_t + c_{\text{out}}F_\theta(c_{\text{in}}(t)\mathbf{x}_t, c_{\text{noise}}(t)) - \mathbf{x}_0\|^2 \right]$$

where $\mathbf{x}_t = a_t\mathbf{x}_T + b_t\mathbf{x}_0 + \sqrt{c_t}\boldsymbol{\epsilon}$ for $\boldsymbol{\epsilon} \sim \mathcal{N}(\mathbf{0}, \boldsymbol{I})$. To derive $c_{\text{in}}(t)$, we set the variance of the resulting input $c_{\text{in}}(t)\mathbf{x}_t$ to be 1, where

$$c_{\text{in}}^2(t)\Big(a_t^2\sigma_T^2 + b_t^2\sigma_0^2 + 2a_t b_t \sigma_{0T} + c_t\Big) = 1 \tag{74}$$

$$\implies c_{\text{in}}(t) = \frac{1}{\sqrt{a_t^2\sigma_T^2 + b_t^2\sigma_0^2 + 2a_t b_t \sigma_{0T} + c_t}} \tag{75}$$

For simplicity we denote the neural network as $F_\theta$, and the inner square loss can be expanded to be

$$\tilde{w}(t)\left\| c_{\text{skip}}(t)\Big(a_t \mathbf{x}_T + b_t \mathbf{x}_0 + \sqrt{c_t}\boldsymbol{\epsilon}\Big) + c_{\text{out}}F_\theta - \mathbf{x}_0 \right\|^2 \tag{76}$$

$$= \tilde{w}(t)c_{\text{out}}^2(t)\left\| F_\theta - \frac{1}{c_{\text{out}}(t)}\left(\Big[1 - c_{\text{skip}}(t)b_t\Big]\mathbf{x}_0 - c_{\text{skip}}(t)\Big[a_t\mathbf{x}_T + \sqrt{c_t}\boldsymbol{\epsilon}\Big]\right)\right\|^2 \tag{77}$$

And we want the prediction target to have variance 1, thus

$$\frac{1}{c_{\text{out}}^2(t)}\left(\Big[1 - c_{\text{skip}}(t)b_t\Big]^2\sigma_0^2 + c_{\text{skip}}(t)^2\Big[a_t\sigma_T^2 + c_t\Big] - 2\Big[1 - c_{\text{skip}}(t)b_t\Big]c_{\text{skip}}(t)a_t\sigma_{0T}\right) = 1 \tag{78}$$

and

$$c_{\text{out}}^2(t) = \Big[1 - c_{\text{skip}}(t)b_t\Big]^2\sigma_0^2 + c_{\text{skip}}(t)^2\Big[a_t\sigma_T^2 + c_t\Big] - 2\Big[1 - c_{\text{skip}}(t)b_t\Big]c_{\text{skip}}(t)a_t\sigma_{0T} \tag{79}$$

Following reasoning in Karras et al. (2022), we minimize $c_{\text{out}}(t)^2$ w.r.t. $c_{\text{skip}}(t)$ by taking derivative and set to 0, which is

$$-2(1 - c_{\text{skip}}(t)b_t)b_t\sigma_0^2 + 2c_{\text{skip}}(t)(a_t^2\sigma_T^2 + c_t) - 2(1 - 2c_{\text{skip}}(t)b_t)a_t\sigma_{0T} = 0 \tag{80}$$

and this implies

$$c_{\text{skip}}(t) = \frac{b_t\sigma_0^2 + a_t\sigma_{0T}}{a_t^2\sigma_T^2 + b_t^2\sigma_0^2 + 2a_tb_t\sigma_{0T} + c_t} \tag{81}$$

$$= \Big(b_t\sigma_0^2 + a_t\sigma_{0T}\Big) * c_{\text{in}}(t)^2 \tag{82}$$

And

$$c_{\text{out}}^2(t) = \sigma_0^2 - 2c_{\text{skip}}(t)b_t\sigma_0^2 + \Big(b_t\sigma_0^2 + a_t\sigma_{0T}\Big)c_{\text{skip}}(t) - 2c_{\text{skip}}(t)a_t\sigma_{0T} \tag{83}$$

$$= \sigma_0^2 - \Big(b_t\sigma_0^2 + a_t\sigma_{0T}\Big)c_{\text{skip}}(t) \tag{84}$$

$$= \frac{a_t^2(\sigma_0^2\sigma_T^2 - \sigma_{0T}^2) + \sigma_0^2 c_t}{a_t^2\sigma_T^2 + b_t^2\sigma_0^2 + 2a_tb_t\sigma_{0T} + c_t} \tag{85}$$

$$\implies c_{\text{out}}(t) = \sqrt{a_t^2(\sigma_0^2\sigma_T^2 - \sigma_{0T}^2) + \sigma_0^2 c_t} * c_{\text{in}}(t) \tag{86}$$

Finally, $\tilde{w}(t)c_{\text{out}}(t)^2(t) = 1 \implies \tilde{w}(t) = 1/c_{\text{out}}(t)^2$, and for time, we simply reuse that proposed in Karras et al. (2022) as no significant change in time's distribution.

**EDM (Karras et al., 2022) as a special case.** In the case of unconditional diffusion models, we have $\mathbf{x}_T = \mathbf{x}_0 + T\boldsymbol{\epsilon}$, so $\sigma_T^2 = \sigma_0^2 + T^2$ and $\sigma_{0T} = \sigma_0^2$. Additionally, $a_t = t^2/T^2$, $b_t = (1 - t^2/T^2)$,

$c_t = t^2(1 - t^2/T^2)$. Substituting in these into the coefficients, we have

$$c_{\text{in}}(t) = \frac{1}{\sqrt{\frac{t^4}{T^4}(\sigma_0^2 + T^2) + (1 - \frac{t^2}{T^2})^2\sigma_0^2 + 2\frac{t^2}{T^2}(1 - \frac{t^2}{T^2})\sigma_0^2 + t^2(1 - \frac{t^2}{T^2})}} \tag{87}$$

$$= \frac{1}{\sqrt{\frac{t^4}{T^4}\sigma_0^2 + \frac{t^4}{T^2} + (1 - \frac{t^2}{T^2})^2\sigma_0^2 + 2\frac{t^2}{T^2}\sigma_0^2 - 2\frac{t^4}{T^4}\sigma_0^2 + t^2 - \frac{t^4}{T^2}}} \tag{88}$$

$$= \frac{1}{\sqrt{\sigma_0^2 + t^2}} \tag{89}$$

$$c_{\text{skip}}(t) = \frac{(1 - \frac{t^2}{T^2})\sigma_0^2 + \frac{t^2}{T^2}\sigma_0^2}{\sigma_0^2 + t^2} \tag{90}$$

$$= \frac{\sigma_0^2}{\sigma_0^2 + t^2} \tag{91}$$

$$c_{\text{out}}(t) = \sqrt{\frac{t^4}{T^4}(\sigma_0^2(\sigma_0^2 + T^2) - \sigma_0^4) + \sigma_0^2 t^2(1 - \frac{t^2}{T^2})} * c_{\text{in}}(t) \tag{92}$$

$$= \sqrt{\frac{t^4}{T^4}(\sigma_0^4 + \sigma_0^2 T^2 - \sigma_0^4) + \sigma_0^2 t^2(1 - \frac{t^2}{T^2})} * c_{\text{in}}(t) \tag{93}$$

$$= \sqrt{\frac{t^4}{T^2}\sigma_0^2 + \sigma_0^2 t^2 - \sigma_0^2\frac{t^4}{T^2}} * c_{\text{in}}(t) \tag{94}$$

$$= \frac{\sigma_0 t}{\sqrt{\sigma_0^2 + t^2}} \tag{95}$$

And $\tilde{w}(t) = 1/c_{\text{out}}^2(t) = (\sigma_0^2 + t^2)/(\sigma_0^2 t^2) = 1/t^2 + 1/\sigma_0^2$.

### A.6 SAMPLER DISCRETIZATION

EDM introduces Heun sampler, which discretizes the sampling steps into $t_0 < t_1 \cdots < t_N$ where

$$t_{i>0} = \left(T^{\frac{1}{\rho}} + \frac{N-i}{N-1}(t_{\min}^{\frac{1}{\rho}} - T^{\frac{1}{\rho}})\right)^\rho \quad \text{and} \quad t_0 = 0 \tag{96}$$

and $\rho = 7$ is a default choice. It then integrates over the probability flow ODE path with second-order Heun steps for each such discretization step. We reuse this discretization for all our experiments.

## B EXPERIMENT DETAILS

**Hybrid Sampler.** We present in Algorithm 1 our hybrid sampler.

**Architecture.** For unconditional generation, architectures are reused from Karras et al. (2022) for both CIFAR-10 and FFHQ-64×64. For pixel-space translation, we use ADM (Dhariwal and Nichol, 2021) architecture for both 64×64 and 256×256 resolutions. For latent-space translation, which reduces to 32×32 resolution in the latent space, we use ADM (Dhariwal and Nichol, 2021) architecture for 64×64 resolution but change the channel dimensions from 192 to 256 and reduce the number of residual blocks from 3 to 2, and we fix everything else to be same as that for 64×64 resolution. We use 0.1 dropout for all models. Conditioning is done via concatenation at the input level.

**VE and VP bridge parameterization.** There are many schedules we can choose for both types of bridges. For all our experiments, VE bridges follow $\sigma_t = t$ and $\alpha_t = 1$ and VP bridges follow a linear drift schedule (Song et al., 2020b) with $\mathbf{f}(\mathbf{x}_t, t) = -0.5t(\beta_1 - \beta_0) - 0.5\beta_0$. We choose $\beta_1 = 2.1$ and $\beta_0 = 0.1$ because the resulting bridge is close to VE schedule. We observe that dramatically increasing drift causes the max noise to shift towards a higher $t$ and the noise decreases faster to 0 at $t = T$ than for a symmetric bridge. This makes the learning process more difficult and degrades performance.

**Training.** We use AdamW optimizer with 0.0001 learning rate and no weight decay. The batch size is 256 for all image size less than 256 and training is done on 4 NVIDIA A100 40G. For

---

**Algorithm 1** Denoising Diffusion Bridge Hybrid Sampler

---

**Input:** model $D_\theta(\mathbf{x}_t, t)$, time steps $\{t_i\}_{i=0}^N$, max time $T$, guidance strength $w$, step ratio $s$, distribution $q_{\text{data}}(\mathbf{y})$
**Output:** $\mathbf{x}_0$
**Sample** $\mathbf{x}_N \sim q_{\text{data}}(\mathbf{y})$
**for** $i = N, \ldots, 1$ **do**
    **Sample** $\boldsymbol{\epsilon}_i \sim \mathcal{N}(\mathbf{0}, \boldsymbol{I})$
    $\hat{t}_i \leftarrow t_i + s(t_{i-1} - t_i)$
    $\boldsymbol{d}_i \leftarrow -\mathbf{f}(\mathbf{x}_i, t_i) + g^2(t_i)\Big(\mathbf{s}(\mathbf{x}_i, t_i, \mathbf{x}_N, T) - \mathbf{h}(\mathbf{x}_i, t_i, \mathbf{x}_N, T)\Big)$
    $\hat{\mathbf{x}}_i \leftarrow \mathbf{x}_i + \boldsymbol{d}_i(\hat{t}_i - t_i) + g(t_i)\sqrt{\hat{t}_i - t_i}\boldsymbol{\epsilon}_i$
    $\hat{\boldsymbol{d}}_i \leftarrow -\mathbf{f}(\hat{\mathbf{x}}_i, \hat{t}_i) + g^2(\hat{t}_i)\Big(\frac{1}{2}\mathbf{s}(\hat{\mathbf{x}}_i, \hat{t}_i, \mathbf{x}_N, T) - w\mathbf{h}(\hat{\mathbf{x}}_i, \hat{t}_i, \mathbf{x}_N, T)\Big)$
    $\mathbf{x}_{i-1} \leftarrow \hat{\mathbf{x}}_i + \hat{\boldsymbol{d}}_i(t_{i-1} - \hat{t}_i)$
    **if** $i \neq 1$ **then**
        $\boldsymbol{d}'_i \leftarrow -\mathbf{f}(\mathbf{x}_{i-1}, t_{i-1}) + g^2(t_{i-1})\Big(\frac{1}{2}\mathbf{s}(\mathbf{x}_{i-1}, t_{i-1}, \mathbf{x}_N, T) - w\mathbf{h}(\mathbf{x}_{i-1}, t_{i-1}, \mathbf{x}_N, T)\Big)$
        $\mathbf{x}_{i-1} \leftarrow \hat{\mathbf{x}}_i + (\frac{1}{2}\boldsymbol{d}'_i + \frac{1}{2}\hat{\boldsymbol{d}}_i)(t_{i-1} - \hat{t}_i)$
    **end if**
**end for**

---

$256\times256$ resolution, the batch size is 4 accumulated 4 times such that the effective batch size is 64, trained on 4 NVIDIA A100 40G. The training is terminated at 500K iterations. During training, for image-to-image translation, we set $\sigma_0 = \sigma_T = 0.5$, $\sigma_{0T} = \sigma_0^2/2$, and for unconditional generation, we set $\sigma_0 = 0.5$, $\sigma_T = \sqrt{\sigma_0^2 + T^2}$ and $\sigma_{0T} = \sigma_0^2$. We use random flipping as our data augmentation for image-to-image translation and reuse augmentation from Karras et al. (2022) for generation.

**Baselines.** All baselines are trained using the same architecture as ours for each experiment. For SDEdit, we use pretrained EDM model on $\mathbf{x}_0$ and conduct image-to-image translation by first noising $\mathbf{x}_T$ and denoising using the pretrained model. We reuse the noise schedule proposed by (Karras et al., 2022) and for reasonable generation while retaining global structure of the image conditions, we noise $\mathbf{x}_T$ using the noise variance indexed at 1/3 of EDM noise schedule and denoise starting from this noised image for the remaining 1/3 of total of $N$ steps. For DDIB, we train two separate unconditional models starting for $\mathbf{x}_0$ and $\mathbf{x}_T$ separately and perform translation by reversing DDIM starting from $\mathbf{x}_T$ and generating using DDIM for $\mathbf{x}_0$. We reuse the original baseline code for all baselines while.

**Sampling.** For all experiments we evaluate models on a low-step regime, *i.e.* the same number of sampling steps. For all experiments, we set guidance scale $w = 0.5$ and for image translation and unconditional generation, we use euler step ratio ratio $s = 0.33$ and $s = 0$ respectively. In case of $s = 0$, no Euler step is done. With these settings, we set $N = 18$, or NFE $= 53$, for $32\times32$ resolution image translation, and for all other resolutions, we use $N = 40$, or NFE $= 118$, for image translation. For unconditional generation, $N = 18 \implies$ NFE $= 36$ for CIFAR-10 and $N = 40 \implies$ NFE $= 79$ for FFHQ-64$\times$64. FID and IS scores are calculated using the entire training set for all datasets for image translation tasks. They are calculated using 50K samples for unconditional generation tasks.

**Additional visualization** We give additional visualization from our model below.

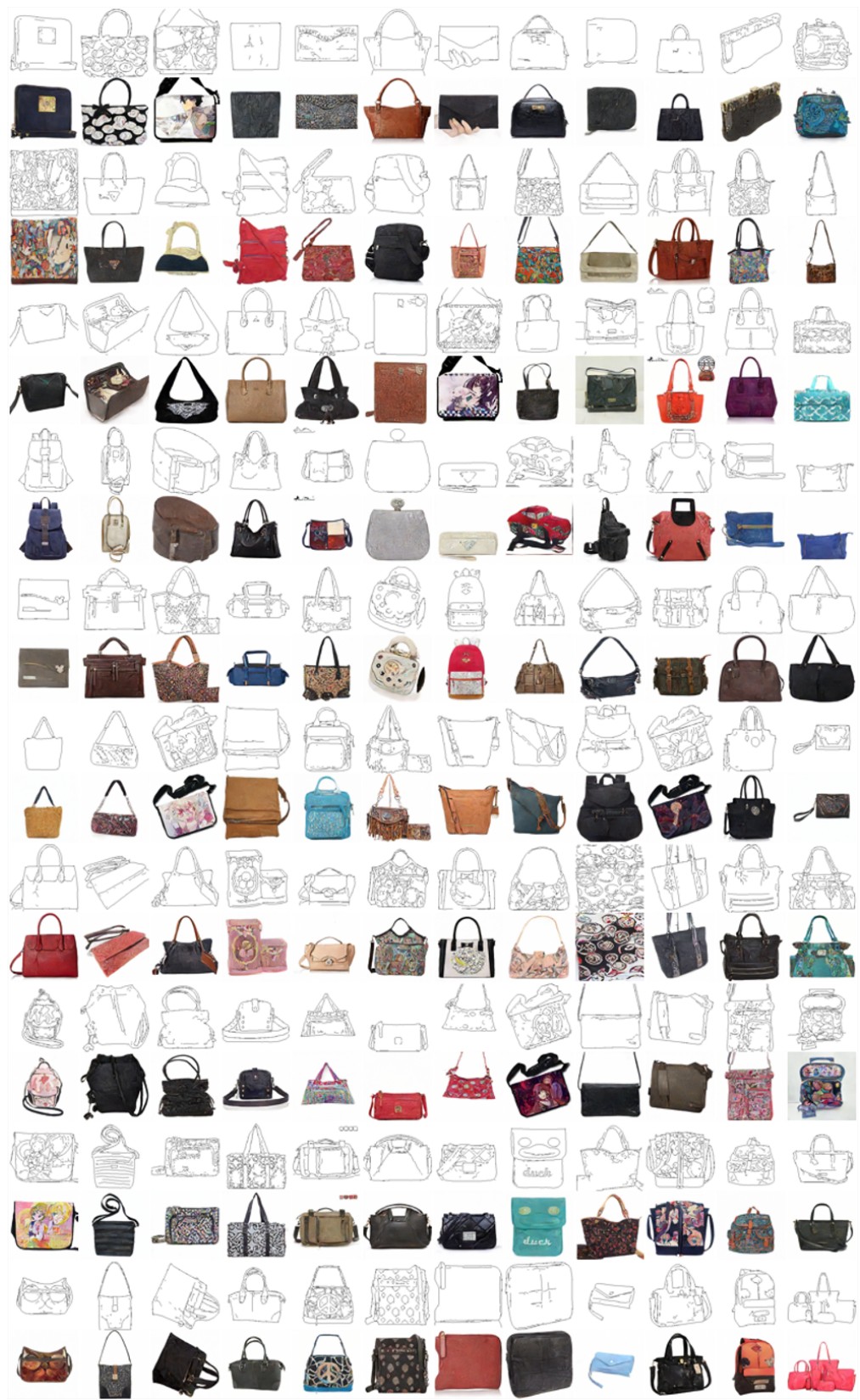

Figure 6: Additional Edges→Handbags results.

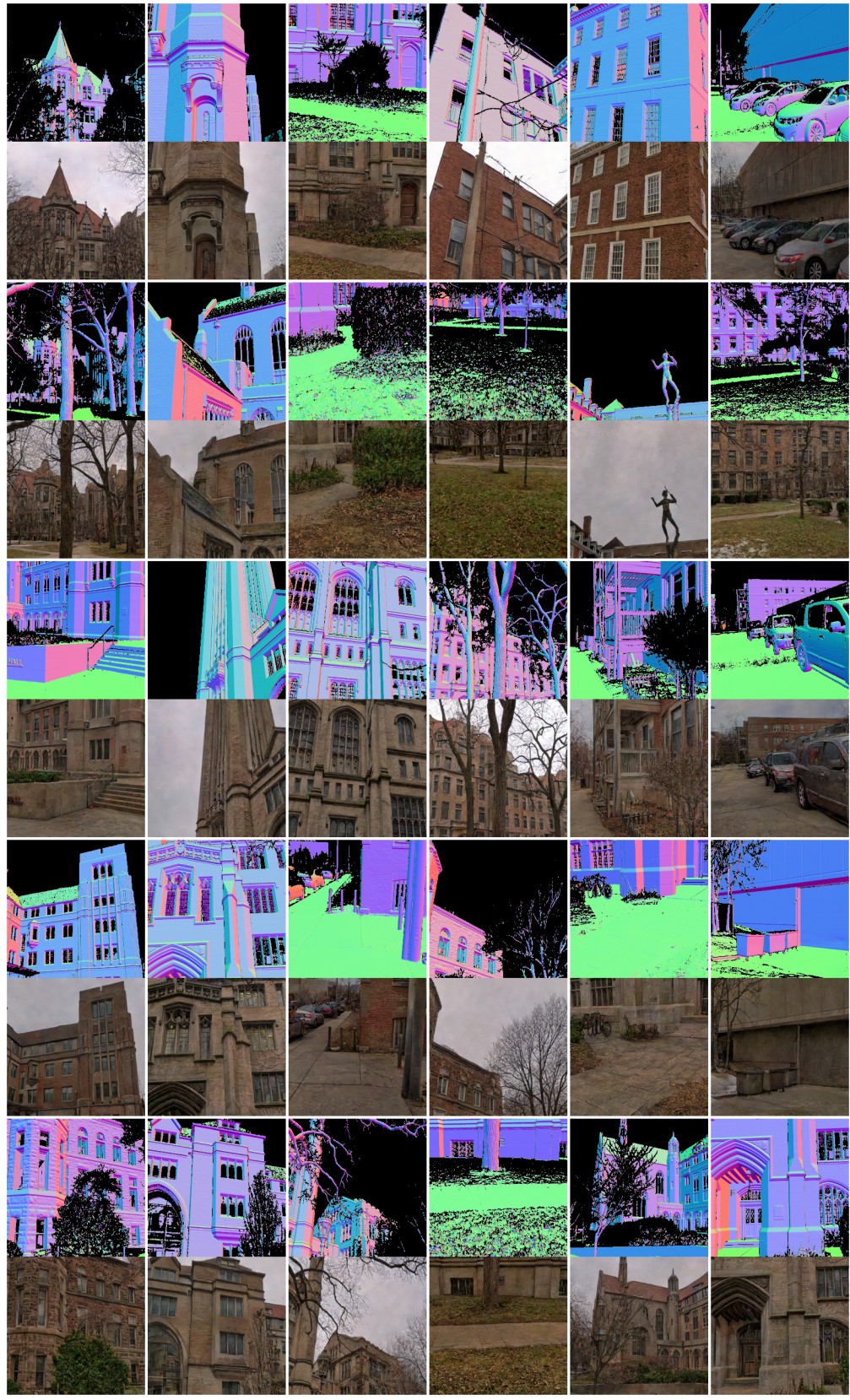

Figure 7: Additional DIODE results.

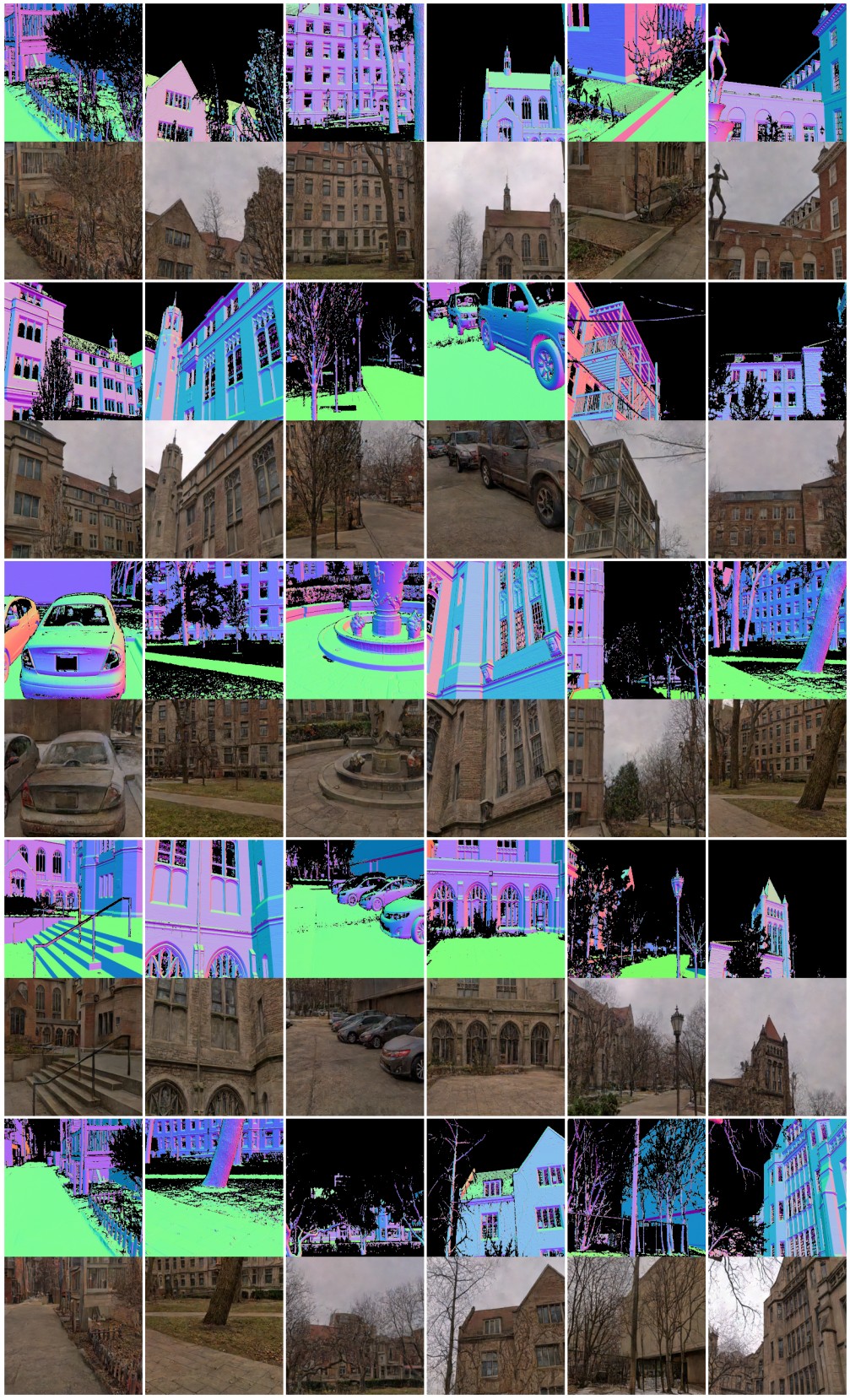

Figure 8: Additional DIODE results.