# OpenReview forum: "Denoising Diffusion Bridge Models"
_ICLR.cc/2024/Conference — ICLR 2024 poster_

### Official Review · Reviewer_BgL3 · 2023-10-29

**Soundness:** 4 excellent
**Presentation:** 2 fair
**Contribution:** 2 fair
**Rating:** 6
**Confidence:** 4

**Summary:**

The authors describe diffusion generative model as a time reversal of a Brownian bridge forward process, where the Brownian bridge end points are paired samples from a coupling of two marginal distributions. The generative model is learnt as a time reversal of a Brownian bridge, amortized across pairs by learning the score conditioned on an end point.

The authors show strong empirical results and adapt parameterization of networks from current state of the art diffusion models.

This is closely related to existing diffusion model, diffusion bridge models and bridge matching methods.

Essentially this is a regular denoising diffusion model where the forward and backward process is conditioned on the terminal point. This is also a conditional bridge matching approach where the reverse bridge is matched.

**Strengths:**

- The authors show excellent empirical performance of the method and adapt network parameterization from current state of the art diffusion models to bridges.
- The authors extend the time reversal approach of [1] to the generative setting by amortizing the time reversal across multiple pairs of points from the marginal distributions. Whereas [1] uses general and in particular nonlinear forward process, this work uses a Brownian bridge where the bridge conditioned on end points can now be sampled in closed form and the forward h-transform is also known. Although this is not useful for the applications considered in [1], it is useful for generative modelling where one does not care about the reference process.
- The work is well explained and is essentially a continuous time version of [2], which does not detail the time reversal interpretation very well and does not condition on the end marginal in the network, but does in the h-transform.
- Ultimately this is a conditional bridge matching approach which seems to empirically have a better coupling than non conditioned bridges.

[1] Heng et al. Simulating Diffusion Bridges with Score Matching, 2021 \
[2] Li et al BBDM: Image-to-Image Translation with Brownian Bridge Diffusion Models, 2022

**Weaknesses:**

**Novelty**
Ultimately this work trains a conditional diffusion model through score matching, and conditioning on the terminal state x_T from a given coupling. The difference to regular diffusion models is simply conditioning the forward process as well as the backward process with additional information from a coupling. This can also be viewed as conditional bridge matching, by matching the reverse bridge.

Although I very much like the connection to bridge matching, time reversal and diffusion bridges, this is not discussed to a satisfactory standard.

**Lack of discussion to prior work**

The authors' work severely lacks a detailed related works section. Many related works and contributions have been briefly commented in passing. Without proper acknowledgement prior work, the authors' contributions appear inflated.

1) Lacking reference [2], this appears to be the same idea but for discrete interpretation of diffusion models / bridge, derived through a variational approach similar to DDPM rather than time reversal - but which are well known to be equivalent. [2] does not condition on the end marginal in the network, but does in the h-transform.

2) Lack of discussion to bridge matching prior work

There is a lack of discussion to [1,4,5,6,9], which the authors are aware of given these papers are cited. Although phrased differently they appear highly related to this work. [1,4,6,9] similarly sample from a coupling and train a network for the drift of the of the backward or forward diffusion process (which also involves a h transform), but not conditioned on a one of the marginal points. This work considers learning the the drift of the backward process by learning the score (the score and h-transform (forward drift) make up the reverse drift, as shown in [5]), but instead conditioned on the starting point, x_T. I imagine this work is essentially a conditional bridge matching approach.

As discussed above, [5] is highly related in that the high level approach is the same: train a network between two points by reversing a diffusion bridge. The differences lie in [5] focusing on general / nonlinear forward process for sampling bridges rather than generative modelling. [5] requires simulating the forward diffusion as nonlinear forward bridges cannot be sampled in closed form. [5] is for a fixed pair rather than amortized across pairs. Indeed, to further illustrate the similarities **Equation 6 within Theorem 1 of this work appears to coincide with the equation written below Equation (4) in Section 2.1 of [5].**.

3) Misrepresenting prior work

The authors claim "A related work (Somnath et al.,2023) similarly establishes a diffusion bridge that translates between two distributions and has seen success in protein differentiable domain, but the training objective requires sampling an entire SDE trajectory for computation".

As far as I am aware this is not true. I do not understand how the authors came to this conclusion.

Minor:
-  is it not clear why [3] is being cited for the Schrodinger bridge interpretation of diffusion models. The approach of [3] is not a Schrodinger bridge between marginals but two diffusion models back to back with a Gaussian between, there are significantly more relevant works. If anything this is misleading the reader.

- "A recent work (Liu et al., 2023) considers a special case of SB" . I2I SB [9] does not actually result in a Schrodinger Bridge, neither does the same method detailed in Aligned SB [1]. They both perform bridge matching with respect to Brownian bridge reference diffusion but for a data driven coupling. Given the data driven coupling does not correspond to the coupling from Brownian motion, the bridge matching procedure returns a Markovian projection and hence breaks the coupling. See discussion in [7] "the Schrödinger Bridge is the unique path measure which satisfies the initial and terminal conditions, is Markov and is in the reciprocal class of Q, see (Léonard, 2014b)." One needs to iterate on the coupling and drift in order to obtain a SB.

This is a tangential remark as one does not need an optimal diffusion in order to perform conditional generation if it can be supervised with given paired samples from a coupling. However there appears to be many errors in papers being published in this area and I feel this should be addressed.

**Claim of generalizing flow matching/ bridge matching**

The connection of diffusion bridges to flow matching has been established and detailed in [7], detailing the limit as the noise coefficient goes to 0. The original flow matching paper [8] is even derived using result from [4] on bridge matching, so this is widely known. Note: [7] was public well before submission but not published.

However, given the score function in this work is conditioned on x_T, the learnt backward diffusion is non-Markovian, hence retains the initial marginal coupling and would not recover rectified flow / schrodinger bridge by iterative Markovian fitting as in [7] or in rectified flow. So I would argue this work does not subsume flow matching / drift matching.

**Summary**
Whilst I like the core idea of this paper and believe there is a contribution, I believe "contextualization relative to prior work" is lacking.

[1] Somnath et al 2023, Aligned Diffusion Schrödinger Bridges \
[2] Li et al 2022, BBDM: Image-to-Image Translation with Brownian Bridge Diffusion Models \
[3]  Su et al. 2022, Dual Diffusion Implicit Bridges for Image-to-Image Translation \
[4] Peluchetti 2022, Non-Denoising Forward-Time Diffusions \
[5] Heng et al. Simulating Diffusion Bridges with Score Matching \
[6] Liu et al., 2022, Let us Build Bridges: Understanding and Extending Diffusion Generative Models \
[7] Shi et al Diffusion Schrödinger Bridge Matching 2023 \
[8] Lipman et al 2022, Flow Matching for Generative Modeling \
[9] Liu et al 2023, I2SB: Image-to-Image Schro ̈dinger Bridge

**Questions:**

How does this work relate to bridge matching? Can the same result not be achieved through conditional bridge matching?

Is it true this work is a continuous time version of [2]?

Am I right in thinking that Theorem 1 equation (6) is the same as that derived in Section 2.1 of [1]?

[1] Heng et al. Simulating Diffusion Bridges with Score Matching, 2021
[2] Li et al BBDM: Image-to-Image Translation with Brownian Bridge Diffusion Models, 2022

**Details Of Ethics Concerns:**

I have concerns about lack of discussion of highly related and cited work. Whilst I hope this is not intentional, without proper discussion of prior work, the contributions appear inflated.

---

> ### Author Response · Authors · 2023-11-19
> **Thank you for your detailed and insightful review (Part 1).**
>
> We thank the review for pointing out our paper’s lack of discussion of related works. We greatly appreciate the thoughtful suggestions. Due to space limitation of the main text, we could not properly discuss the large body of related works in detail. According to suggestions from you and reviewer H8Nr, we have included a revised version of related works in the new supplementary files and we intend to replace the corresponding section in main text with the revised version. Please be sure to take a look.
>
> We first briefly summarize the mentioned references and contextualize our contributions before addressing individual concerns from the reviewer.
> - [1] learns to bridge two distributions by matching Doob’s h-function and proposes a simulation-free training procedure for training. [4] proposes a similar approach which learns a generative model by constructing a mixture of bridges where the intermediate $x_t$ can be marginally sampled. Both works propose to generate via forward-time simulation with the trained model. Our method instead adopts a reverse-time perspective. The key benefit from this perspective is the ability to integrate key diffusion model techniques into the bridge framework, such as an accurate higher-order sampler and network preconditioning. Since these previous works are developed with other perspectives, it is less obvious how one would integrate these diffusion model design choices and achieve empirical improvement. We also extend beyond the Brownian bridge construction (as used in [1,2,6,7], which is a special case of VE bridge) and show that both VE and VP bridge can achieve competitive results than state-of-the-art.
>
> - [2] proposes to connect two image domains via discrete Brownian bridges, and proposes to reverse the bridge for generation. This is a discrete-time special case of the VE bridge we consider, and our method also shows success on VP bridges that work well in continuous time. In addition, our sampler takes much fewer steps for best generation results.
>
> - [5] adopts a reverse-time perspective of diffusion bridges pinned on both ends and proposes score-matching for inference time simulation. Our method considers a more general case where both ends are drawn from distributions rather than fixed, and our results are developed for this case. In addition, [5] requires simulation of an entire trajectory for score-matching training, while our method extends directly from diffusion models and naturally results in a simulation-free training procedure
> - [6] is also built on learning Doob’s h-function, similar to [1]. However, they also propose to perform forward time simulation during training for matching against the Doob’s h-function, which we avoid.
> - [7] proposes bridge-matching and Iterative Markovian Fitting, an iterative procedure for solving the Schrodinger bridge problem. This procedure involves an inner loop of optimization and simulation, with the exception of the first iteration where the intermediate $X_t$ can be marginally sampled given endpoints. Our method is different in that our method is completely simulation-free, and directly extend diffusion models for the bridge construction. We further discuss connections in the main response.
> - [8] is a normalizing-flow based generative model which relies on deterministic ODEs for generation. Our method is instead based on SDEs, and we also show the introduced stochasticity in our sampler is integral for empirical success. In the noiseless limit of our method with the VE bridge, our method reduces to the method introduced in [8].
> - [9] is an image-to-image method based on a special case of SB which is tractably computable and also results in a simulation-free algorithm. Our method is functionally similar but theoretically different. Different from [9], our method can theoretically subsume other classes of models and directly adapt successful design choices to improve quality and speed. We also empirically outperform them on image translation.
>
> We now address our the reviewer’s concern below.
>
> > Is our model a continuous-time version of (Li et al., 2022)?
>
> Our method is a superset of the method proposed in (Li et al., 2022). Their work proposes a simple Brownian bridge formulation in discrete time and seeks to reverse the discrete-time bridge using ancestral sampling adopted from diffusion models. In continuous-time, their Brownian bridge formulation is a special case of our VE bridges. Conversely, our VP bridges, which generalize VP diffusions, are separate and are also shown to work well empirically.

---

> ### Author Response · Authors · 2023-11-19
> **Thank you for your detailed and insightful review (Part 2).**
>
> > Discussion with bridge-matching.
>
> We want to note that our work was concurrently developed with bridge matching (Shi et al., 2023) and some property of our method may seem similar. Bridge matching directly aims to solve Schrodinger Bridge problem and find the minimum cost bridge through Iterative Markovian Fitting, which relies on an inner loop for optimizing bridge approximation and simulating the bridge trajectory (with the exception of the first iteration where they marginally sample intermediate $X_t$ ). It is shown to be a superset of Flow-Matching. Somnath et al. (2023) also adopts a learning procedure similar to the first iteration of bridge matching, and learns to fit Doob’s h-function. We also acknowledge that a similar procedure is developed by Peluchetti, (2022) which shows a model can learn a mixture of bridges to map from one distribution to the next.
>
> Our framework is related in that we also build from Doob’s h-transform for bridging between distributions and in theory propose to match scores of intermediate distributions. However, we note that we do not try to solve the Schrodinger Bridge problem to find the optimal coupling but try to directly use a preexisting diffusion schedule to establish this coupling. This allows us to reuse previously successful design choices for diffusion models and generalize them for the bridge case. This coupling (Eq. 8) is also written in light of diffusion models which sheds light on its connection with the underlying diffusion process. Both Shi et al. (2023) and Somnath et al. (2023) empirically consider Brownian bridge for connecting two distributions. We go beyond this simple choice and investigate an alternative VE bridge and a VP bridge design. The VE bridge uses different interpolation velocity inspired by EDM (Karras et al., 2022), which is crucial for our higher-order sampler. The VP bridge directly builds on the VP diffusion model (which few prior works consider), and we similarly empirically show its success on the same tasks. Since our method is directly built on diffusion models, we generalize successful designs such as high-order sampler, better time-discretization, and different noise-scheduling to show that a bridge framework is similarly powerful equipped with these generalized design choices. Therefore, our framework is both theoretically and empirically motivated, which we hope contributes to the larger body of works to push bridge-based frameworks further.
>
>
>
>
> > Discussion with [1,4,5,6,9] as cited.
>
> We have included their discussion in our write-up in the new supplementary. It is true that these works all seek to establish bridges from one distribution to another. [1,4,6] are built on Doob’s h-transform and share the high-level idea that one can match the Doob’s h-function given intermediate sample $X_t$, and the learned network can guide one towards high-likelihood regions during inference. [1,4] consider marginally sampling any intermediate $X_t$ while [6] consider simulating a path for loss calculation. [9] is built on a special case of Schodinger Bridge when one distribution is reduced to Dirac delta distribution, although empirically it is shown to work well and results in a similar algorithm for marginally sampling $X_t$ for score-matching. [5] is also highly related in that, as kindly pointed out, it reverses the diffusion bridge trajectory via score-matching. Different from more recent approaches, it simulates a forward trajectory for obtaining intermediate samples $X_t$.
>
> Our work is similar in theory to these mentioned prior works as we are also built on learning bridges via h-transform. Different from them, our work is theoretically developed by directly extending diffusion models in the hope that it can extend the empirical success of diffusion models (both in terms of quality and efficiency) on high-dimensional image synthesis tasks. The reverse-time formulation is crucial for generalizing previous successful designs to push the empirical limit on high-dimensional image translation with the bridge framework. Interestingly, the reverse-time perspective allows us to connect with different generative frameworks such as diffusion models and Flow-Matching/Rectified Flow.
>
> > Misinterpreting (Somnath et al.,2023)
>
> We appreciate the reviewer pointing this out. This mistake has been corrected.
>
> > Should cite other more related works than [3].
>
> Thank you for pointing this out. We will more prominently discuss other more related works as suggested.
>
> > I$^2$SB is not SB?
>
> Precisely speaking, according to Liu et al. (2023), the class of SB they consider is a “tractable SB with the Dirac delta boundary” (Corollary 3.2), which assumes special structure on data distribution for the algorithm to be tractably handled. We do agree that Aligned SB do not result in SB and is doing bridge matching (as the first iteration specified in the algorithm in Shi et al. (2023)).

---

> ### Author Response · Authors · 2023-11-19
> **Thank you for your detailed and insightful review (Part 3).**
>
> > Does DDBM subsume Flow-Matching/Rectified Flow?
>
> The additional input of $x_T$ is a helpful design choice that guides generation towards the data distribution. This is mostly a practical decision. However, note that our score-matching objective is sound without injecting $x_T$ as additional condition, in which case the reverse-time SDE is Markovian. Since the target score as described exactly reduces to that of Flow-Matching (the OT formulation) and Rectified Flow, our framework reduces these special cases. We have noted this point in our additional write-up.
>
> > Is Theorem 1 different from Section 2.1 in (Heng et al. 2021)?
>
> Section 2.1 in Heng et al. 2021 presents time reversal for a single diffusion bridge (pinned on both ends). Our Theorem 1 generalizes it to tackle $x_0$ drawn from a distribution. In addition, we present the probability-flow ODE version (which is crucial for our ODE sampler) that produces the same marginal distribution as the SDE.

---

> ### Author Response · Authors · 2023-11-22
> **We have addressed the concerns in Peluchetti's comment.**
>
> We have addressed the concerns in Peluchetti's comment. Let us know if anything else can be addressed.

---

> > ### Comment · Reviewer_BgL3 · 2023-11-22
> >
> > Thank you for the detailed response and engagement - I appreciate the thorough discussion.
> >
> > I will properly read the discussion / new draft and update my score appropriately.

---

> ### Comment · Reviewer_BgL3 · 2023-11-22
>
> Minor comments:
> - **"We want to note that our work was concurrently developed with bridge matching (Shi et al., 2023)"** \
> When I refer to bridge matching; I really mean the work of Peluchetti; Qiang Liu etc. Shi et al perform iterative bridge matching to get to the SB. Similar to iterated flow matching gets to rectified flow (reflow)
>
> - **"Precisely speaking, according to Liu et al. (2023), the class of SB they consider is a “tractable SB with the Dirac delta boundary” (Corollary 3.2), which assumes special structure on data distribution for the algorithm to be tractably handled. We do agree that Aligned SB do not result in SB and is doing bridge matching (as the first iteration specified in the algorithm in Shi et al. (2023)).**
>
> I think Liu et al and Aligned SB have almost identical losses in the main text. Liu et al 23 also has an extension conditioning on a marginal point too. The brownian bridge between points from an OT coupling with respect to brownian reference measure is SB; but Liu et al use a data driven coupling which is not brownian optimal, hence not a SB. Indeed, the coupling between diracs is trivial so one could argue it is a mixture of SB between diracs, but this is not a SB between the continuous support marginal measures. This is out of scope of this review, but I think it's important to mention here and not necessarily in the paper - there are lots of errors being propagated in this literature.
>
> Actually even some of the authors of Liu et al 23 recognise this is not a SB in recent work [10] "In Somnath et al.
>  (2023); Liu et al. (2023), it remains debatable whether the training dataset pairs inherently represent the solution to the static Schro ̈dinger Bridge equation. In particular, this assumption is not ensured in real-world applications and can be easily violated even in low-dimensional cases as shown in Figure 1. Hence, the bridge matching procedure does not preserve the original coupling in general."
>
>
> -----------------
>
> Less Minor:
>
> - **[8] is a normalizing-flow based generative model which relies on deterministic ODEs for generation. Our method is instead based on SDEs, and we also show the introduced stochasticity in our sampler is integral for empirical success. In the noiseless limit of our method with the VE bridge, our method reduces to the method introduced in [8]**
>
> Again I think there is actually a big distinction that this is not the case given your method conditions on x_T - this is completely fine but I think it's important to make it clear.
>
> For example, iterated versions of your method similar to Shi et al Iterative Markovian Fitting will not result in a SB but retain the initial coupling (whether independent or data driven). Very recent work appearing after submission details this distinction [10]
>
> I actually think one of the contributions of your work is indeed that you retain the coupling; this could be useful.
>
> - **Section 2.1 in Heng et al. 2021 presents time reversal for a single diffusion bridge (pinned on both ends)**
> Heng et al first presents the time reversal with a single marginal (x_0) pinned. To me this coincides to what is here? In particular the relationship between conditioned score; unconditioned score and h-transform. Then Heng et al shows if you initialize your reverse diffusion from x_T, the resulting process is a bridge from x_T to x_0.
>
> [10] Augmented Bridge Matching https://arxiv.org/abs/2311.06978
>
> -----------------
>
> **Summary**
> - Overall I think it's a good piece of work; for performance alone it is valuable to the community.
> - The discussion of prior work has been addressed to a satisfactory degree, which was my main concern. I do think it would be better to have it in the main text. I sympathise that there is a space restriction.
> - I think Peluchetti has some valid points - I appreciate his comment but also that it is not his place to push too much.
> - It is still a mystery whether the performance gap between forward bridge matching vs reverse bridge matching comes from as it does seem they are equivalent; perhaps learning the score as oppose to drift is easier? (the difference is only the h-transform). Perhaps there are better parameterisations of forward bridge matching that work equally well and the empirical results of this work paves the way for that.
> - I also still believe the connection between the forward/ reverse bridges through the unconditional score; conditional score and h-transform was already established in Heng et al 2021 which should be acknowledged (if it is the same); though the motivation was different and experiments less impressive. This work is a lot clearer and more suitable for the ML community for generative modelling.
> - I disagree that this work generalizes FM; BM etc which do not retain data driven couplings. I think the stronger selling point for this work is that it retains the coupling used in training at deployment time rather than breaking it as is done in prior work like I2I SB
>
> I have raised my score.

---

> > ### Author Response · Authors · 2023-11-22
> > **Thank you for your review**
> >
> > Thank you for your review and insights. We will take these into consideration when updating our final draft.

---

> ### Comment · Reviewer_BgL3 · 2023-11-23
>
> I want to stress, that although I like Shi et al 2023 and they have some very nice and relevant insights; that paper does not invent bridge matching but instead iterated bridge matching similar to reflow. The paper under review here does not perform iterative bridge matching. There are significantly more relevant papers on bridge matching than Shi et al.
>
> In addition, although I feel that papers should fully acknowledge all related work whether published or not. I understand this work under review here was also submitted to neurips concurrently to Shi et al. 2023. Furthermore; although Shi et al 2023 was accepted to neurips - the proceedings are not yet released or at least not at the time of submission. Hence under ICLR guidelines the authors may be forgiven for not fully discussing or comparing to it.
>
> Q: Are authors expected to cite and compare with very recent work? What about non peer-reviewed (e.g., ArXiv) papers? (updated on 7 November 2022)
>
> A: We consider papers contemporaneous if they are published (available in online proceedings) within the last four months. That means, since our full paper deadline is September 28, if a paper was published (i.e., at a peer-reviewed venue) on or after May 28, 2022, authors are not required to compare their own work to that paper. Authors are encouraged to cite and discuss all relevant papers, but they may be excused for not knowing about papers not published in peer-reviewed conference proceedings or journals, which includes papers exclusively available on arXiv. Reviewers are encouraged to use their own good judgement and, if in doubt, discuss with their area chair.
> https://iclr.cc/Conferences/2023/ReviewerGuide

---

### Official Review · Reviewer_2Uac · 2023-10-31

**Soundness:** 3 good
**Presentation:** 3 good
**Contribution:** 3 good
**Rating:** 8
**Confidence:** 3

**Summary:**

This paper presents a new class of generative models, Denoising Diffusion Bridge Models (DDBM), that can define a generative path between any 2 paired distributions. This is in contrast to current diffusion models where the path is always from Gaussian to a data distributions. DDBMs are a more generic representation, and they allow solving tasks like image-to-image translation (which is not trivial to do with regular diffusion models). DDBMs work by building a stochastic bridge between a paired samples. DDBMs share several attributes with diffusion models, which allows reusing many of the techniques already available in this field. These new models can be trained in a similar fashion as score matching, and the authors present extensive theoretical backing to allow for a loss formulation and transport between distributions via a reverse SDE and flow ODE. The authors show results in image-to-image tasks with reasonable results. The authors also show competitive results when running tasks currently done by diffusion models, mainly text-to-image generation.

**Strengths:**

The authors tackle an important and challenging problem, mainly how do we use our current generative pipelines (diffusion) to solve more general generative problems. The authors do a good job at presenting a sounds alternative. I highlight the following strengths:
- The formulation of DDBMs seems solid. The authors present a good comparison between this method and score matching and flow matching. In fact, the authors show that DDBMs are a generalization of these methods.
- The authors formulate the model in a way that shares many commonalities with current diffusion models. I appreciate this effort for two reasons. First, it makes understanding and adopting them easier given the familiarity they have with current methods. Second, it reuses many of the formulations already developed in things like score matching, which should lead to better training.
- The authors go to good length to present and explain all relevant mathematical formulations. These look sound, although I did not fully explore all the details.
- The authors show favorable results in image-to-image tasks and comparable results in text-to-image generation. Given that these come from the same model formulation (DDBMs), this combination of results is a strength of the formulation.
- The paper is well written and structured.

**Weaknesses:**

I have 2 concerns I believe are minor, but would like to hear from the authors:
- The authors only validate the work with image-to-image translation. In this domain, ControlNet is the dominant solution. I value that the 2 models might not be competing solutions. Can the authors clarify why they didn't compare against ControlNet in the image-to-image task?
- With generalized flow matching models we can in theory move between any distributions. What is the advantage of DDBMs compared to flow matching? I found some reasoning in the paper but didn't fully understand, therefore I appreciate a brief explanation from the authors.

Overall I believe this are minor issues and I hope the authors provide answers during rebuttal.

**Questions:**

- Why didn't the authors compare against ControlNet?
- What is the advantage of DDBMs when compared to FM given that both can in theory model flow between arbitrary distributions?

---

> ### Author Response · Authors · 2023-11-19
> **Thank you for your review and appreciation.**
>
> Thank you very much for your appreciation of our work! We want to address the remaining concerns below.
>
> > Why not comparing with ControlNet?
>
> You are correct. Our framework is orthogonal to ControlNet, which is an architectural design at its core. Our framework can also use ControlNet architecture and start from pretrained diffusion weights for the image-to-image tasks. However, due to limited resources and time, we did not proceed to larger scale experiments using ControlNet. Nevertheless, we do plan to scale up our framework using more popular architectures in the future.
>
> > What is the advantage of DDBM compared to Flow-Matching?
>
> In theory they both aim to learn a model to move between distributions. However, we designed our framework with the practical motivation about how can can maximally transfer empirical success of diffusion models to the bridge framework, which has seen limited empirical success in these high-dimensional image translation tasks. The framework is developed to allow reuse/generalization of many prior successful implementation choices such as the preconditioning in EDM and higher-order sampler, which Flow-Matching lack. With these choices, our framework can achieve better performance in quality and speed.

---

### Official Review · Reviewer_H8Nr · 2023-11-01

**Soundness:** 2 fair
**Presentation:** 2 fair
**Contribution:** 2 fair
**Rating:** 6
**Confidence:** 4

**Summary:**

This paper proposes a means of adapting standard denoising diffusion models, which parameterize the score in the Ornstein Uhlenbeck process to map a Gaussian to a data distribution, so that it can be more properly used to connect arbitrary densities, e.g. for image-to-image translation.

The main tool at play is Doob's h-transform, which they use to derive an SDE that has a learnable score conditioned on some endpoint $x_T$. They write a variant of the score matching loss that accommodates this.

Following this, they address the usual topics of a SBDM paper, which is how to re-write the score in a way that allows you to predict $x_t$, how to consider the signal to noise ratio in the diffusion path (which is not a very meaningful name in the case of the ODE), how to write the probability flow, and what sampler to use.

They benchmark the method on various image-to-image translation tasks, quantifying image quality metrics with a consideration for the number of function calls necessary to achieve said quality.



--generalized time reversal is not proven to converge to the correct distribution. Equation 13 should be shown to be justified.



-- can only do this given a pair

**Strengths:**

In the specific context of denoising (score-based) diffusion models (DDPM),  this paper introduces the useful perspective from the bridge literature of how to alleviate the limitation of DDPM as a generative model in that it could only connect a data distribution to a Gaussian.

The experimentation is thorough which shows that the method is performant on the image tasks presented. The ablation study is useful (even if a bit contrary to their claims in the abstract of wanting to move away from "cumbersome methods like guidance"). The authors usefully summarize how many of the tricks that make diffusions highly performant fit into their extension.

**Weaknesses:**

Let me begin by saying, thank you for the effort you put into this. The experiments are very thorough, and the doob-h formulation is sound.

Unfortunately, this paper is written in a way that highlights some misconceptions. It is unclear what the authors see as the contribution of this paper. It has become clear from the wider literature than just the narrow perspective of the same score-based diffusion equation how to do image-to-image translation. In fact, the works that this paper already cite do this, even for both the ODE and SDE between arbitrary densities.

A quick summary of related work that should be properly addressed in this paper, and the authors should highlight what actually sets this method apart:
- [1,2,3] propose a means of learning an ODE to do connect any two densities, e.g. for your image-to-image translation in finite time $ t \in [0,1]$ without bias. In fact [1] does this image-to-image translation experiment there directly.
- [4] shows how to do this for **both** an ODE and an SDE (with the score), also in finite time. It also shows how to avoid the added complexity of doob, and how SBDM is a subcase.
- [5, 6] shows the influence of changing the coupling $p(x_0, x_1)$ between the densities $p(x_0)$, $p(x_1)$
- [7] also shows how to use either and ODE or SDE to connect the densities under varying coupling.

The related worked section, in addition to the intro, should be thoroughly reworked to not overlook these clear contributions. While it is certainly beneficial to have a clear connection to how to do this with SBDM using doob-h and experiments of it, the paper is currently written so as to suggest that there is a clear need to come up with tools to do this as if many works have not addressed it. The reviewer is sympathetic to the fact that this field moves very fast, but also believes that overselling a concept by overlooking other work is detrimental to the field.

With regards to the statement that the ODE based methods [1,2,3] "tend to severely underperform when compared to diffusion models," there is no evidence of this presented, and in fact the purpose of those papers is to present evidence of the contrary, e.g. see [3]. If the authors would like to make this statement, they should demonstrate it in experiments. It is of the reviewers mindset that the differentiating factor between most simulation-free transport generative models is just whether or not the model was conditionally trained, e.g. as $s(x,y,t)$ vs $s(x,t)$. Many papers report "unconditional sampling" FIDs by training a conditional model and using a null-token e.g. $y=-1$ to sample unconditionally, which improves their score.

There is perhaps a misinterpretation of what optimal transport is, and what role it may play in generative models that are constructed from continuous-time transport plans. The authors refer to "VE (OT)" bridges without providing a clear definition of what they mean by OT. Here are the two ways to learn an optimal transport between distributions:

- 1. Choose the optimal coupling $p(x_0, x_1)$, with "schedule" $x_t = (1-t)x_0 + tx_1$. This is hard to do, and why people rely, e.g. on Sinkhorn.
- 2. For independent coupling $p(x_0)p(x_1)$, learn a process $x_t = f_{\theta}(t,x_0,x_1)$ which gives a time dependent velocity/score and density $v_t, p_t(x)$ that induces a minimum action (or least transport cost) in terms of Benamou-Brenier transport cost.

- In the abstract, the authors reference "OT-Flow-Matching", though the work [3] does not propose to solve OT. At most ,they use a relationship about McCann displacement maps (which is related to OT from a Gaussian to a Gaussian) to motivate choosing a straighter conditional probability path -- they are not doing OT.


-Throughout the text, the authors need to make sure all acronyms are defined.

- The introduction of the weighting factor in equation (13) is not shown in the text to preserve an exact/unbiased transport. The authors should justify this or state that it introduces a bias, even if a beneficial one.


*The reviewer is not opposed to adjusting their score, but there are many organizational and presentational aspects of this work that suggest it is not fit for contribution to the larger body of work at ICLR at this stage. Please find questions below that could help better demonstrate the utility of the method (in controlled experiments or new ones that e.g. exploit an interesting coupling) as well as the remarks above that need addressing.*

[1] Flow Straight and Fast: Learning to Generate and Transfer Data with Rectified Flow. *Xingchao Liu, Chengyue Gong, Qiang Liu*, Sept 2022.

[2] Building Normalizing Flows with Stochastic Interpolants. *Michael S. Albergo and Eric Vanden-Eijnden*, Sept 2022.

[3] Flow Matching for Generative Modeling. *Yaron Lipman, Ricky T. Q. Chen, Heli Ben-Hamu, Maximilian Nickel, Matt Le*, Oct 2022.

[4] Stochastic Interpolants: A Unifying Framework for Flows and Diffusions. *Michael S. Albergo, Nicholas M. Boffi, Eric Vanden-Eijnden*, March 2023.

[5] Improving and Generalizing Flow-Based Generative Models with Minibatch Optimal Transport. *Alexander Tong, Nikolay Malkin, Guillaume Huguet, Yanlei Zhang, Jarrid Rector-Brooks, Kilian Fatras, Guy Wolf, Yoshua Bengio*, Feb 2023.

[6] Multisample Flow Matching: Straightening Flows with Minibatch Couplings. *Aram-Alexandre Pooladian, Heli Ben-Hamu, Carles Domingo-Enrich, Brandon Amos, Yaron Lipman, Ricky T. Q. Chen*, April 2023.

[7] Simulation-free Schrödinger bridges via score and flow matching. *Alexander Tong, Nikolay Malkin, Kilian Fatras, Lazar Atanackovic, Yanlei Zhang, Guillaume Huguet, Guy Wolf, Yoshua Bengio*, July 2023.

**Questions:**

- The authors write down the doob-$h$ transformed SDE for arbitrary coupling, but it's not clear in the experiments that they consider any coupling besides $p(x_0, x_1) = p(x_0)p(x_1)$. Can the authors consider any experiments which make use of a more interesting coupling?
- An important ablation should be based on the sampler the authors used. Did the authors retrain the models they compared against and fixed e.g. the sampling strategy? They discuss a higher-order Euler-Maruyama sampler, but it is unclear if all the comparison to other methods used the same integrator. Many of the methods listed could use a different integrator for the same trained model, and it's unclear to the reviewer what was held fixed across comparisons and what wasn't. The reviewer points this out to stress that a paper is not a method -- the method presented in the paper should be recognizably connected to the conclusions of the experiments. If an auxiliary factor influenced the outcome, e.g. like choosing a better sampler, this diminishes the message about the method itself.

---

> ### Author Response · Authors · 2023-11-19
> **Thank you for your detailed and insightful comments. (Part 1)**
>
> Thank you very much for your detailed review and suggestions. We greatly appreciate your time and expertise. In addition to clarifying a few points of concern, we also wish to note that we have a reworked version of related works in the new supplementary docs. We wish to properly contextualize our framework within a larger body of works as mentioned.
>
> We want to start by briefly summarizing the referenced works and addressing important differences to highlight our contributions.
>
> - [1] proposes to learn an OT mapping between two different distributions by approximating the time-invariant velocity field, while our method is not based on OT. [1] is based on an ODE, while we have an SDE for distribution mapping, which in the noiseless limit, reduces to that in [1] in a special case. Our work enables a higher-order sampler and output parameterization that are not directly usable for [1].
> - [2,3] are flow-based and aim to learn transport maps to push forward a prior distribution to data distribution, and utilizes deterministic ODEs for generation. Our theory is developed using SDE and we introduce higher-order sampler with additional stochasticity, which are also not directly usable for these works. The resulting method can further subsume the case in [3] developed with OT displacement map.
> - [4] is a general theory that directly constructs a bridge using an interpolation map and avoids the use of Doob’s h-function. Our method is based on Doob’s h-function from which a direct interpolation map can be constructed for marginal sampling. Furthermore, the training objective is quite different as ours is based on denoising score-matching without any regularization, while theirs is not. Our construction also shows stronger tie with diffusion models, which allow extending existing designs for the new bridge construction. These include reusing noise schedules, speicalized higher-order sampler, and network preconditioning. It’s not clear if these choices can be directly applied to [4].
> - Extending [3], [5,6] are similarly flow-based methods that show success on exploiting potential couplings between the two distributions via minibatch simulation-free OT plan. Again, our method is fundamentally based on SDE and does not aim to solve OT. Our simulation free sampling is directly constructed from VP and VE diffusion processes. [3,5,6] all consider brownian bridge paths as as the interpolation map, while we show that this straight interpolation map is a special case of VE bridge. VP bridge exhibits curved interpolation paths. Nevertheless, when implemented correctly, we show both bridges can perform well in practice.
> - [7] constructs a simulation-free training plan using brownian bridge as the interpolation map, and aims to match against both flow vector field and bridge score. Our method does not only consider brownian bridge as the bridge construction, and shows success for other types of bridge in practice. We only consider a denoising score matching loss without additional flow-matching loss as proposed. A set of practical design choices are also proposed to further improve quality and speed of our bridge model.
>
> In the following sections we address the reviewer’s concerns one by one.
>
> > What is the contribution?
>
> Our framework aims to show that the bridge-based frameworks, when developed from the perspective of diffusion models, can also achieve competitive empirical results for both high-dimensional image-to-image and unconditional generation, which previous bridge-based frameworks have seen limited success on. The practical motivation inspires the theoretical development in directly extending diffusion models, which naturally results in a simulation-free bridge construction. Furthermore, different from previous works, our method allows for constructing bridges beyond the Brownian bridge (as mostly used in previous works such as [5,6,*]) and shows promising empirical success. Brownian bridges are a special case of VE bridges we consider. Empirically speaking, we design a set of practical choices, such as higher-order sampler with stochasticity, improved network preconditioning, by directly adapting those from prior successful designs, which are not directly applicable to prior bridge-based works.
>
> Our core contribution (in one sentence) is “connecting bridge-based frameworks with diffusion models to adapt existing diffusion model techniques for improved empirical performance”. We believe that this is important since prior bridge methods have seen limited success on tasks that they should be better suited for.

---

> ### Author Response · Authors · 2023-11-19
> **Thank you for your detailed and insightful comments. (Part 2)**
>
> > Properly addressing previous works’ contributions
>
> We thank the reviewer for the references and we appreciate the reviewer pointing out the lack of properly addressing previous works. We have thoroughly incorporated them in our new related works in supplementary. Please see our comments to all reviewers in the main response comment up above.
>
> > Statement regarding ODE-based methods’ performance.
>
> For ODE-based methods like the papers [1, 2, 3] mentioned above, we apologize for the misleading statement (which unfairly discredits these papers as being fundamentally unable to compete with diffusion).
>
> However, we wanted to emphasize that, although these methods have seen some improvements over the last year, they have yet to see the empirical success that diffusion models are accustomed to. In particular, the only case where ODE methods have consistently outperformed SDE methods (that we are aware of) is in the CIFAR10 dataset, but even then most of the improvements have stemmed from core work in the diffusion model/probability flow ODE space (e.g. EDM). We believe that this gap is fundamentally caused by a difference in research volume: diffusion models have seen many more small improvements (e.g. sampling, network parameterization, noise schedule)
>
> As such, we have updated this statement to be “ODE methods have not achieved the same empirical success as diffusion models”.
>
>
> > Misinterpreting OT?
>
> We want to stress that our framework is not solving OT, and the VE bridge resembles OT plan in that they share a similar coupling in the form of $(1-t)x_0+t x_1$ in the noiseless limit (with factor changed as $t^2$ instead of $t$ for our case). We have deleted the “OT” in brackets after “VE” in the corresponding section to avoid the potentially misleading meaning. The OT-Flow-Matching we use in our text is only short for the Flow-Matching case in [3] where the author designs their conditional probability path by OT plan. We agree that this is a slight misnomer, and our updates intend to fix any possible misinterpretation issues.
>
> > Bias introduced in the weighting factor
>
> Indeed, the weighting factor introduced causes the process to no longer follow the same marginal distribution as the bridge process if $w\neq 1$. We empirically find that this factor is useful because it allows us to interpolate between probability-flow ODE of unconditional diffusion model and that of a bridge model (by setting $w=0.5$ vs $w=1$), which is beneficial for unconditional generation with the ODE sampler generalized from EDM.
>
>
> > Different coupling between x_1 and x_0.
>
> Note that for image translation tasks we assumed distribution coupling of the form $p(x_0, x_1)=p(x_0|x_1)p(x_1)$ (e.g. colored image samples depends on edge map) while for unconditional generation tasks we consider $p(x_0, x_1)=p(x_0)p(x_1)$. In both cases, we construct VP or VE bridges (coupling derived from underlying VP and VE processes), and we find both cases work well in practice for translation and unconditional generation.
>
>
>
> > Questions on sampler used in experiments.
>
> We present the ablation on our sampler (and preconditioning effect) in Table 3. We do not retrain our model for sampler ablation studies. We observe that our higher-order sampler performs better than the EDM ODE integrator out-of-the-box.
>
> When comparing with other methods, our motivation is to “strongman” the baselines. In particular, we need to compare against the best configuration for all models because the practical design choices have been key for empirical success. In particular, these extend beyond the loss function and SDE/ODE formulations.
>
> We therefore use our best configuration to compare against the best of other methods on standard benchmarks, and we present additional ablation studies on these design choices to see the effect on our model’s performance. We also note that it is also not always feasible to exactly compare only the methods. For example, OT-based methods like Rectified Flow for flow-based methods do not work well with the additional stochasticity introduced in our sampler, and even our baseline ODE-sampler does not work well with such methods because of the special time-discretization. We have tried applying this to Rectiflied Flow but fails completely in generating quality images. Therefore, reusing some designs for baselines may be detrimental to the baseline methods.
>
> [*] Liu et al., 2022, Let us Build Bridges: Understanding and Extending Diffusion Generative Models

---

> > ### Comment · Reviewer_H8Nr · 2023-11-21
> > **Thanks for your response, some clarifications needed.**
> >
> > Thanks to the authors for their thorough response back. There are some things that I find unsatisfactory about the responses, and hopefully you have some time to clarify it.
> >
> >  - It seems that your characterization of some alternative works is not befitting, especially with regards to the very thorough public comments made by Peluchetti. Indeed, it is unclear to the reviewer, as the public comment says, what statements like "our model is equally valid" means in the context of solving the measure transport problem, when the public comment shows that the proposed method does not do that.
> >
> > - To the reviewer, it is really poor form to just stuff citations in the supplementary material. The purpose of a submission is to present work *in the context of the literature to add to our shared knowledge*. While I appreciate you taking the time to assemble a thorough related work, it is not a bad thing for your work to be similar to others, indeed it can be a *good* thing! It means multiple perspectives arising to similar conclusions, which is what we hope to do (I think!). As such, as a reviewer, I don't aim to suggest you need to differentiate your work as better or above other perspectives. Just contextualize it :)
> >
> > - The characterization I think of [4] is incorrect -- indeed, that paper seems be showing that you don't need any special coupling to build a bridge, as you are requiring here. Moreover, it can be done with arbitrary linear or nonlinear processes. As Peluchetti pointed out, needing Doob-h and conditioning the score model on x_t does not seem to engender flexibility. What do you mean when you say this allows for a more flexible model?
> >
> > - This work [4] does not specify anything about relying on an ODE, as you've written in the additional supplemental material. In fact, it seems to be saying that both the SDE and the ODE are available from the same model, with the learning being totally independent of choice. What does it mean to say: "Different from these methods, our model is SDE-based, directly builds from diffusion models, and uses a different denoising bridge score-matching loss than this class of models."? All of these works build off the notion of diffusion, however the reviewer doesn't understand why this is a blessing -- indeed, it's why Doob-h and such complicate the validity of the flow map, no?
> >
> > - I believe that this paper and the Peluchetti's discussion need to be properly addressed in the text. Particularly Peluchetti's derivation regarding the solution to the transport equation. If you can do that, I will bump my score, but there seems to be some technical inconsistencies in the work, and to be honest I don't understand some of the language justifying the method.
> >
> >
> > - With regards to the remark about mis-interpreting OT: I am a bit confused by the response, as $x = (1-t)x_0 + t x_1$ is not a coupling, that's a function. A coupling would be how you draw $x_0, x_1 \sim p(x_0, x_1)$. If you had the OT, the function you wrote above, which does interpolation, would be $x = (1-t)x_0 + t*T(x_0)$ where $T$ is the OT map such that $T(x_0) = x_1$.
> >
> >
> > - Finally, the remark about experiments. You write that "For example, OT-based methods like Rectified Flow for flow-based methods do not work well with the additional stochasticity introduced in our sampler." This confuses me as well. Whatever sampler you use, you need to show that the objects in that sampler actually correspond to the objects you learn. If you learned the solution to a probability flow, e.g. a velocity field $v(x,t)$ that gives the map $\dot X_t(x) = v(t, X_t(x))$ with $X_0(x) = x$, then you can't go ahead and plug this into an SDE, you'd need a different vector field, for example, the one presented in [4], equation 2.33.
> >
> >
> > I cannot raise my score until some of the above is addressed. While the supplemental related work is thorough and greatly appreciate the author's efforts to do this, there are some remaining inconsistencies that I think do not merit publication yet (but are some which I think are potentially addressable!). Thanks again for your efforts and the thoroughness of your response.

---

> ### Author Response · Authors · 2023-11-22
> **Additional response (Part 1)**
>
> We thank the reviewer again for the patience and suggestions. We want to address the remaining concerns.
>
>
> > It seems that your characterization of some alternative works is not befitting, especially with regards to the very thorough public comments made by Peluchetti. Indeed, it is unclear to the reviewer, as the public comment says, what statements like "our model is equally valid" means in the context of solving the measure transport problem, when the public comment shows that the proposed method does not do that.
>
> We have additionally clarified that in the case of noiseless limit, the magnitude of the score $\nabla_{x_t} \log q(x_t | x_T)$ (Eq. 10) goes to infinity because $\sigma_t \rightarrow 0$, so we can no longer match this term directly with score-matching. Although the presented score-matching model cannot handle the noiseless limit case, we can design new losses to predict the combined $-g^2(t)(1/2 s(x_t,t,y,T) - h(x_t,t,y,T))$, which we’ve shown in the appendix reduces to $x_T - x_0$ (special case 2). Our framework subsumes this in the sense that the ODE drift terms becomes the same terms, e.g.  $x_T - x_0$. In addition, thanks to the generalized parameterization of networks in Section 4, we can design our score model to be $s(x_t, x_T, t; \theta) = A x_t +B  x_T + C V(x_t, t; \theta)$ where $V$ is the neural network with parameter $\theta$ and simply choose $A=B=0$, $C=1$ and match it against $x_T - x_0$ with loss $\mathbb{E}_{x_t,t,x_0,x_T}[ \lVert s(x_t,x_T, t; \theta) - ( x_T - x_0) \rVert^2] $.
>
> The above to loss then reduces to $\mathbb{E}_{x_t,t,x_0,x_T}[\lVert V(x_t, t; \theta) -  (x_T - x_0) \rVert^2]$
>
> where $s$ and $V$ are parameterized by $\theta$ and the loss becomes loss of OT-Flow-Matching and Rectified Flow.
>
>
>
>
> > To the reviewer, it is really poor form to just stuff citations in the supplementary material. The purpose of a submission is to present work in the context of the literature to add to our shared knowledge. While I appreciate you taking the time to assemble a thorough related work, it is not a bad thing for your work to be similar to others, indeed it can be a good thing! It means multiple perspectives arising to similar conclusions, which is what we hope to do (I think!). As such, as a reviewer, I don't aim to suggest you need to differentiate your work as better or above other perspectives. Just contextualize it :)
>
> Unfortunately, due to ICLR rules, we are only allowed to submit a 9 page paper (increasing this would otherwise disqualify our paper). Furthermore, we are also running out of time to address all reviewer comments due to the shortened discussion period (where we can update the main pdf) and conflicting deadlines, meaning that it would be extremely untenable for us to cut the main manuscript to fit in our enhanced discussion. We included the discussion in the appendix to give us the freedom to write down all of our thoughts while also avoiding damaging the flow of the main paper. We apologize if this came off as “stuffing” citations in the supplementary material; our goal was simply to find space to address all reviewer concerns about prior work before fully integrating the enhanced discussion.
>
> Generally, we agree that our work fits into this (extremely) broad field of diffusion models with bridge additions. In particular, we agree that most of our developed theory can be (relatively easily) rederived from existing frameworks, as most papers in this field build off of the same few constructions. In fact, this is actually similar to how Schrodinger Bridge and Entropic Optimal Transport are fundamentally the same thing despite their different formulations. However, the difference with our paper (that we must stress) is the ability to connect with the existing diffusion model literature, which is necessary for achieving good empirical results. As such, it’s worth emphasizing why the existing literature has largely been unable to scale to these types of problems that we consider in our paper.

---

> ### Author Response · Authors · 2023-11-22
> **Additional response (Part 2)**
>
> > The characterization I think of [4] is incorrect -- indeed, that paper seems be showing that you don't need any special coupling to build a bridge, as you are requiring here. Moreover, it can be done with arbitrary linear or nonlinear processes. As Peluchetti pointed out, needing Doob-h and conditioning the score model on x_t does not seem to engender flexibility. What do you mean when you say this allows for a more flexible model?
>
> We don’t believe that we stated that our model is more flexible than [4] (if we did so, then we apologize for the miscommunication). In particular, as we mention explicitly in our subsequent response point, our more focused development is more critical for our goal (of extending existing diffusion model frameworks).
>
> It is also perhaps not fitting to talk about [3] and [4] together in the same sentence as [3] because [3] uses interpolants for constructing normalizing flow, which gives deterministic dynamics. We have put them in a separate sentence and we stress that we do not claim our framework is more flexible.
>
> > This work [4] does not specify anything about relying on an ODE, as you've written in the additional supplemental material. In fact, it seems to be saying that both the SDE and the ODE are available from the same model, with the learning being totally independent of choice. What does it mean to say: "Different from these methods, our model is SDE-based, directly builds from diffusion models, and uses a different denoising bridge score-matching loss than this class of models."? All of these works build off the notion of diffusion, however the reviewer doesn't understand why this is a blessing -- indeed, it's why Doob-h and such complicate the validity of the flow map, no?
>
>
> The reviewer makes a good point that works like [4] and our work (and generally many of the works since the advent of ScoreSDE) build off of the core fundamentals of the original diffusion models papers (with perhaps some modifications like the Doob-h transform, presentation, etc…). Our statement was primarily meant to characterize our work as particularly suitable with the empirical frameworks developed for diffusion models (DDPM, EDM, VDM). In particular, by directly introducing our framework with SDEs, a connected score matching loss, and similar noise schedules as existing work, we can better adapt ubiquitous design choices. However, for a work like [4], although there is undoubtedly a much larger degree of generalizability (ie arbitrary interpolation strategies between arbitrary distributions), these additional mathematical formulations make it harder to see the connections with the existing literature and, as such, adapt the existing empirical design choices.
>
>
>
> > Remaining confusion on mis-interpreting OT.
>
> We want to emphasize that we are simply borrowing the terminology from Flow Matching. Our method does not solve OT, but we are somewhat compelled to reuse this terminology as the naming of Flow Matching “OT” path is extremely popular. Generally, Flow Matching calls this the “OT” path as it is the optimal transport path between a dirac delta distribution and a Gaussian, even though the full probability path is not OT.
>
>
> > The remark about using proper samplers for different methods.
>
> The additional stochasticity example may be inapplicable in the context of Rectified Flow. Our point was that each sampler is tailored to a specific model. For example, diffusion model samplers [A,B,C] cannot be directly used with flow models as diffusion samplers heavily rely on model-specific definitions such as log SNR, noise schedule, ODE step-schedule, etc. and are not general-purpose integrators. Even though many are ODE-based, it would require significant redesign to work on other flow-based models. We therefore refrain from doing so.
>
>
>
>
> [A] Lu, Cheng, et al. "Dpm-solver: A fast ode solver for diffusion probabilistic model sampling in around 10 steps."
>
> [B] Lu, Cheng, et al. "Dpm-solver++: Fast solver for guided sampling of diffusion probabilistic models."
>
> [C] Karras, Tero, et al. "Elucidating the design space of diffusion-based generative models."

---

> > ### Comment · Reviewer_H8Nr · 2023-11-22
> > **Final clarification**
> >
> > Ok, thanks again folks. I'm satisfied with your response regarding Peluchetti's comments.
> >
> >  However, I really think it is valuable to not just propagate misconceptions because they are popular (regarding OT). Choosing $x_1 - x_0$ as the conditional learning problem has almost *nothing* to do with OT, unless as you say you reduce it to the tautological case of a point mass to a Gaussian...I think we should all strive (for posterity's sake) to make accurate statements in our work.
> >
> > > The reviewer makes a good point that works like [4] and our work (and generally many of the works since the advent of ScoreSDE) build off of the core fundamentals of the original diffusion models papers (with perhaps some modifications like the Doob-h transform, presentation, etc…). Our statement was primarily meant to characterize our work as particularly suitable with the empirical frameworks developed for diffusion models (DDPM, EDM, VDM). In particular, by directly introducing our framework with SDEs, a connected score matching loss, and similar noise schedules as existing work, we can better adapt ubiquitous design choices. However, for a work like [4], although there is undoubtedly a much larger degree of generalizability (ie arbitrary interpolation strategies between arbitrary distributions), these additional mathematical formulations make it harder to see the connections with the existing literature and, as such, adapt the existing empirical design choices.
> >
> > This wasn't really my point. My point was that the connection between the two densities *need not rely on SDEs at all* and by forcing it to you have to tie your sampling to your noise schedule in these ways. But I understand wanting the continuity of readership etc. In reality the noise schedule stuff is actually totally independent of the learning problem, but I understand the continuation in ideas.
> >
> > The real clarification I was trying to make with [4] is now elucidated in your remark about samplers:
> >
> > >  Our point was that each sampler is tailored to a specific model. For example, diffusion model samplers [A,B,C] cannot be directly used with flow models as diffusion samplers heavily rely on model-specific definitions such as log SNR, noise schedule, ODE step-schedule, etc. and are not general-purpose integrators. Even though many are ODE-based, it would require significant redesign to work on other flow-based models. We therefore refrain from doing so.
> >
> > This is not completely true, and is the point I was making about the takeaway from [4]. Indeed, the diffusion coefficient in the stochastic sampling need not be related to the process connecting the densities.
> >
> > Thank you for being available for responses. I will raise my score to a 6. I think there are some broad re-writings of the exposition here that could be useful, but you've been thorough in your engagement and have clarified a number of points, particularly wrt to the public comments..

---

> > > ### Author Response · Authors · 2023-11-22
> > > **Thank you for your review**
> > >
> > > Thank you for your review and insights. We will take these into consideration when updating our final draft.

---

### Official Review · Reviewer_zUzA · 2023-11-01

**Soundness:** 4 excellent
**Presentation:** 4 excellent
**Contribution:** 4 excellent
**Rating:** 8
**Confidence:** 4

**Summary:**

This paper proposed a novel formulation on diffusion-based generative model, i.e., denoising diffusion bridge models (DDBMs). This formulation further inspire applications in important tasks such as image-to-image translation. Detailed theoretic derivation and empirical results are provided.

**Strengths:**

1. The proposed method and formulation is novel and can potentially empower some pivotal applications such as image-to-image translation.
2. The paper is well-written and easy-to-follow.
3. Sufficient qualitative and quantitative results are provided to validate the effectiveness of the proposed method.

**Weaknesses:**

1. More results on larger-scale generation would be better (e.g., DDBM with Stable Diffusion).
2. It would be better if the author could compare the DDBMs with some more advanced image-to-image translation algorithm such as controlnet.

**Questions:**

1. Is there any advantage of DDBMs over some previous image-to-image translation methods such as controlnet?
2. It would be better to show the variation ability of DDBMs in terms of the generation results. For example, when translating edges into an image, there exists various solutions. Are DDBMs able to generate these various results?

---

> ### Author Response · Authors · 2023-11-19
> **Thank you for your review and appreciation.**
>
> We thank the reviewer for your appreciation of our work. We hope the following answers address your concerns.
>
> > Is there any advantage of DDBMs over some previous image-to-image translation methods such as controlnet?
>
> We want to note that our framework is orthogonal to ControlNet as ControlNet is primarily an architectural design. Our framework is theoretically different from (and based on) diffusion models, and we can also use ControlNet with pretrained diffusion weights as our starting point for training our bridge framework. One of the motivations for our work is to develop a bridge-based framework such that it is easy to transfer many successful design choices over to improve the empirical results of bridge-based frameworks, which have been met with limited empirical success. We have experimented with those from EDM, but ControlNet is certainly another choice one can adopt. However, due to limited resources and time we did not proceed to larger-scale experiments with ControlNet.
>
> > It would be better to show the variation ability of DDBMs in terms of the generation results. For example, when translating edges into an image, there exists various solutions. Are DDBMs able to generate these various results?
>
> Yes indeed. This is one of the motivations for introducing stochasticity to the bridge sampler, as the conditional generation is multimodal. We indeed observe some variations in image translation when we set different seeds, and this variation comes from the introduced stochasticity in our hybrid sampler. We show some variations in the additional pdf page in the supplementary.

---

> > ### Comment · Reviewer_zUzA · 2023-11-23
> >
> > Thanks for your reply. These have resolved my concerns, and I will keep my score of 8.

---

### Public Comment · ~Stefano_Peluchetti1 · 2023-11-12
**Proper Contextualization of Contributions [part 1...]**

I read with interest this work, as it relates to some research I am carrying out.
While the paper contains various interesting findings, I found that the presentation does not properly contextualize its contributions, which prompted me to write this comment.

## Proposed Method

Firstly, I detail my own understanding, in case I am mistaken.
The paper starts with SDE (1) with distribution $P$.
Consider as given a joint distribution $Q_{0,T}$, which is denoted with $q_\text{data}$ in this work, with marginals $Q_0$, $Q_T$.

Conditioning (1) on both endpoints gives the diffusion bridge with distribution $P_{•|0,T}$ (this notation means "everything else given the values at times 0 and T").
Note that (5) instead describes such diffusion bridge with an initial distribution $Q_0$, or equivalently a mixture of diffusion bridges **over their initial endpoint** according to $Q_0$.

Remarks:

1. there seems to be a typo: $Q_{0|T}$ is the initial distribution for (5);
2. the term "diffusion bridge" refers only to the case where both endpoints of a given diffusion are pinned down, not to (5) as stated in the text;
3. the law of this process can be denoted with $Q_{•|T}$ (this is not clearly stated in the text).

Thus, the time reversal of (5) yields a diffusion starting from $x_T = y$ and yielding a sample from $Q_{0|T}$ as terminal value, whose evolution is described by (6), while (7) is meant to represent a marginal-matching ODE for (6) (more on this later).
Finally, (9) provides the (conditional) score matching estimator for $∇_{x_t}\log q(x_t|x_T)$.

To recap: start with (1), compute the diffusion bridge, mix the diffusion bridge initial endpoint over $Q_{0|T}$, perform the time reversal, simulate from the (learned via (9)) reverse-time SDE (6) or ODE (7) started from a sample $x_T ∼ Q_T$.

## Alternative Construction and Prior Works

There is no need for time reversal arguments.
Given SDE (1), the corresponding diffusion bridge from $x$ to $y$ follows (5) with $x_0 = x, x_T = y$ (for instance, Theorem 7.11 of [Särkkä2019]).
We can then construct the mixture of these diffusion bridges **over their terminal endpoint** (instead of the initial one) requiring $x_T ∼ Q_{T|0}$.
This is still a diffusion, a standard result (see Theorem 8.4.3 of [Øksendal2013] or Theorem 7.12 of [Lister1977]) gives its representation:

$$
x_0 = x, \quad dx_t = \Big[f(x_t,t)+g(t)^2𝔼_{x_T ∼ Q_{T|t,0},x_t ∼ p_{t|0,T}}\big[∇_{x_t}\log p_{T|t}(x_T|x_t)\ |\ x_t,x_0\big]\Big]dt+g(t)dw_t.\tag{$F$}
$$

Then ($F$) and (6) are equivalent, after accounting for the opposite time ordering, i.e. swapping $g(T - t)$ for $g(t)$, $Q_{T,0}$ for $Q_{0,T}$.
Equivalently, ($F$) can also be directly derived from (1) through Doob's h-transform machinery, as diffusion bridges are.
The process ($F$), mapping a point to a terminal distribution via a diffusion, is sometimes referred to as Schrödinger-Föllmer process, and has been extensively explored in the literature in various contexts, see [Wang2021] for a generative application.

The dynamics of ($F$) makes it clear why it is not possible to define a probability-flow ODE matching the marginal distributions of ($F$) on $[0,T]$: in this hypothetical ODE, i.e. (7), both the initial condition and the dynamics would be deterministic, with no randomness left.
The mentioned "averaged out" results correspond to the evolution of this deterministic ODE.

The process specified by ($F$) is also a specific instance of the DBM (Diffusion Bridge Mixture Matching) transport of [Peluchetti2021] obtained by: (i) constructing a (non diffusion) process by mixing diffusion bridges over **both initial and terminal endpoints**; (ii) matching the marginal distributions of this process with a diffusion process.
When the mixing on the initial endpoint puts all the mass to a single value $x_0$, ($F$) is recovered.
This specific instance of the DBM transport is discussed in Sections 3.2 and 6 of [Peluchetti2021].

Indeed, the training objective presented in Equation (24) of [Peluchetti2021] is, conforming to the notation here employed,

$$
𝔼_{t ∼ 𝒰(0,T),x_T ∼ Q_{T|0},x_t ∼ p_{t|0,T}}[‖∇_{x_t}\log p_{t|0,T}(x_t|x_0,x_T) - s_θ(X_t,t)‖^2],
$$

where $s_θ(⋅)$ is a neural network with sufficient representation power.
Unless I misunderstood the present work, this is identical to the training objective (9) for a given $x_0$ (denoted by $x_T$ in (9) due to opposite time ordering).
In order to learn $s_θ(x_t,t)$ for multiple values of $x_0$, $x_0$ needs to be included as argument to the neural network, $s_θ(x_t,t,x_0)$, and needs to be sampled as well from a distribution which matches the support of $Q_0$ (as it is done for $t$).
Taking $Q_0$ itself for simplicity yields the same objective as (9) (accounting for the opposite time ordering).

---

> ### Public Comment · ~Stefano_Peluchetti1 · 2023-11-12
> **Proper Contextualization of Contributions [part 2...]**
>
> The learning goal is to estimate the conditional expectation of $(F)$, i.e.
>
> $$
> A(x_t,t,x_0) ≔ 𝔼_{x_T ∼ Q_{T|t,0}}\big[∇_{x_t}\log p_{T|t}(x_T|x_t)\ |\ x_t,x_0\big].\tag{$A$}
> $$
>
> Equation (23) of [Peluchetti2021] establishes that $A(x_t,t,x_0) = s_{θ^*}(x_t,t,x_0) - ∇_{x_t}\log p_{t|0}(x_t|x_0)$, which reads the same as the term $s(x_t,t,y,T) − h(x_t,t,y,T)$ in (6) (again, accounting for the opposite time ordering), making the equivalence of (6) and ($F$) explicit.
>
> It is noteworthy that simpler objectives can be derived, see Equation (25) of [Peluchett2021].
> Indeed, ($A$) immediately ("conditional expectations minimize MSE") yields the regression objective
>
> $$
> A(x_t,t,x_0) = \arg\min_{α_θ(⋅)}𝔼_{t ∼ 𝒰(0,T),x_T ∼ Q_{T|0},x_t ∼ p_{t|0,T}}[‖∇_{x_t}\log p_{T|t}(x_T|x_t) - α_θ(x_t,t,x_0)‖^2],
> $$
>
> for a neural network $α_θ(x_t,t,x_0)$, and its generalization over multiple $x_0$
>
> $$
> A(x_t,t,x_0) = \arg\min_{α_θ(⋅)}𝔼_{t ∼ 𝒰(0,T),(x_0,x_T) ∼ Q_{0,T},x_t ∼ p_{t|0,T}}[‖∇_{x_t}\log p_{T|t}(x_T|x_t) - α_θ(x_t,t,x_0)‖^2].\tag{$O_F$}
> $$
>
> Finally, we note that the generic DBM transport corresponds to "removing the conditioning on $x_0$" from ($F$) and ($O_F$):
>
> $$
> x_0 ∼ Q_0, \quad dx_t =\Big[f(x_t,t)+g(t)^2A(x_t,t)\Big]dt+g(t)dw_t,\tag{$M$}
> $$
>
> and
>
> $$
> A(x_t,t) = \arg\min_{α_θ(⋅)}𝔼_{t ∼ 𝒰(0,T),(x_0,x_T) ∼ Q_{0,T},x_t ∼ p_{t|0,T}}[‖∇_{x_t}\log p_{T|t}(x_T|x_t) - α_θ(x_t,t)‖^2].\tag{$O_M$}
> $$
>
> The difference between ($O_F$) and ($O_M$) is that is the neural network is not taking $x_0$ as input.
> For unconditional generation, such as the experiment of Section 7.3 of this work, there seems to be no advantage in employing ($F$) instead of ($M$): (i) we can obtain the probability-flow ODE matching the marginal distributions of ($M$) as $x_0$ has full support; (ii) in this setting ($F$) forces $x_T$ to be independent of $x_0$, while in ($M$) a dependency is learned that corresponds to a crude approximation to the Schrödinger-Bridge, resulting in a "simpler drift".
> See [Peluchetti2023], which also carries out an application of the DBM transport ($M$) to generative modeling in Section 5.3.
>
> In conclusion, on the methodological side:
> - this work employs the "point to distribution" diffusion process, which been studied by prior works for unconditional generative applications (which is derived differently, through reverse-time arguments)
> - in particular, the proposed training objective and dynamics bear close relationship to the corresponding quantities for a specific instance of the DBM transport of [Peluchetti2021]
> - unless I missed prior relevant works, the novelty is in the application of this transport to conditional modeling of the data distribution when alignment (paired data) is available, that is by employing the transport over multiple initial values $x_0$
> - ODE (7) is not a probability flow ODE matching the marginal distribution of SDE (6)
>
> In view of the above considerations, and of the Related Works section below, the claims
>
> > We consider a reverse-time perspective of diffusion bridges, ..., and use this perspective to establish a general framework for distribution translation. We then note that this framework subsumes existing generative modeling paradigms such as score matching diffusion models (Song et al., 2020b) and flow matching optimal transport paths (Albergo and Vanden-Eijnden, 2023; Lipman et al., 2023; Liu et al., 2022a).
>
> > In contrast, our method ... can theoretically subsume many previous methods such as diffusion models (Song et al., 2020b), OT-Flow-Matching (Lipman et al., 2023), and Rectified Flow (Liu et al., 2022a), as illustrated below.
>
> do not appear to be correct nor supported.
>
> As previously stated, the idea of applying the "point to distribution" diffusion for conditional modeling is (to the best of my knowledge) novel and has merit.
> Moreover, this work presents additional contributions in Sections 4, 5 and 7, deriving an improved sampler, adapting many recent advancements in SDE-based generative modeling, and carrying out an extensive empirical assessment of the proposed approach for various applications.
> These are all interesting and valid contributions on their own, and are required to demonstrate competitive results in generative visual applications.

---

> > ### Public Comment · ~Stefano_Peluchetti1 · 2023-11-12
> > **Proper Contextualization of Contributions [part 3]**
> >
> > ## Related Works
> >
> > It seems a bit ungenerous to cite [Peluchetti2021] for "diffusion bridges".
> >
> > > A recent work (Liu et al., 2023) considers a special case of SB and is able to reduce the algorithm complexity significantly. Nevertheless, its connection between the proposed SB formulation and prior methodologies, such as unconditional diffusion models (Ho et al., 2020) and Flow-Matching (Lipman et al., 2023; Tong et al., 2023), remains unclear and requires further elucidation.
> >
> > The proposal of [Liu2023] is almost identical to the DBM transport or [Peluchetti2021].
> > Inspecting Algorithms 1 and 2 (training and sampling) of [Peluchetti2021] for the case of a scaled Brownian motion reference process, i.e. Equation (11) with $Γ = I$, and Algorithms 1 and 2 (same) of [Liu2023], where the loss is defined by Equation (12) and the dynamics by Equation (11), reveals that the differences are confined to the choice of the discretization scheme used to sample the learned SDE, and to the choice of the target for the loss computation (noise, vs denoised terminal sample, vs…), both of which are free choices in SDE-based generative models.
> >
> > More in general, the proposals of [Liu2023], [Somnath2023], of flow-matching (conditional and not), and of additional works, are shown to be equivalent under certain conditions to DBM in Appendix A.1 of [Shi2023], where the DBM approach is referred to as Bridge Matching.
> > I do not wish to detract in any way from the otherwise excellent contributions of these works, which include exhaustive empirical validations and additional theoretical findings and insights due to the introduced modeling frameworks.
> > But it is now well understood that these approaches are closely related.
> > They can be seen as a crude approximation to the Schrödinger-Bridge: [Peluchetti2023] and [Shi2023] show that the iterated application of the DBM transport converges to the Schrödinger-Bridge.
> >
> > ## References:
> >
> > - [Särkkä2019] Applied Stochastic Differential Equations
> > - [Øksendal2013] Stochastic Differential Equations: An Introduction with Applications
> > - [Lister1977] Statistics of random processes: General theory
> > - [Wang2021] Deep Generative Learning via Schrödinger Bridge
> > - [Peluchetti2021] Non-Denoising Forward-Time Diffusions
> > - [Peluchetti2023] Diffusion Bridge Mixture Transports, Schrödinger Bridge Problems and Generative Modeling
> > - [Liu2023] $\mathrm{I}^2$ SB: Image-to-Image Schrödinger Bridge
> > - [Somnath2023] Aligned Diffusion Schrödinger Bridges
> > - [Shi2023] Diffusion Schrödinger Bridge Matching

---

> > > ### Author Response · Authors · 2023-11-19
> > >
> > > Thank you for mentioning these works. Other reviewers had similar concerns regarding noting previous works. We have included a reworked related works section in the supplementary with more detailed discussion.

---

> ### Author Response · Authors · 2023-11-19
>
> Thank you for the insightful comments. For introduction of diffusion bridge, our main point is that Doob’s h-transform allows us to bring any point to the given terminal y. We have modified the text to address that for diffusion bridge we require pinned starting point. As for the need of time-reversal, you do not need it in theory, but a major motivation for our work is that we can directly reuse successful design choices from diffusion models to adapt to the bridge framework for good empirical performance. Reformulating the problem without time-reversal makes it difficult to achieve this goal.
>
> For subsuming OT-Flow-Matching and Rectified Flow, we show that when we use VE bridge formulation and take noise to 0 limit, the drift for probability-flow ODE $f(x_t, t) - g^2(t)(\frac{1}{2}\nabla_{x_t}\log p(x_t | x_0, x_1)-h(x_t, t, x_T, T))$ of these works reduce to the straight velocity $x_T - x_0$. It’s the matter of reparametrization for the network to predict these two terms together, which is the case of OT-Flow-Matching and Rectified Flow. The SDE we proposed then becomes the corresponding transport plan.

---

> ### Public Comment · ~Stefano_Peluchetti1 · 2023-11-21
> **Not addressed points**
>
> > As for the need of time-reversal, you do not need it in theory, but a major motivation for our work is that we can directly reuse successful design choices from diffusion models to adapt to the bridge framework for good empirical performance. Reformulating the problem without time-reversal makes it difficult to achieve this goal.
>
> It would be helpful if the Authors could explain in detail what aspects of the forward-time construction make it difficult to reuse design choices from diffusion models.
>
> Reversing the time index, I have shown in my previous comments how the proposed SDE (6) is identical, term-by-term, to the SDE of Equation (8) with Equation (23) plugged in from [Peluchetti2021].
> In the same way, the training objective (9) is also identical to the training objective of Equation (24) from [Peluchetti2021], provided that $x_0$ is sampled as well, in which case the neural network needs to take as input $x_0$.
>
> In practice, with a reverse time construction as proposed here, $x_T$ follows the source data distribution and $x_0$ follows the target data distribution.
> With a forward construction, $x_0$ follows the source data distribution and $x_T$ follows the target data distribution.
> For time-dependent drift $f(x, t)$ and diffusion $g(t)$ coefficients in the reverse time construction (diffusion models), $f(x, T - t)$ and $g(T - t)$ can be employed in the forward time construction.
> Then the resulting algorithms are identical, for training and sampling.
> I struggle to see where mentioned the mentioned difficulty is: it cannot possibly be the need to compute $T - t$.
>
> If anything, the forward time construction makes it clear that ODE (7) is not a probability flow ODE matching (the marginal distributions of) SDE (6), which might not appear obvious in the reverse time construction.
>
> > For subsuming OT-Flow-Matching and Rectified Flow, we show that when we use VE bridge formulation and take noise to 0 limit, the drift for probability-flow ODE $f(x_t,t) - g^2(t)(\frac{1}{2}∇_{x_t}\log p(x_t|x_0,x_1) - h(x_t,t,x_T,T))$ of these works reduce to the straight velocity. It’s the matter of reparametrization for the network to predict these two terms together, which is the case of OT-Flow-Matching and Rectified Flow. The SDE we proposed then becomes the corresponding transport plan.
>
> This is not correct, the proposed method does not subsume OT-Flow-Matching and Rectified Flow.
> The limiting case of vanishing noise is already discussed carefully in [Peluchetti2023] and [Shi2023]: the noiseless case ($σ = 0$) of the DBM transport with reference SDE $dx_t = σdw_t$ gives the Rectified Flow iteration.
> However, the method proposed here also conditions on $x_0$ (forward construction), and the resulting transport is different.
>
> Example: $Q_0 = Q_1 = \mathrm{Uniform}(-1/2,1/2)$, $Q_{0,1} = Q_0 Q_1$ (independent coupling).
> Then the Rectified Flow drift satisfies $μ_\mathrm{RF}(x_t=0, t=1/2) = 0$.
> The drift of the proposed method satisfies $μ(x_t=0, t=1/2, x_0=-1/2) = 1$, $μ(x_t=0, t=1/2, x_0=0) = 0$, $μ(x_t=0, t=1/2, x_0=1/2) = -1$ (no other $x_0$ can result in $x_t = 0$ at $t=1/2$).
>
> From the additional material:
>
> > One additional caveat is that our framework as presented needs to take in $x_T$ as an additional condition. This means the resulting reverse SDE is not Markovian. However, we note that this design choice is there to help translation given the condition. Our model is equally valid if we do not take in $x_T$ as additional input and it is a matter of practical design choice for the network to predict $x_T − x_0$ together instead of matching $∇_{x_t} \log q(x_t | x_0, x_T )$ only, which we find helpful for image translation. This exactly reduces to that of OT-Flow-Matching and Rectified Flow. The corresponding probability-flow ODE also becomes the corresponding transport map.
>
> It is not clear what "our model is equally valid" means.
> When the neural network $s(⋅)$ entering (9) does not take $x_T$ as input, $s(x_t,t)$ learns $∇_{x_t}\log q(x_t)$, where $q(x_t) = ∫∫q(x_t|x_0,x_T)q_\mathrm{data}(x_0,x_T)dx_0dx_T$.
> At a quick glance, the resulting SDE (6), from $t=T$ to $t=0$, does not in general result in $x_0 ∼ q_\mathrm{data}(x)$.
> Otherwise, a proof is warranted.
>
> > Our method instead shows how to construct a bridge model from any existing VP and VE diffusion processes in continuous time, and Brownian Bridge (as considered in most previous works) is but a special case of VE bridges.
>
> In [Peluchetti2021] both VE and VP parametrization are considered, see Equations (11) and (12) in Section 4.
> The formulations are slightly more general as they cover non-diagonal diffusion coefficients as well.
>
> > In contrast, other works (Somnath et al., 2023; Peluchetti), while also adopting Doob’s h-transform, propose simulation-free algorithms for efficient sampling of intermediate variables.
>
> Surely, given all the the similarities between [Peluchetti2021] and the current proposal a different kind of acknowledgement could be employed.

---

> > ### Author Response · Authors · 2023-11-22
> > **Additional response (Part 1)**
> >
> > We would like to thank the commenter for the constructive criticisms.
> >
> > Unfortunately, many of the points are incorrect and are broad overclaims of the commenter’s previous work. As the commenter mentioned, many of these conclusions were drawn from just “a quick glance”. We would like to kindly ask the commenter to refrain from rushing to hasty judgments without a proper check, as, although commenters do not have the same obligations to correctness as reviewers, their words are often weighted similarly.
> >
> >
> > > It would be helpful if the authors could explain in detail what aspects of the forward-time construction make it difficult to reuse design choices from diffusion models.
> >
> >
> > We stress that most of the previous implementations of diffusion models rely on the noise-prediction model proposed by DDPM and later, v-prediction, as proposed by [7], which does not suffer from exploding magnitude issues as the forward time construction for Doob’s h-function, whose magnitude goes to infinity as time increases. As a result, the losses $O_F, O_M$ are likely problematic (empirically) because they do not take this into account.
> >
> > Many well-known prior works [1,2,3] in diffusion also define the noise schedule based on (log) SNR instead of the $f(x_t,t)$ and $g(t)$ directly. This makes forward-time construction difficult to reuse these choices. However, we have given clear connection with these definitions through both Table 1 and the marginal sampling probabilities.
> >
> > Higher-order samplers from previous works [2,4,5] are also derived using and heavily rely on the (log) SNR construction of reverse-time diffusion, which does not directly translate to forward-time construction with Doob’s h-transform.
> >
> >
> > We also emphasize that our proposed probability flow ODE (which is needed to connect with methods such as EDM) comes about from our reverse-time perspective and, as the commenter mentioned, is not obvious from the forward time construction. (Note that we disagree with the commenter’s characterization of this ODE as incorrect; it is correct since it is more technically involved than the commenter has made it out to be).
> >
> >
> > From our point of view, the connections with prior diffusion formulations are rather obfuscated as it is not directly clear what connection the learned “A” function has with the “noise-prediction” or “v-prediction” framework of most existing diffusion works (e.g. DDPM, EDM, VDM). Even if it is possible to use diffusion model design choices for the forward-time construction, it is dismissive to state that there is no “difficulty” in doing so or that “there is no need for time-reversal arguments”, as such a connection was not made in the 2 years since the submission of [Peluchetti2023].

---

> > ### Author Response · Authors · 2023-11-22
> > **Additional response (Part 2)**
> >
> > > Reversing the time index, I have shown in my previous comments how the proposed SDE (6) is identical, term-by-term, to the SDE of Equation (8) with Equation (23) plugged in from [Peluchetti2021].
> >
> > We agree that the SDEs are fundamentally the same thing. This is natural, since both constructions are based off of Doob’s h-transform. That being said, we have copied over our equation
> >
> > $d \mathbf{x}_t=\left[\mathbf{f}\left(\mathbf{x}_t, t\right)-g^2(t)\left(\mathbf{s}\left(\mathbf{x}_t, t, y, T\right)-\mathbf{h}\left(\mathbf{x}_t, t, y, T\right)\right)\right] d t+g(t) d \hat{\mathbf{w}}_t$
> >
> > $\mathbf{s}(x, t, y, T)=\nabla_{\mathbf{x}_t} \log q(\mathbf{x}_t \mid \mathbf{x}_T) \quad \text{where}  \quad \mathbf{x}_t=x, \mathbf{x}_T=y$
> >
> > $\mathbf{h}(x, t, y, T)= \nabla_{\mathbf{x}_t} \log p\left(\mathbf{x}_T \mid \mathbf{x}_t\right) \quad  \text{where}  \quad \mathbf{x}_t=x, \mathbf{x}_T=y$
> >
> > $q\left(\mathbf{x}_t \mid \mathbf{x}_0, \mathbf{x}_T\right)  =\mathcal{N}\left(\hat{\mu}_t, \hat{\sigma}_t^2 \boldsymbol{I}\right), \quad \text { where } $
> >
> > $\hat{\mu}_t =\frac{\mathbf{S N R}_T}{\mathbf{S N R}_t} \frac{\alpha_t}{\alpha_T} \mathbf{x}_T+\alpha_t \mathbf{x}_0\left(1-\frac{\mathbf{S N R}_T}{\mathbf{S N R}_t}\right) $
> >
> > $\hat{\sigma}_t^2 =\sigma_t^2\left(1-\frac{\mathbf{S N R}_T}{\mathbf{S N R}_t}\right)$
> >
> > and the equation from [Peluchetti2021].
> >
> > $dx_t = \mu(x_t,t)dt + g(x_t,t)dw_t$
> >
> > $\mu\left(x_t, t\right)=f\left(x_t, t\right)+G\left(x_t, t\right) \underbrace{\int \nabla_{x_t} \ln p_{\tau \mid t}\left(x_\tau \mid x_t\right) \frac{p_{t \mid 0, \tau}\left(x_t \mid x_0, x_\tau\right)}{\pi_t\left(x_t\right)} \Pi_{0, \tau}\left(d x_0, d x_\tau\right)}_{A\left(x_t, t\right)}$
> >
> > $\pi_t\left(x_t\right)=\int p_{t \mid 0, \tau}\left(x_t \mid x_0, x_\tau\right) \Pi_{0, \tau}\left(d x_0, d x_\tau\right)$
> >
> > Note that these serve fundamentally different functions in the paper:
> >
> > - Our equation serves as a **starting point** for our analysis into connections with existing diffusion model literature through the score function.
> >
> > - The equation from [Peluchetti2021] is a general time reversal that is described using a “multiplicative drift adjustment factor” $A$. It is only later in equation 23 that [Peluchetti2021] mentions some connections with score matching for efficient training.
> >
> > As such, our SDE is much more suited for our purposes (analysis with existing diffusion methods), with explicit reverse-time construction and connection with SNR schedules, etc., while any such attempt from [Peluchetti2021] must necessarily first rewrite into our proposed form (which the commenter seems to agrees with, as they emphasize that they can perform such a rewrite). Additionally, by writing it this way, we can also draw a connection with FM-OT and RF that the commenter incorrectly claims is impossible for the forward-time perspective.
> >
> >
> > > In the same way, the training objective (9) is also identical to the training objective of Equation (24) from [Peluchetti2021], provided that x0 is sampled as well, in which case the neural network needs to take as input x0.
> >
> > It is rather disingenuous to claim that one can **modify** the training and loss and architecture from [Peluchetti2021] (by sampling $x_0$ and inputting it into the network and regressing a new quantity which was not done originally) to be the same as our training objective in Theorem 2. Note that this edit (adding in the $x_0$) was done in the original comment but is not present in the paper. This is especially true since our parameterization with $x_0$ is crucial for understanding it as a denoising value.
> >
> > As a result, our actual loss function is not the simple score-matching loss presented in Theorem 2. We reparameterized it to predict $x_0$ in the EDM style with newly derived and optimized scaling terms $c_\text{skip}(t)$, $c_\text{output}(t)$, $c_\text{input}(t)$, etc., and the crucially different loss weighting term $w(t)$ (Section 4), which are essential for empirical success (see ablation studies). For these choices, it is an overclaim to say that one can “directly” or “easily” rewrite the loss in [Peluchetti2021] to arrive at the same thing, even if one assumes that adding the conditioning on $x_0$ can be done for free.

---

> ### Comment · Reviewer_BgL3 · 2023-11-21
>
> Thank you for these comments, I agree with a lot and have many of the same questions / concerns.

---

> ### Author Response · Authors · 2023-11-22
> **Additional response (Part 3)**
>
> > Then the resulting algorithms are identical, for training and sampling. I struggle to see where mentioned the mentioned difficulty is: it cannot possibly be the need to compute T - t.
>
> Again, it’s very disingenuous to state that the commenter “struggles to see where the mentioned difficulty is" after they modify the construction from [Peluchetti2021] to explicitly mimic our highlighted SDE and alter the existing training objective to be ours. The **entire point** is that our proposed formulation makes it easier to connect to existing diffusion model design choices. If the commenter must go through our formulation to demonstrate this connection (and presumably, this was not done for the original submission of [Peluchetti2021] as it was not obvious), then this implies that the original formulation is insufficient for developing this line of thinking.
>
> Furthermore, given the clarification on the ODE below, it’s clear that our reverse-time perspective allows for further derivations/connections with existing models.
>
> > If anything, the forward time construction makes it clear that ODE (7) is not a probability flow ODE matching (the marginal distributions of) SDE (6), which might not appear obvious in the reverse time construction.
>
> We stress that (and also updated the text to include this) that our proposed deterministic process is more technically involved than simply defining a deterministic initial condition and dynamics. In particular, the introduction of randomness comes from the fact that our ODE is only well-defined on $0 \le  t < T$, as Doob’s h-function causes a singularity at the boundary $T$. When sampling using the SDE, we need to approximate $x_{T - \epsilon} \approx y$ and follow the backwards SDE. For the ODE, the source of randomness comes from the initial distribution $p(x_{T - \epsilon})$, which is not the same as $y$. Instead, we sample $x_{T - \epsilon}$ (specifically approximating $x_{T - \epsilon’} \approx y$ and then taking an Euler-Maruyama back to $T - \epsilon$), with which we can then sample with the valid probability flow ODE. Note that this clears the singularity and injects randomness while enabling ODE sampling.
>
> We additionally find that with our higher-order sampler with stochasticity, the generation quality further improves, and this is likely due to fact that the first-step stochasticity (as proposed above) is not distinguishing enough for visual variability and sharpness due to numerical reasons (as opposed to being flat-out theoretically incorrect).
>
>
>
> > Clarification on the relationship with OT-Flow-Matching and Rectified Flow.
>
> In the noiseless limit, the magnitude of the score $\nabla_{x_t} \log q(x_t | x_T)$ (Eq. 10) goes to infinity because $\sigma_t \rightarrow 0$, so we can no longer match this term directly with score-matching. Although the presented score-matching model cannot handle the noiseless limit case, we can design new losses to predict the combined $-g^2(t)(1/2 s(x_t,t,y,T) - h(x_t,t,y,T))$, which we’ve shown in the appendix reduces to $x_T - x_0$ (special case 2). Our framework subsumes this in the sense that the ODE drift terms becomes the same terms, e.g.  $x_T - x_0$. In addition, thanks to the generalized parameterization of networks in Section 4, we can design our score model to be $s(x_t, x_T, t; \theta) = A x_t +B  x_T + C V(x_t, t; \theta)$ where $V$ is the neural network with parameter $\theta$ and simply choose $A=B=0$, $C=1$ and match it against $x_T - x_0$ with loss $\mathbb{E}_{x_t,t,x_0,x_T}[ \lVert s(x_t,x_T, t; \theta) - ( x_T - x_0) \rVert^2] $.
>
> The above to loss then reduces to $\mathbb{E}_{x_t,t,x_0,x_T}[\lVert V(x_t, t; \theta) -  (x_T - x_0) \rVert^2]$
>
> where $s$ and $V$ are parameterized by $\theta$ and the loss becomes loss of OT-Flow-Matching and Rectified Flow.
>
> > This is not correct, the proposed method does not subsume OT-Flow-Matching and Rectified Flow. Example: (example given)
>
> (Note that the commenter means $x_T$ for our framework, not $x_0$). From the above clarification, we note that we can simply drop the reliance on $x_T$, which recovers the correct Rectified Flow path.

---

> ### Author Response · Authors · 2023-11-22
> **Additional response (Part 4)**
>
> > In [Peluchetti2021] both VE and VP parametrization are considered, see Equations (11) and (12) in Section 4. The formulations are slightly more general as they cover non-diagonal diffusion coefficients as well.
>
> While we do agree that [Peluchetti2021] considered both of these noise schedules, we emphasize that they are also readily apparent from existing work (in physics, statistics, diffusion, etc…). Thus, we generally believe that claiming theoretical novelty for either noise schedule to be inappropriate since it is “reinventing the wheel” (this is akin to claiming novelty for a VE/VP SDE in the original diffusion).
>
> Instead, these parameterizations are important for their empirical properties, as such this is the axis that we evaluate on. For this purpose, we note that our VE/Brownian bridge is not very novel since existing work has already shown that such a scheme can work on complex high dimensional tasks (as such we never claimed novelty here). Conversely, for the vastly underexplored VP parameterization, we believe that we are the first to derive a recipe (e.g. how to pick schedule, values, etc…) to make it work in these settings. In particular, this is critical for most of our experimental section. As such, our work is the first to thoroughly explore this parameterization.
>
> As for the commenter’s claims, we want to explicitly point out that the comments are directed primarily at the first “theoretical” axis. This is seen in the phrase “the formulations are slightly more general as they cover non-diagonal diffusion coefficients”, which is ultimately an appeal to mathematical generalization. Furthermore, it is highly unlikely that the commenter is appealing to the empirical results of [Peluchetti2021], as we don’t believe that empirical results were presented for the VP case (although a toy example is given for the VE case). As such, we again reiterate our view that, for machine learning tasks, empirical results matter much more as the theory tends to be well-developed in an alternative fields (like physics, statistics, etc…).
>
> However, we recognize that [Peluchetti2021] has indeed previously proposed these schedules, and will mention this in our updated draft.
>
>
>
> > Surely, given all the similarities between [Peluchetti2021] and the current proposal a different kind of acknowledgement could be employed.
>
> While we understand that the commenter may have frustrations about proper contextualization of their prior work, we were taken aback by their tone. We hope that this is not intentional, but the tone comes off as condescending and unprofessional. Fundamentally, while there are some similarities in theoretical setup (which is natural, as all of the papers in this field are built off of similar principles), the smaller differences constitute the majority of each paper (as again, this is a fundamentally empirical field). We were also dismayed that the commenter seems to hold this attitude towards many important works in the field (such as [6], which is described as “almost identical to” their own work despite a clear difference in formulation and a convincing demonstration on a variety of translation tasks, which was not shown in [Peluchetti2021]). Ultimately, we feel that this perceived attitude is counterproductive to a fruitful discussion.
>
>
> Despite this, we can agree that [Peluchetti2021] can derive a similar formulation using the forward time perspective. However, we want to emphasize that there do exist core differences that separate the two works. In particular, our reverse-time derivation affords us significantly more empirical (through network parameterization + sampling) and theoretical (through the ODE and connections with OT-FM + RF) benefits.  As such, these “similarities” are only warranted at first glance, and one should not directly pass off our paper because of it, especially when it comes to elucidating the connection with diffusion models and producing good empirical performance.
>
>
>
>
> [1] Kingma, Diederik, et al. "Variational diffusion models."
>
> [2] Karras, Tero, et al. "Elucidating the design space of diffusion-based generative models."
>
> [3] Hoogeboom, Emiel, Jonathan Heek, and Tim Salimans. "simple diffusion: End-to-end diffusion for high resolution images."
>
> [4] Lu, Cheng, et al. "Dpm-solver: A fast ode solver for diffusion probabilistic model sampling in around 10 steps."
>
> [5] Lu, Cheng, et al. "Dpm-solver++: Fast solver for guided sampling of diffusion probabilistic models."
>
> [6] Liu et al, “I2SB: Image-to-Image Schro ̈dinger Bridge”
>
> [7] Salimans, Tim, and Jonathan Ho. "Progressive distillation for fast sampling of diffusion models."

---

> > ### Public Comment · ~Stefano_Peluchetti1 · 2023-11-23
> > **Final comments [part 1]**
> >
> > > While we understand that the commenter may have frustrations about proper contextualization of their prior work, we were taken aback by their tone. We hope that this is not intentional, but the tone comes off as condescending and unprofessional.
> >
> > I apologize with the Authors that this was how my comment came across, this was definitely not my intention.
> >
> > > Unfortunately, many of the points are incorrect and are broad overclaims of the commenter’s previous work. As the commenter mentioned, many of these conclusions were drawn from just “a quick glance”. We would like to kindly ask the commenter to refrain from rushing to hasty judgments without a proper check, as, although commenters do not have the same obligations to correctness as reviewers, their words are often weighted similarly.
> >
> > In truth, the "quick glance" I wrote only applies to the generic case of SDE (6) when $s(⋅)$ does not take $x_T$ as an argument.
> > While I cannot guarantee the absence of mistakes from my arguments, which also is why I started with "Firstly, I detail my own understanding, in case I am mistaken", I can assure the readers that I gave considerable thought and spent considerable time in writing down my detailed comments.
> > That said, I am not a reviewer, and I could not find the time to also carry out the computations required for a toy example to figure out whether $x_0$ follows the data distribution or not in SDE (6) when $s(⋅)$ does not take $x_T$ as an argument and $σ > 0$ (this is what I mean by "generic case").
> > I am worried that it does not, in which case the coupling $q_\mathrm{data}$ would not be preserved, and I could not find results in this work to support the fact that it would.
> >
> > I also maintain disagreement with some of the additional Authors' replies regarding differences between the forward-time construction with Doob’s h-transform and their proposal, and more in general with the way their proposal is portrayed from a methodological point of view.
> > However, I will refrain from discussing further technical aspects regarding the proposed work, as Reviewer BgL3 noted is it not my place to push too much.
> > We will have to agree to disagree.
> >
> > I just want to stress one specific point.
> > As Reviewer BgL3 also mentions, including or not $x_T$ is not an implementation detail.
> > It is what allows the coupling to be preserved in the proposal of this work.
> >
> > > We were also dismayed that the commenter seems to hold this attitude towards many important works in the field (such as [6], which is described as “almost identical to” their own work despite a clear difference in formulation and a convincing demonstration on a variety of translation tasks, which was not shown in [Peluchetti2021])
> >
> > Quoting my own comment, when referring to [6] and related works:
> >
> > > I do not wish to detract in any way from the otherwise excellent contributions of these works, which include exhaustive empirical validations and additional theoretical findings and insights due to the introduced modeling frameworks.
> >
> > My claim regarding [6] is that the resulting method, derived in a completely different way, is equivalent to the DBM transport ($M$) modulo the previously reported differences.
> > This can be verified from the Equations' and Algorithms' numbers I provided for both works.
> > Based on the best of my knowledge, I do claim novelty over the definition of ($M$).
> > Other than that, I fully acknowledged the further contributions, and I did so with very positive wording (**excellent contributions**).
> >
> > > As for the commenter’s claims, we want to explicitly point out that the comments are directed primarily at the first “theoretical” axis.
> >
> > This is correct, my comments pertains only and exclusively to the methodological side of this work.
> > Indeed, when I summarized my notes on this work I explicitly wrote "In conclusion,**on the methodological side**:..." (emphasis added only now). Which I complemented with (quoting myself again):
> >
> > > As previously stated, the idea of applying the "point to distribution" diffusion for conditional modeling is (to the best of my knowledge) novel and has merit. Moreover, this work presents additional contributions in Sections 4, 5 and 7, deriving an improved sampler, adapting many recent advancements in SDE-based generative modeling, and carrying out an extensive empirical assessment of the proposed approach for various applications. These are all interesting and valid contributions on their own, and are required to demonstrate competitive results in generative visual applications.
> >
> > fully acknowledging also all the other contributions.
> > While I am not a reviewer, and thus these additional positive comments on the quality of this work were not due, I thought their inclusion to be a nice gesture to smooth the tone of what could have been otherwise been perceived as an overly critic take on the presented work.
> > It is certainly not my intention to downplay the efforts required to achieve good empirical performance.

---

> ### Public Comment · ~Stefano_Peluchetti1 · 2023-11-23
> **Final comments [part 2]**
>
> > {{When talking about the theoretical focus}} This is seen in the phrase “the formulations are slightly more general as they cover non-diagonal diffusion coefficients”, which is ultimately an appeal to mathematical generalization.
>
> The motivation for considering non-diagonal diffusion coefficients was (i) to consider a noise distribution resembling some properties of the data distribution, hoping that this could help empirically and (ii) to pave the way to the infinite-dimensional, or functional, formulation of the transport, which is hinted at in Section 8.
> Unfortunately, due to page restrictions, (i) have been relegated to Appendix C.3 of [Peluchetti2021] during the rebuttal, to make space for the toy experiment of Section 7.
> As for (ii), it has never been pursued further by myself, as various concurrent works developed this perspective first some months later.
>
> > While we do agree that [Peluchetti2021] considered both of these noise schedules, we emphasize that they are also readily apparent from existing work (in physics, statistics, diffusion, etc…). Thus, we generally believe that claiming theoretical novelty for either noise schedule to be inappropriate since it is “reinventing the wheel” (this is akin to claiming novelty for a VE/VP SDE in the original diffusion).
>
> We could not be more in agreement, I never claimed any novelty over VE/VP.
> Indeed, in Section 4 of [Peluchetti2021] I show that VE/VP parametrizations are time-wrappings of the Brownian Motion and of the Ornstein-Uhlenbeck process, as they are more commonly known in the stochastic process literature.
>
> On the same note, I also never claimed any novelty on the "point to distribution" process ($F$).
> In appendix C.1 of [Peluchetti2021] (Closely Realted Work) I noted:
>
> > For an initial $x_0$, i.e. for case of $A(x_t, t, x_0)$ in Section 3.2 and for the more limited dynamics considered in Wang et al. (2021), the achieved transport is the same.
>
> > It is rather disingenuous to claim that one can **modify** the training and loss and architecture from [Peluchetti2021] (by sampling $x_0$ and inputting it into the network and regressing a new quantity which was not done originally) to be the same as our training objective in Theorem 2. Note that this edit (adding in the $x_0$) was done in the original comment but is not present in the paper.
>
> This was acknowledged (disingenuity aside), I wrote "...this is identical to the training objective (9) **for a given
> $x_0$**..." (emphasis added only now), and one of the conclusions of my notes stated:
>
> > unless I missed prior relevant works, the novelty is in the application of this transport to conditional modeling of the data distribution when alignment (paired data) is available, that is by employing the transport over multiple initial values $x_0$
>
> It is correct to state that $x_0$ does not enter the neural network, nor is sampled, in [Peluchetti2021], as the focus there was unconditional generation and $x_0$ was fixed.
>
> For full disclosure, and to contextualize my own comments which might appear bitter to the readers, I did reach out to one of the Authors privately a few weeks back (as I am not a reviewer I am not bound to the doubly-blind requirements).
> This did not elicit a reply, which I understand is not owed, and prompted my public comment.
> It goes without saying, but I am not particularly enjoying this public debate either.
> Despite differences of opinions, I wish the Authors the best of luck with this submission.

---

### Public Comment · ~Huanran_Chen1 · 2023-11-18
**Several question about this paper**

This paper is both interesting and insightful. It has greatly motivated me and provided numerous insights. I have a few questions:


- Can we directly derive Eq. (6) from Eq. (5) using the method described in Anderson et al. (1982)? By considering $g(t)^2h(x_t,t,y,T)$ as part of the drift term, is it possible to arrive at the reverse-time SDE directly? This approach could potentially circumvent the need to derive Eq. (6) using the Fokker-Planck equation.

- One major challenge in applying the concepts of this paper to real-world situations seems to be the need for paired data. Is there a possibility that we could start with a traditional diffusion model and then perform minimal fine-tuning on a smaller dataset with paired data for image-to-image transformations, and then perform SDEedit? t appears to me that DDBM could still work well with SDEdit. Maybe just sligntly fine-tuning $s(x,t,y,T)$ could shift the predicted score from $\nabla_{x_t} \log p(x_t)$ to $\nabla_{x_t} \log p(x_t|x_T)$ a little bit, and leading to a better result?

- Lastly, and most importantly, when is the code release planned? I am very excited to explore and experiment with your model!





**Reference:**

Anderson, Brian DO. "Reverse-time diffusion equation models." Stochastic Processes and their Applications 12.3 (1982): 313-326.

---

### Author Response · Authors · 2023-11-19
**Author Rebuttal Summary**

We thank all reviewers and additional comments for their time and thoughtful comments. We want to address a common concern regarding properly addressing prior works and contextualizing our contribution. **We have incorporated all the suggestions and references in our new related works which can be found in new supplementary files**, with which we plan to substitute the section in the main text. For each reviewer raising the concern, we have listed bullet points addressing the mentioned works and important differences with our work in the respective comments. For clarity, we present them together here,

- [1] proposes to learn an OT mapping between two different distributions by approximating the time-invariant velocity field, while our method is not based on OT. [1] is based on an ODE, while we have an SDE for distribution mapping, which in the noiseless limit, reduces to that in [1] in a special case. Our work enables a higher-order sampler and output parameterization that are not directly usable for [1].
- [2,3] are flow-based and aim to learn transport maps to push forward a prior distribution to data distribution, and utilizes deterministic ODEs for generation. Our theory is developed using SDE and we introduce higher-order sampler with additional stochasticity, which are also not directly usable for these works. The resulting method can further subsume the case in [3] developed with OT displacement map.
- [4] is a general theory that directly constructs a bridge using an interpolation map and avoids the use of Doob’s h-function. Our method is based on Doob’s h-function from which a direct interpolation map can be constructed for marginal sampling. Furthermore, the training objective is quite different as ours is based on denoising score-matching without any regularization, while theirs is not. Our construction also shows stronger tie with diffusion models, which allow extending existing designs for the new bridge construction. These include reusing noise schedules, speicalized higher-order sampler, and network preconditioning. It’s not clear if these choices can be directly applied to [4].
- Extending [3], [5,6] are similarly flow-based methods that show success on exploiting potential couplings between the two distributions via minibatch simulation-free OT plan. Again, our method is fundamentally based on SDE and does not aim to solve OT. Our simulation free sampling is directly constructed from VP and VE diffusion processes. [3,5,6] all consider Brownian bridge paths as as the interpolation map, while we show that this straight interpolation map is a special case of VE bridge. VP bridge exhibits curved interpolation paths. Nevertheless, when implemented correctly, we show both bridges can perform well in practice.
- [7] constructs a simulation-free training plan using Brownian bridge as the interpolation map, and aims to match against both flow vector field and bridge score. Our method does not only consider Brownian bridge as the bridge construction, and shows success for other types of bridge in practice. We only consider a denoising score-matching loss without additional flow-matching loss as proposed. A set of practical design choices are also proposed to further improve quality and speed of our bridge model.

- [8] learns to bridge two distributions by matching Doob’s h-function and proposes a simulation-free training procedure for training. [11] proposes a similar approach which learns a generative model by constructing a mixture of bridges where the intermediate $x_t$ can be marginally sampled. Both works propose to generate via forward-time simulation with the trained model. Our method instead adopts a reverse-time perspective. The key benefit from this perspective is the ability to integrate key diffusion model techniques into the bridge framework, such as an accurate higher-order sampler and network preconditioning. Since these previous works are developed with other perspectives, it is less obvious how one would integrate these diffusion model design choices and achieve empirical improvement. We also extend beyond the Brownian bridge construction (as used in [8,9,13,14], which is a special case of VE bridge) and show that both VE and VP bridge can achieve competitive results than state-of-the-art.

- [9] proposes to connect two image domains via discrete Brownian bridges, and proposes to reverse the bridge for generation. This is a discrete-time special case of the VE bridge we consider, and our method also shows success on VP bridges that work well in continuous time. In addition, our sampler takes much fewer steps for best generation results.

---

> ### Author Response · Authors · 2023-11-19
> **Author Rebuttal Summary (continued.)**
>
> - [12] adopts a reverse-time perspective of diffusion bridges pinned on both ends and proposes score-matching for inference time simulation. Our method considers a more general case where both ends are drawn from distributions rather than fixed, and our results are developed for this case. In addition, [12] requires simulation of an entire trajectory for score-matching training, while our method extends directly from diffusion models and naturally results in a simulation-free training procedure
> - [13] is also built on learning Doob’s h-function, similar to [8]. However, they also propose to perform forward time simulation during training for matching against the Doob’s h-function, which we avoid.
> - [14] proposes bridge-matching and Iterative Markovian Fitting, an iterative procedure for solving the Schrodinger bridge problem. This procedure involves an inner loop of optimization and simulation, with the exception of the first iteration where the intermediate $X_t$ can be marginally sampled given endpoints. Our method is different in that our method is completely simulation-free, and directly extend diffusion models for the bridge construction. We further discuss connections in the main response.
> - [15] is a normalizing-flow based generative model which relies on deterministic ODEs for generation. Our method is instead based on SDEs, and we also show the introduced stochasticity in our sampler is integral for empirical success. In the noiseless limit of our method with the VE bridge, our method reduces to the method introduced in [15].
> - [16] is an image-to-image method based on a special case of SB which is tractably computable and also results in a simulation-free algorithm. Our method is functionally similar but theoretically different. Different from [16], our method can theoretically subsume other classes of models and directly adapt successful design choices to improve quality and speed. We also empirically outperform them on image translation.
>
>
> To reiterate, our contribution lies in designing a simulation-free bridge framework which can directly extend many successful design choices of diffusion models and enjoy nice theoretical properties. It shows that both VP bridge and VE bridge built from the corresponding diffusion processes can achieve good empirical results, while prior works mostly use Brownian bridge (such as in [5,6,8,9,13,14]), which is a special case of VE bridges. Our framework also elucidates connection with the underlying diffusion processes, pushes state-of-the-art bridge-based image translation, and retain strong performance for unconditional generation.
>
> We address additional concerns individually in the comments. Thank you again for your insights.
>
>
> [1] Flow Straight and Fast: Learning to Generate and Transfer Data with Rectified Flow. Xingchao Liu, Chengyue Gong, Qiang Liu, Sept 2022.
>
> [2] Building Normalizing Flows with Stochastic Interpolants. Michael S. Albergo and Eric Vanden-Eijnden, Sept 2022.
>
> [3] Flow Matching for Generative Modeling. Yaron Lipman, Ricky T. Q. Chen, Heli Ben-Hamu, Maximilian Nickel, Matt Le, Oct 2022.
>
> [4] Stochastic Interpolants: A Unifying Framework for Flows and Diffusions. Michael S. Albergo, Nicholas M. Boffi, Eric Vanden-Eijnden, March 2023.
>
> [5] Improving and Generalizing Flow-Based Generative Models with Minibatch Optimal Transport. Alexander Tong, Nikolay Malkin, Guillaume Huguet, Yanlei Zhang, Jarrid Rector-Brooks, Kilian Fatras, Guy Wolf, Yoshua Bengio, Feb 2023.
>
> [6] Multisample Flow Matching: Straightening Flows with Minibatch Couplings. Aram-Alexandre Pooladian, Heli Ben-Hamu, Carles Domingo-Enrich, Brandon Amos, Yaron Lipman, Ricky T. Q. Chen, April 2023.
>
> [7] Simulation-free Schrödinger bridges via score and flow matching. Alexander Tong, Nikolay Malkin, Kilian Fatras, Lazar Atanackovic, Yanlei Zhang, Guillaume Huguet, Guy Wolf, Yoshua Bengio, July 2023.
>
> [8] Somnath et al 2023, Aligned Diffusion Schrödinger Bridges
>
> [9] Li et al 2022, BBDM: Image-to-Image Translation with Brownian Bridge Diffusion Models
>
> [10] Su et al. 2022, Dual Diffusion Implicit Bridges for Image-to-Image Translation
>
> [11] Peluchetti 2022, Non-Denoising Forward-Time Diffusions
>
> [12] Heng et al. Simulating Diffusion Bridges with Score Matching
>
> [13] Liu et al., 2022, Let us Build Bridges: Understanding and Extending Diffusion Generative Models
>
> [14] Shi et al Diffusion Schrödinger Bridge Matching 2023
>
> [15] Lipman et al 2022, Flow Matching for Generative Modeling
>
> [16] Liu et al 2023, I2SB: Image-to-Image Schro ̈dinger Bridge

---

### Public Comment · ~Kaiwen_Zheng2 · 2023-12-05
**Welcome to read our paper Bridge-TTS for similar techniques and their application on Text-to-Speech Synthesis**

Congrats on your great work DDBM and the high rating (8866) it got! I am one of the co-first authors of [Bridge-TTS](https://openreview.net/forum?id=F9ApWtHVac), which is also submitted to ICLR2024. In this work, we made many similar explorations to DDBM such as introducing noise schedules to bridges and proposed novel sampling schemes. We focus on the Text-to-Speech Synthesis (TTS) task and demonstrate that bridges significantly outperform diffusion models. However, the reviews in this area seem more focused on the tricky tuning of fancy network architectures and losses, and are unaware and ignorant of the theoretical contributions. Unfortunately, we got ratings of 5555 and have withdrawn. Still, we think Bridge-TTS is a "TTS" version of DDBM and hope readers can get inspiration.

---

### Meta-Review · Area_Chair_rGVJ · 2023-12-09

**Metareview:**

This submission introduces denoising diffusion bridges that learn the score of the diffusion bridge by interpolating between two paired distributions given as endpoints. The reviewers originally acknowledged the completeness of the experiments, however they raised concerns regarding incremental novelty and the lack of in-detail discussion of closely related works. The rebuttal discussion including with expert reviewers and public commenters shed light on this submission and helped the authors better contextualize their contributions. Albeit the concerns, overall the reviewers are positive and thus we are happy to recommend acceptance.

**Justification For Why Not Higher Score:**

Although the rebuttal contextualized the work better w.r.t. prior arts, the overall contributions seem rather incremental.

**Justification For Why Not Lower Score:**

All the reviewers are positive and recommend accept.

---

### Decision · Program_Chairs · 2024-01-16

Accept (poster)